# Can LLMs Reason Structurally? An Evaluation via the Lens of Data Structures

## Abstract

As large language models (LLMs) take on increasingly complex tasks, understanding their algorithmic reasoning abilities has become essential. However, existing evaluations focus on distinct and isolated tasks. We propose a unified diagnostic lens: *structural reasoning*—understanding and manipulating relationships like order, hierarchy, and connectivity. We introduce `DSR-Bench`, the first benchmark to systematically evaluate LLM structural reasoning through canonical data structures, which serve as interpretable, algorithmically meaningful abstractions. `DSR-Bench` spans 20 data structures, 35 operations, and 4,140 synthetically generated problem instances with minimal contamination. The benchmark's hierarchical design pinpoints specific failure modes, while its fully automated evaluation ensures objective and consistent assessment. Benchmarking ten state-of-the-art LLMs reveals critical limitations: the top-performing model scores only 0.498 out of 1 on challenging instances. Three additional evaluation suites reveal further weaknesses: models perform poorly on spatial data and natural language scenarios, and fail to reason over their own generated code. `DSR-Bench` offers a principled diagnostic tool for structural reasoning, helping expose reasoning bottlenecks and guide the development of more capable and reliable LLMs.

## 1 Introduction

As large language models (LLMs) tackle increasingly complex real-world challenges, strengthening their *algorithmic reasoning* abilities is essential for interpretability, safety, and efficiency (Eberle et al., 2025). Yet scaling alone is reaching its limits: models face data scarcity and diminishing returns (Villalobos et al., 2024). A complementary path is to study the algorithms LLMs learn and employ, which can reveal their internal representations and inspire algorithm-centric architectures that embed reasoning capabilities directly into model design (Bounsi et al., 2024; Eberle et al., 2025).

The call to prioritize algorithmic reasoning has intensified as the field shifts toward evaluating *general reasoning* without external tools. While LLMs can solve many tasks through code generation and execution, recent initiatives like Gemini-Deep-Think's and OpenAI's IMO competitions (Luong & Lockhart, 2025; Wei, 2025) explicitly prohibit coding and proof assistance. These efforts highlight a growing emphasis on end-to-end reasoning as a critical step toward artificial general intelligence.

Despite rapid progress, existing evaluations of algorithmic reasoning remain fragmented, each probing only a single facet, such as arithmetic operations (Zhou et al., 2022; 2024; Lee et al., 2024), simulation of textbook algorithms (Markeeva et al., 2024), or domain-specific tasks like graph problems (Wang et al., 2023a; Fatemi et al., 2024). Meanwhile, interpretability research, such as mechanistic interpretability (Olah et al., 2020; Wang et al., 2023b), examines internal model representations rather than reasoning over algorithmic structures. What is missing is a unified framework that bridges these threads, offering interpretable diagnostics of algorithmic reasoning across diverse problem types.

To address this gap, we propose to evaluate algorithmic reasoning through *the lens of data structures*—fundamental abstractions that organize information and govern algorithmic operations. Data structures provide an ideal evaluation framework because they are both systematic (covering diverse relationship types) and interpretable (offering clear diagnostic insights). They span the spectrum of relationships underlying algorithmic thinking: arrays represent sequences, stacks and queues capture temporal ordering, trees encode hierarchies, and graphs represent complex networks. This diversity

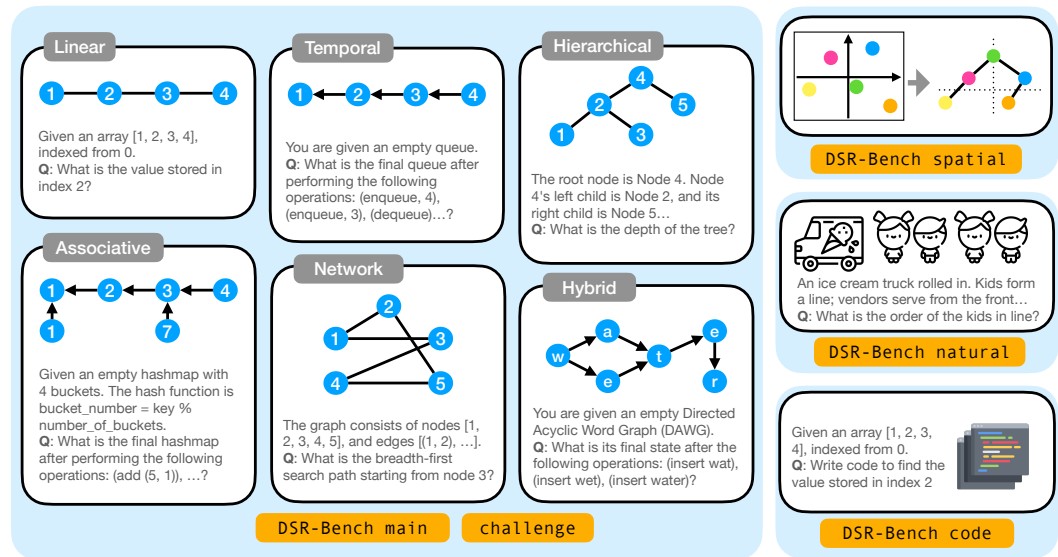

Figure 1: Overview of `DSR-Bench`'s `main` suite with six data structure categories capturing distinct relationships, plus the `challenge` subset. Three specialized suites that holistically evaluate structural reasoning under different settings: `spatial` (multi-dimensional data), `natural` (realistic natural language scenarios), and `code` (code generation).

enables us to holistically assess whether models can construct, maintain, and manipulate different types of relationships—the core capability we term *structural reasoning*.

Structural reasoning underlies many real-world applications. Trip planning requires interpreting maps as graphs and managing priorities with queues, while supply chain optimization depends on hierarchical resource allocation and temporal sequencing. In safety-critical domains such as robotics (Sado et al., 2023) and healthcare (Sadeghi et al., 2024), failures in structural reasoning abilities can have serious consequences, making systematic evaluation crucial before deployment.

**Contributions** We introduce `DSR-Bench`, the first benchmark to systematically isolate and evaluate LLM structural reasoning through data structure manipulation tasks. `DSR-Bench` comprises 4,140 problem instances spanning 20 data structures (grouped into six relationship categories; see Figure 1) and 35 operations across three difficulty levels (*short*, *medium*, and *long*) and five evaluation suites (`main`, `challenge`, `spatial`, `natural`, and `code`). Each task provides interpretable, automatically verifiable assessments of whether models can construct, maintain, and reason about relationships. The highlights and main takeaways of `DSR-Bench` are:

- **Hierarchical and diagnostic design.** Simpler tasks (e.g., queue operations) serve as prerequisites for complex ones (e.g., graph traversal). This enables fine-grained localization of failure modes.
- **Deterministic, contamination-resistant evaluation.** Synthetic data minimizes contamination risk. The evaluation is deterministic and fully automated, supporting objective and fair assessment.
- **Five evaluation suites covering diverse reasoning settings.**
  - `main`: Canonical data structure tasks reveal core reasoning gaps—e.g., instruction-tuned models struggle with multi-attribute and multi-hop reasoning, while reasoning models fail to override memory for non-standard tasks with user-defined constraints. Simpler prompts consistently help, but CoT requires careful design.
  - `challenge`: Hybrid and compositional structures remain difficult, with the best model scoring only 0.498 out of 1, revealing reasoning limitations of frontier models.
  - `spatial`: On multi-dimensional data, the common format found in real-world applications, models struggle as dimensionality increases or when data follows non-uniform distributions.
  - `natural`: In realistic natural language scenarios, models struggle to extract structure and navigate ambiguity, exposing a gap to real-world deployment.

- ○ `code`: An auxiliary probe contrasting reasoning with and without tool use. Models rarely benefit from reasoning over their own generated code; external interpreters help on familiar tasks but consistently fail on non-standard or natural-language variants.
- **Comprehensive empirical analysis.** We evaluate ten state-of-the-art LLMs and analyze prompting strategies, distribution shifts, and qualitative error patterns (e.g., implicit priors, instruction-following failures). These findings provide actionable insights for developing architectures and training methods that better capture algorithmic reasoning.

To welcome community engagement and collaboration, we release all code and datasets at `https://anonymous.4open.science/r/DSR-Bench-C40D` for full reproducibility.

## 2  RELATED WORK

**Algorithmic reasoning with LLMs**   Prior works target distinct but isolated facets of algorithmic reasoning, often limited in scope: arithmetic tasks and length generalization in transformers (Zhou et al., 2022; 2024; Lee et al., 2024), or specific graph problems like cycle detection and connectivity (Wang et al., 2023a; Fatemi et al., 2024). CLRS-Text (Markeeva et al., 2024) evaluates whether LLMs can simulate 30 classical algorithms (Veličković et al., 2022; Cormen et al., 2009). We take a holistic and foundational approach using data structures, which serve as interpretable building blocks of algorithms and support fine-grained diagnosis of specific reasoning failures to inform future research.

**Reasoning benchmarks**   Existing LLM reasoning benchmarks are predominantly high-level and domain-specific, targeting math (Cobbe et al., 2021; Liu et al., 2024a;b), STEM (Hendrycks et al., 2021), and logic puzzles (White et al., 2025; Giadikiaroglou et al., 2024). These often require complex responses with intertwined reasoning steps, relying on subjective human or LLM-based evaluation (Chiang et al., 2024; Feuer et al., 2024; Ye et al., 2024). We focus on structural reasoning, an implicit requirement underlying problem-solving across domains. By using data structures as clear abstractions of these relationships, we isolate structural reasoning from domain-specific complexities.

**Coding benchmarks**   Coding benchmarks evaluate how well LLMs write syntactically correct code or function as coding agents (Chen et al., 2021; Zheng et al., 2023; Jimenez et al., 2024; Jain et al., 2025; White et al., 2025; Aider-AI, 2025), typically requiring external interpreters for verification. While useful, these benchmarks conflate reasoning with tool execution and are limited to domains where coding applies. In contrast, we target *general reasoning* independent of external tools, reflecting the broader goal of assessing progress toward general intelligence. Prior works (Malfa et al., 2024; 2025; Liu et al., 2025) study code simulation as a lens to probe general reasoning. Instead, we specifically focus on structual reasoning via data structure tasks, and include the `code` suite to probe whether code generation aids in such a process.

## 3  DSR-BENCH: THE DATA STRUCTURE REASONING BENCHMARK

In `DSR-Bench`, we propose a taxonomy of data structures grouped into six categories based on the types of data relationships they encode: Linear, Temporal, Associative, Hierarchical, Network, and Hybrid. This taxonomy captures the diversity of relational patterns found in real-world data, enabling systematic evaluation across distinct reasoning challenges.

### 3.1  TASKS

`DSR-Bench` includes the following data structures and tasks, further detailed in Section A.2.

- **Linear (Sequential):** This category includes ARRAY and its operations: access, insert, delete, reverse, and search. Linear structures introduce ordered relationships, enabling reasoning about position, sequence, and iteration. They serve as a foundation for more complex data abstractions.
- **Temporal (Time-based ordering):** Temporal structures include STACK, QUEUE, PRIORITY QUEUE, and SKIP LIST, which operate under last-in-first-out, first-in-first-out, priority-based, or probabilistic rules. Their operations are compound actions of insertions and deletions. Temporal reasoning is essential in systems that require ordered execution over time, such as event queues and schedulers.

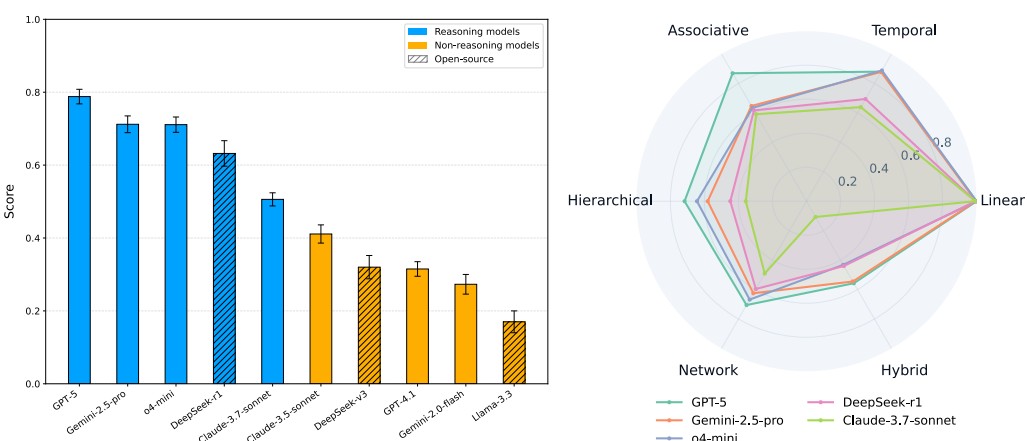

Figure 2: Left: Scores of ten models on `DSR-Bench-main`, averaged across three runs. Right: Radar chart showing scores of top-performing models across six data structure categories. We note `DSR-Bench` also includes a `challenge` suite, where the best model scores only 0.498.

- **Associative (Key-value mapping):** We evaluate HASHMAP, TRIE, and SUFFIX TREE through tasks focused on construction and compound insertion-deletion operations. Tasks test the ability to handle key-based insertions, retrievals, and pattern matching in nested or hashed structures. These skills are core for efficient lookup and structured access in systems such as databases.
- **Hierarchical (Tree-like):** This group includes BINARY SEARCH TREE (BST), HEAP, RED-BLACK (RB) TREE, and B+ TREE, with tasks involving traversals and compound operations. These structures are common in file systems and database indexing. Tasks assess whether the model can maintain structural invariants, perform updates efficiently, and simulate recursive behavior.
- **Network (Connectivity and group membership):** This category includes GRAPH and DISJOINT SET UNION (DSU) tasks. Graph tasks cover breadth-first and depth-first traversal, testing the ability to reason about connectivity and explore many-to-many relationships, as commonly found in social networks. DSU tasks evaluate union–find operations, which are essential for network connectivity analysis and clustering algorithms.
- **Hybrid (Combined relationships):** Real-world systems often rely on hybrid data structures that combine multiple forms of structural reasoning. This category includes LRU CACHE, which integrates temporal and memory management; BLOOM FILTER, for probabilistic set membership; and DIRECTED ACYCLIC WORD GRAPH (DAWG), a trie-like hierarchical graph. Tasks test whether models can compose different principles and generalize beyond individual structures.

**Hierarchical organization**    Tasks are organized by increasing complexity for fine-grained diagnosis of reasoning failures. For example, success on LIST but failure on HASHMAP isolates the issue to multi-attribute access rather than linear sequencing. Similarly, since TREE TRAVERSAL depends on QUEUE or STACK operations, testing these primitives becomes a prerequisite.

**Operations**    We design a diverse set of operation tasks for each data structure (full list in Section A.2), spanning three categories: construction, inspection (e.g., access, traversal), and manipulation (e.g., insert, delete). Beyond these *atomic* operations, `DSR-Bench` includes *compound* operations: sequences of manipulations (e.g., insert, insert, delete, ...) designed to test whether models can maintain structural integrity across multiple steps.

**Difficulty levels**    We assign tasks to three difficulty levels based on input length: *short* (5–10), *medium* (11–20), and *long* (21–30), to assess length generalization. For atomic operations, length refers to the number of input elements (e.g., list items, tree nodes). For compound operations, tasks begin with an empty structure, and length is defined by the number of sequential operations.

**Evaluation suites**   Beyond the `main` suite, we curated a `challenge` subset targeting particularly complex structures to stress-test advanced reasoning capabilities. To more comprehensively evaluate structural reasoning beyond canonical data structures, we introduce three additional suites: `spatial`, which evaluates reasoning on multi-dimensional data; `natural`, which tests reasoning when tasks are embedded in realistic natural language scenarios; and `code`, which assesses whether models can leverage code generation for structural reasoning. Each suite examines how well structural reasoning capabilities transfer to scenarios critical for real-world deployment.

## 3.2   PROMPT DESIGN

For each task, we design a prompt template and populate it with synthetic data to produce individual problem instances. Each prompt follows a consistent format: (i) a concise description of the data structure; (ii) a detailed explanation of the operations to be performed, written to avoid ambiguity; (iii) the initial state of the data structure (e.g., an existing tree or list) and any additional inputs required to execute the task (e.g., new elements to insert or delete); and (iv) a specific question requesting the final computed outcome. Following recommended practices in prompt engineering (OpenAI, 2025), we also append the instruction *"Answer the question in*

> **Example prompt for QUEUE compound.**
>
> A queue is a data structure in which items are added at one end and removed from the other, maintaining a first-in, first-out (FIFO) order. You should create a queue. There are two types of operations: (`enqueue, k`) adds `k` to the back. (`dequeue`) removes the front. You are given an empty queue initially.
> **Q**: What is the final queue after performing:
> - (enqueue, 49)
> - (dequeue)
> - ...
> Answer the question in 8000 tokens.

*<number> tokens"* to guide models toward producing concise outputs within a specified token budget. We also implement five different prompting strategies for each task, as we detail in Section A.3.

## 3.3   PIPELINE

**Data generation**   All data is synthetically generated to ensure efficiency, scalability, and minimal risk of training data contamination (Zhang et al., 2024a; Xu et al., 2024a). Each data structure and its operations are programmatically implemented to produce ground-truth outputs. Inputs are randomly sampled—numbers uniformly from 0–100 and strings from lowercase English letters. This modular approach enables accurate, reproducible evaluation and easy integration of new data structure types.

**Automated evaluation**   The evaluation pipeline is fully objective and reproducible through automated and deterministic verification. This avoids subjective judging in proof-writing or math benchmarks (Chiang et al., 2024; Feuer et al., 2024; Ye et al., 2024). We integrate *Structured Output* using a predefined JSON Schema (natively supported by most models or via the *Instructor* library) and *Pydantic*, ensuring strict schema adherence. For example, a BST pre-order traversal must return a `list[int]`. Across 1,624 trials with nine models and six structures, we observed zero schema violations (Table 33 in Appendix), confirming robustness of Structured Output.

**Scoring system**   We compare model outputs to ground-truth answers, scoring 1 for correct and 0 for incorrect responses. Overall performance is reported as average accuracy across categories and difficulty levels from three runs. To ensure determinism, each task has a single well-defined solution. Any potential ambiguity (hash functions, tie-breaking rules) is explicitly specified in the prompts. We include an ablation with Levenshtein distance as a partial scoring metric in Section A.10.

## 4   EVALUATION

**Models**   We evaluate ten state-of-the-art LLMs across two categories: instruction-tuned models and reasoning models. We select the latest flagship models from each major provider, with each problem evaluated three times (166,230 total evaluations). **Instruction-tuned models** are widely deployed due to their efficiency and scalability, making their structural reasoning capabilities practically important: we cover GPT-4.1-2025-04-14 (OpenAI et al., 2024), Llama3.3-70B (Grattafiori et al., 2024), Gemini-2.0-Flash-001 (Team et al., 2024), Claude-3-5-Sonnet-20241022 (Anthropic, 2024), and DeepSeek-V3 (DeepSeek-AI et al., 2025). **Reasoning models** are explicitly trained for complex,

Table 1: Scores on `DSR-Bench-main` across ten LLMs. The table aggregates results by data structure and relationship type across three runs, including category scores and an overall score.

| Relationship | Data Structure | GPT-5 (med) | o4-mini | Gemini-2.5-Pro | Claude-3.7-Sonnet | DeepSeek-R1 | GPT-4.1 | Gemini-2.0-Flash | Claude-3.5-Sonnet | DeepSeek-V3 | Llama 3.3-70B |
|---|---|---|---|---|---|---|---|---|---|---|---|
| Linear | Array | 1.00 | 1.00 | 1.00 | 0.99 | 0.99 | 0.94 | 0.90 | 0.96 | 0.98 | 0.69 |
| | *Category avg.* | *1.00* | *1.00* | *1.00* | *0.99* | *0.99* | *0.94* | *0.90* | *0.96* | *0.98* | *0.69* |
| Temporal | Stack | 1.00 | 1.00 | 1.00 | 0.97 | 0.99 | 0.55 | 0.36 | 0.99 | 0.41 | 0.04 |
| | Queue | 1.00 | 1.00 | 1.00 | 1.00 | 0.98 | 0.55 | 0.36 | 0.99 | 0.43 | 0.25 |
| | LRU | 1.00 | 1.00 | 1.00 | 0.30 | 0.16 | 0.85 | 0.78 | 0.82 | 0.01 | 0.50 |
| | Priority Queue | 0.52 | 0.55 | 0.51 | 0.28 | 0.65 | 0.25 | 0.16 | 0.33 | 0.20 | 0.08 |
| | *Category avg.* | *0.88* | *0.89* | *0.84* | *0.64* | *0.69* | *0.55* | *0.42* | *0.79* | *0.26* | *0.22* |
| Associative | Hashmap | 0.87 | 0.51 | 0.28 | 0.63 | 0.33 | 0.06 | 0.10 | 0.16 | 0.00 | 0.00 |
| | Trie | 0.94 | 0.68 | 0.62 | 0.08 | 0.49 | 0.18 | 0.17 | 0.49 | 0.02 | 0.00 |
| | Suffix Tree | 0.98 | 0.73 | 0.90 | 0.91 | 0.96 | 0.00 | 0.01 | 0.08 | 0.67 | 0.00 |
| | Skip List | 0.68 | 0.62 | 0.78 | 0.75 | 0.69 | 0.07 | 0.06 | 0.42 | 0.02 | 0.01 |
| | *Category avg.* | *0.87* | *0.63* | *0.65* | *0.59* | *0.62* | *0.08* | *0.09* | *0.29* | *0.18* | *0.00* |
| Hierarchical | BST | 1.00 | 0.86 | 0.97 | 0.64 | 0.73 | 0.59 | 0.43 | 0.71 | 0.58 | 0.34 |
| | Heap | 0.61 | 0.68 | 0.38 | 0.40 | 0.48 | 0.20 | 0.10 | 0.27 | 0.15 | 0.07 |
| | RB tree | 0.76 | 0.65 | 0.49 | 0.30 | 0.62 | 0.12 | 0.12 | 0.46 | 0.09 | 0.05 |
| | B+ tree | 0.98 | 0.97 | 0.97 | 0.38 | 0.31 | 0.23 | 0.12 | 0.23 | 0.08 | 0.01 |
| | K-D Tree | 0.85 | 0.47 | 0.59 | 0.00 | 0.45 | 0.00 | 0.01 | 0.03 | 0.00 | 0.00 |
| | K-D Heap | 0.10 | 0.10 | 0.10 | 0.05 | 0.11 | 0.04 | 0.04 | 0.05 | 0.03 | 0.01 |
| | *Category avg.* | *0.72* | *0.64* | *0.58* | *0.33* | *0.45* | *0.20* | *0.14* | *0.34* | *0.16* | *0.08* |
| Network | Graph | 0.96 | 0.87 | 0.78 | 0.11 | 0.67 | 0.15 | 0.05 | 0.06 | 0.06 | 0.02 |
| | DSU | 1.00 | 0.98 | 0.97 | 0.99 | 1.00 | 0.02 | 0.00 | 0.02 | 0.82 | 0.00 |
| | Geom Graph | 0.16 | 0.16 | 0.13 | 0.02 | 0.13 | 0.07 | 0.02 | 0.02 | 0.01 | 0.02 |
| | *Category avg.* | *0.71* | *0.67* | *0.63* | *0.38* | *0.60* | *0.08* | *0.03* | *0.03* | *0.30* | *0.01* |
| Hybrid | Bloom Filter | 0.77 | 0.61 | 0.86 | 0.16 | 0.74 | 0.04 | 0.04 | 0.04 | 0.04 | 0.02 |
| | DAWG | 0.34 | 0.25 | 0.23 | 0.06 | 0.14 | 0.05 | 0.06 | 0.07 | 0.06 | 0.01 |
| | *Category avg.* | *0.56* | *0.43* | *0.55* | *0.11* | *0.44* | *0.05* | *0.05* | *0.06* | *0.05* | *0.02* |
| **Score** | *Overall avg.* | **0.79** | **0.72** | **0.71** | **0.51** | **0.63** | **0.31** | **0.27** | **0.41** | **0.32** | **0.17** |

multi-step reasoning: we evaluate GPT-5-2025-08-07 with medium thinking effort (OpenAI, 2025), o4-mini-2025-04-16 (OpenAI, 2024), Gemini-2.5-Pro (stable) (Team et al., 2024), Claude-3-7-Sonnet-20250219 (Anthropic, 2024), and DeepSeek-R1 (DeepSeek-AI et al., 2025).

**Prompting strategies**   We also study the impact of prompting strategies on data structure tasks. Unlike reasoning models with internal multi-step inference (e.g., reasoning tokens), instruction-tuned models are particularly sensitive to prompt formulation. We evaluate five prompting strategies: (i) **Stepwise**, which adds a "steps" field to the output JSON schema; (ii) **0-CoT**, which appends *"Let's think step by step"* without examples; (iii) **CoT**, which provides a single example with intermediate reasoning steps; (iv) **3-shot**, which includes three input-output examples; and (v) **None**, the default prompting setting with no added text. See Section A.3 for examples.

## 4.1   CAN LLMS UNDERSTAND AND MANIPULATE DATA STRUCTURES?

In this section, we present our main experimental results on `DSR-Bench`, including `main` and its challenging subset `challenge`. We discuss insights from five instruction-tuned models in Section 4.1.1 and from five reasoning models in Section 4.1.2.

### 4.1.1   INSTRUCTION-TUNED MODELS

**Instruction-tuned models struggle with multi-attribute reasoning.**   As shown in Table 1, these models show sharp accuracy drops in tasks involving elements with multiple attributes. For instance, while they perform well on QUEUE, their accuracy drops 30-50% on PRIORITY QUEUE, where each element includes a priority. Similarly, in the HASHMAP task, manual inspection of errors shows that models confuse keys and values, delete the wrong items, or hallucinate entries. These results reveal a key limitation for real-world deployment, where managing entities with multiple interacting properties, such as deadlines, priorities, or key-value records, is common.

**Multi-hop reasoning in hierarchical or network structures remains a key challenge.** We see from Table 1 that, while models perform reliably on BST, their accuracy drops by over 30% on RED-BLACK TREES, reflecting the difficulty of handling multi-hop properties such as maintaining balance across ancestral levels. Performance declines further on B+ TREES, which requires reasoning over wider spans of child pointers, and on GRAPH traversal tasks with many-to-many relationships. Manual inspection of reasoning traces reveals failures to retain earlier information: in GEOMETRIC GRAPH-*long*, all GPT-4.1 errors stemmed from dropping nodes during intermediate steps.

**Prompting can help, but only when carefully designed.** As shown in Table 2, the **None** prompt performs worst, suggesting that prompts encouraging stepwise reasoning can be beneficial. Our findings indicate two practical strategies: (i) Lightweight prompts such as **Stepwise** and **0-CoT** are easily implemented and consistently improve performance (Section A.5); (ii) Structured prompts like **CoT** and **3-shot** are most effective for uncommon data structures, but require careful design. A representative case is SUFFIX-TREE: across all models, zero-shot accuracy is below 0.40, but a well-designed CoT prompt doubles accuracy for three models (Section A.5). We include more analysis of CoT to provide meaningful takeaways for practitioners in Section A.6.

Table 2: Average scores across all tasks for instruction-tuned models under different prompting strategies.

| Model | Stepwise | 0-CoT | CoT | 3-shot | None |
|---|---|---|---|---|---|
| DeepSeek-V3 | 0.67 | 0.67 | 0.66 | 0.64 | 0.55 |
| Llama-3.3 | 0.46 | 0.46 | 0.48 | 0.46 | 0.34 |
| GPT-4.1 | 0.82 | 0.83 | 0.94 | 0.80 | 0.59 |
| Gemini-2.0-Flash | 0.57 | 0.86 | 0.59 | 0.87 | 0.58 |
| Claude-3.5-Sonnet | 0.72 | 0.92 | 0.72 | 0.93 | 0.69 |

### 4.1.2 REASONING MODELS

Table 3: Scores on the `challenge` suite for five reasoning models.

| Model | Score | Priority Queue | Suffix Tree | Trie | Skip List | Heap | Red-black tree | B+ Tree | K-D Tree | K-D Heap | DSU | Geom Graph | Bloom Filter | DAWG |
|---|---|---|---|---|---|---|---|---|---|---|---|---|---|---|
| GPT-5 | 0.50 | 0.52 | 0.98 | 0.92 | 0.51 | 0.71 | 0.585 | 0.98 | 0.85 | 0.10 | 1.00 | 0.19 | 0.47 | 0.00 |
| Gemini-2.5-Pro | 0.47 | 0.24 | 0.89 | 0.59 | 0.61 | 0.62 | 0.60 | 0.96 | 0.67 | 0.00 | 0.99 | 0.00 | 0.69 | 0.00 |
| DeepSeek-R1 | 0.36 | 0.48 | 0.90 | 0.05 | 0.54 | 0.27 | 0.37 | 0.21 | 0.01 | 0.01 | 0.99 | 0.01 | 0.31 | 0.00 |
| o4-mini | 0.34 | 0.30 | 0.37 | 0.32 | 0.41 | 0.71 | 0.37 | 0.94 | 0.38 | 0.00 | 0.94 | 0.00 | 0.07 | 0.06 |
| Claude-3.7-Sonnet | 0.21 | 0.04 | 0.79 | 0.00 | 0.61 | 0.13 | 0.12 | 0.13 | 0.00 | 0.00 | 0.98 | 0.00 | 0.00 | 0.00 |

**Reasoning models remain brittle on complex and spatial data structures.** From Table 1, we see reasoning models outperform instruction-tuned models in general, especially on hierarchical and networked structures. However, the overall score remains below 0.5 on `challenge` in Table 3, in particular for *long* tasks and complex data structures. For example, the highest score on SKIP LIST is only 0.61, despite its prevalence in introductory-level textbooks and its wide use in dictionaries and maps. Notably, accuracy on K-D TREE, K-D HEAP, and GEOMETRIC GRAPHS is low even for *short* tasks, suggesting that high-dimensional spatial reasoning remains a significant challenge (Section A.4). To further probe these limitations, we introduce `spatial`, described in Section 4.2.

**Implicit priors may hinder instruction following.** In an ablation on K-D HEAP in Table 5, switching the tie-breaking rule from lexicographic order to Euclidean norm causes o4-mini's score to drop by over 0.40, as it continues to apply lexicographic comparisons. Directly querying o4-mini about the default implementation of a K-D heap confirms this point: it assumes lexicographic keys. These results suggest that reasoning models such as o4-mini may struggle to override entrenched priors learned from training, limiting their reliability on tasks with user-defined constraints.

### 4.2 CAN LLMS REASON STRUCTURALLY ON SPATIAL DATA?

Real-world data is often represented in high-dimensional feature spaces. To assess whether LLMs can reason over such spatial data, we extend the benchmark with the `spatial` suite, which includes three multi-dimensional variants: K-D HEAP, K-D TREE, and GEOMETRIC GRAPH embedded in Euclidean space. These structures are common in practice; for instance, K-D trees are key data structures in computer vision and graphics. Given the complexity of these tasks, we use GPT-4.1 with the **Stepwise** prompt to encourage intermediate reasoning steps.

Table 4: Scores for the three spatial data structures with input data of varying dimensionality ($k = 1, 2, 3, 5$).

| $k$ | K-D Heap | | K-D Tree | | Geom. Graph | |
|---|---|---|---|---|---|---|
| | GPT-4.1 | o4-mini | GPT-4.1 | o4-mini | GPT-4.1 | o4-mini |
| 1 | 0.74 | 0.82 | 0.91 | 0.82 | 0.18 | 0.88 |
| 2 | 0.30 | 0.34 | 0.86 | 0.69 | 0.04 | 0.90 |
| 3 | 0.26 | 0.26 | 0.92 | 0.68 | 0.01 | 0.76 |
| 5 | 0.21 | 0.21 | 0.73 | 0.64 | 0.00 | 0.71 |

Table 5: Performance on the two comparison metrics for K-D HEAP with different $k$ values.

| $k$ | Lexicographic | Euclidean |
|---|---|---|
| 1 | 0.79 | 0.82 |
| 2 | 0.87 | 0.34 |
| 3 | 0.84 | 0.26 |
| 5 | 0.90 | 0.21 |

**Performance degrades as input dimensionality increases.** As shown in Table 4, accuracy declines for both models as dimensionality increases. Higher-dimensional data challenges models with more complex computation over distance metrics and partitions, limiting their effectiveness in spatial tasks. For instance, K-D trees are widely used to expedite nearest neighbor queries over 128-dimensional SIFT descriptors in computer vision (Silpa-Anan & Hartley, 2008). Interestingly, 2D outperforms 1D in GEOMETRIC GRAPH, likely due to its more common presence in training data from textbooks.

**Limited robustness to non-uniform data distributions.** We assess LLM robustness to distribution shifts by comparing performance on uniformly sampled versus skewed or clustered data. We test K-D tree construction tasks using three non-uniform distributions from scikit-learn (Pedregosa et al., 2011): circles, moons, and blobs (illustration in Figure 3, Section A.7). As shown in Table 6, GPT-4.1's performance drops sharply on non-uniform inputs, possibly due to a higher likelihood of uniformly distributed examples in the training data. Since task difficulty is held constant, this gap suggests a reliance on pattern memorization rather than true reasoning. In contrast, o4-mini shows a

Table 6: Scores on K-D TREE with varying input data distributions.

| Distribution | GPT-4.1 | o4-mini |
|---|---|---|
| Uniform | 0.86 | 0.69 |
| Moon | 0.42 | 0.62 |
| Blob | 0.33 | 0.62 |
| Circle | 0.31 | 0.67 |

smaller drop, indicating that reasoning models may generalize better to distribution shifts. A more in-depth inspection of errors and discussion on root causes can be found in Section A.7.

### 4.3 CAN LLMs REASON STRUCTURALLY ON NATURAL LANGUAGE TASKS?

While the previous sections evaluated LLMs on canonical data structures, real-world use cases are often expressed in natural language. To bridge this gap, we extend the benchmark with the `natural` suite, which embeds data structure tasks in narrative, real-world contexts, allowing us to test whether LLMs generalize structural reasoning beyond formal descriptions.

> **Example natural language prompt for QUEUE compound.**
>
> On a sunny afternoon in the park, an ice cream truck rolled in... Children began to form a line, each newcomer taking their place at the end while the vendor served from the front...
>
> - Isabella Miller ran over and joined the ice cream line.
> - The next kid in line was served promptly.
> - ...
>
> **Q**: What is the order of the remaining kids in line?

Table 7: Model performance on formal and natural descriptors.

| Model | Task | Formal | Natural |
|---|---|---|---|
| GPT-4.1 | Queue | 1.00 | 0.77 |
| | BST | 0.88 | 0.59 |
| | Graph | 0.42 | 0.43 |
| o4-mini | Queue | 1.00 | 0.83 |
| | BST | 1.00 | 0.93 |
| | Graph | 0.84 | 0.67 |

We design three real-world scenarios that implicitly require data structures: QUEUE (children buying ice cream), BINARY SEARCH TREE (clinic appointments), and GRAPH (galaxy traveling game). Synthetic data follows the same distributions as Section 3.3, with realistic substitutions (e.g., names for integers). Each scenario was written by humans and paraphrased by GPT-4o. All prompts were reviewed by three annotators for clarity and unambiguity. Details are supplemented in Section A.8.

**LLMs struggle when shifting from formal to natural language.** As shown in Table 7, performance drops when tasks are described in natural language compared to formal descriptions, despite

identical problem distribution. The higher accuracy on formal descriptors may stem from training on textbook-style patterns, where integers and explicit syntax are common. This observation suggests that even reasoning models struggle to apply reasoning in language-rich, real-world contexts. Bridging this gap is crucial for reliable deployment and presents a key direction for future research.

### 4.4 CAN LLMS REASON STRUCTURALLY WITH CODE GENERATION?

As motivated in Section 1 and Section 2, our benchmark targets *general reasoning* independent of code execution or tool use. Nonetheless, to assess whether code generation provides any benefit, we run ablations on six models using the `code` suite across three modes: (i) *CodeOnly*, where models generate Python code executed by an external interpreter; (ii) *CodeEnforce*, where models must write code and reason through its execution internally without relying on an interpreter; and (iii) *CodeMaybe*, similar to *CodeEnforce* but makes code generation optional. Full details on the experimental setup and results are provided in Section A.9.

Table 8: Average performance on seven data structures across three code generation modes and the default setting.

| Mode | GPT-4.1 | o4-mini | Gemini-2.0-Flash | Gemini-2.5-Pro | Claude-3.5-Sonnet | Claude-3.7-Sonnet |
|---|---|---|---|---|---|---|
| Default | 0.40 | 0.73 | 0.38 | 0.55 | 0.44 | 0.55 |
| CodeMaybe | 0.38 | 0.76 | 0.41 | 0.55 | 0.43 | 0.57 |
| CodeEnforce | 0.38 | 0.75 | 0.41 | 0.53 | 0.42 | 0.55 |
| CodeOnly | 0.95 | 0.82 | 0.44 | 0.57 | 0.74 | 0.87 |

Table 9: Individual scores on three data structures under *CodeOnly*.

| | o4-mini | Gemini-2.5-pro | Claude-3.7-Sonnet |
|---|---|---|---|
| Geom Graph | 0.99 | 0.98 | 0.96 |
| DAWG | 0.56 | 0.90 | 0.89 |
| Graph-Natural | 0.69 | 0.56 | 0.23 |

**Models cannot reason over generated code.** In Table 8, performance in *CodeMaybe* and *CodeEnforce* matches the default setup, showing that writing code offers little benefit over natural language reasoning when models must internally simulate execution. This reinforces our central claim: LLMs remain limited in their ability to perform structural reasoning, even when guided by their own code.

**Code helps only with standard tasks and fails on natural language ones.** As shown in Table 9, with an external interpreter in *CodeOnly*, models perform well on GEOM GRAPH, a standard structure in computer graphics with widely available implementations. In contrast, they struggle on the less familiar DAWG, where custom constraints enforce unambiguous outputs, suggesting reliance on memorized solutions rather than genuine reasoning. Performance drops further on GRAPH-NATURAL, where models default to brittle pattern matching (e.g., rigidly mapping "A space tunnel links A and B" to "G.add_edge(A, B)") but failing to understand paraphrases like "Couriers frequently travel the tunnel connecting A to B." These results highlight the fragility of structural reasoning under natural language ambiguity, even with code generation and external execution.

## 5 DISCUSSION AND CONCLUSION

Can LLMs reason structurally? Through `DSR-Bench`, we provide a systematic answer: not yet. Instruction-tuned models struggle with multi-attribute reasoning (e.g., database indexing) and multi-hop reasoning (e.g., trip planning), while reasoning models achieve only 0.498 accuracy on complex structures and can ignore user-defined constraints. These limitations highlight the need for architectures that support precise function computation, memory mechanisms, and the flexibility to adapt to personalized requests. Evaluations on high-dimensional data (`spatial`) and natural language scenarios (`natural`) further reveal gaps between current reasoning capabilities and real-world readiness. Code generation modes (`code`) show that models cannot reliably reason over their own code, often reverting to memorized patterns or brittle mappings even with external execution.

`DSR-Bench` provides a systematic framework for evaluating algorithmic reasoning through the lens of structural reasoning. It provides the community with a powerful diagnostic tool: researchers can pinpoint failure modes, test targeted improvements, and measure progress on specific relationship types, paving the way for algorithm-centric model design as an alternative to scaling alone. It also raises new questions: Can LLMs dynamically choose reasoning strategies? Where and why do errors

arise in intermediate steps? Do multimodal LLMs have visual understanding over structures, when DSR-Bench tasks are presented in images? Furthermore, DSR-Bench could serve as a testbed for mechanistic interpretability (e.g., Wang et al., 2024a) as well as RL fine-tuning due to its fine-grained hierarchical organization and the composition, easy-to-verify nature of data structure tasks. We invite the community to use `DSR-Bench` to explore these directions.

## ETHICS STATEMENT

We have carefully read and adhered to the ICLR Code of Ethics in conducting this work.

## REPRODUCIBILITY STATEMENT

We release all code, datasets, and prompts for data generation and model evaluation at https://anonymous.4open.science/r/DSR-Bench-C40D , ensuring full reproducibility.

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

# A APPENDIX

## TABLE OF CONTENTS

### A.1 ADDITIONAL RELATED WORKS

**LLM Benchmarking**     Large language models (LLMs) have demonstrated remarkable performance across a wide range of applications, prompting growing interest in understanding their capabilities and limitations. Recent efforts have focused on systematically benchmarking LLMs on core natural language processing tasks, including language understanding (Wang et al., 2024b; Feng et al., 2024), text generation (Maynez et al., 2023; Singh et al., 2024), reasoning (Feng et al., 2024; Valmeekam et al., 2023), and machine translation (Yao et al., 2023). Additionally, some benchmarks evaluate ethical dimensions such as robustness, bias, and trustworthiness (Siska et al., 2024; Luo et al., 2024; Koo et al., 2023; Li et al., 2024b; Zhang et al., 2024b; Liu et al., 2023c). Specialized benchmarks have also emerged in scientific and technical domains, including mathematics (White et al., 2025; Liu et al., 2024a), programming (White et al., 2025; Jain et al., 2025; Chen et al., 2021; Zheng et al., 2023), data analysis (Sui et al., 2024; Liu et al., 2023b; White et al., 2025; Tang et al., 2023; Jiang et al., 2023), and medicine (Li et al., 2024a; Liu et al., 2023a; Longwell et al., 2024; Xu et al., 2024b; Kanithi et al., 2024). While several studies examine how LLMs convert unstructured inputs into tabular or relational formats (Tang et al., 2023; Jiang et al., 2023), our work explores a distinct question: How well can LLMs construct, manipulate, and reason about classic *data structures* such as stacks, trees, and graphs? To our knowledge, this is the first comprehensive benchmark targeting this capability, offering a conceptually different evaluation from prior work on structured data.

**Algorithmic reasoning with LLMs**     (Zhou et al., 2022; 2024; Lee et al., 2024) study arithmetic tasks on small transformers (addition, subtraction, sine, square root), which target a different aspect of algorithmic reasoning. (Wang et al., 2023a; Fatemi et al., 2024) are graph benchmarks for LLMs (e.g., connectivity, cycle detection); our graph tasks (BFS, DFS) do not appear in their task sets and are integral parts of our hierarchical design, serving as preliminaries for harder tasks (e.g., DAWG). The CLRS-Text Benchmark (Markeeva et al., 2024) covers 30 classical algorithms (e.g., sorting, greedy algorithms), with only BFS and DFS overlapping with ours. In contrast, DSR-Bench focuses on data structures as fundamental building blocks for these algorithms, enabling finer-grained diagnosis and offering more interpretable and actionable insights into reasoning failures. Our prompt design, evaluation pipeline, and specialized suites are also new and different.

### A.2 DETAILS OF DATA STRUCTURES AND OPERATIONS

In this section, we list the data structures and the corresponding operations tested in Table 10. We then provide detailed descriptions of each data structure and explain how we specify their implementations to eliminate ambiguity.

**Array**     An array contains a list of elements stored in contiguous memory. We test its access, deletion, reversal, and search operations. To remove ambiguity, we specify that the array is 0-indexed. For operations like deletion, if duplicates exist, we delete the first occurrence. The final state is a list of elements in the array.

**Stack**     A stack is a linear data structure that follows a Last-In-First-Out (LIFO) order. We test compound operations consisting of random sequences of push and pop from the top of the stack. The final state is a list of remaining elements in the stack.

**Queue**     A queue is a linear data structure that follows a First-In-First-Out (FIFO) order. We evaluate compound operations of enqueue to the back and dequeue from the front. The final state is a list of remaining elements in the queue.

**LRU Cache**     An LRU (Least Recently Used) cache stores a fixed number of items and evicts the least recently accessed one when full. We evaluate this caching operation with a sequence of requests as input. The final state is a set of elements in the LRU cache.

**Priority Queue**     A priority queue stores elements with integer priorities, allowing access to the highest-priority element. We test compound operations including insert, remove, raise key, and decrease key, using a Fibonacci heap. Ties are broken by insertion order. The final state is a level-

Table 10: Summary of data structures and associated operations in `DSR-Bench`. Data structures marked with * are included in the `challenge` suite. All compound operations without explicit specification consist of (insert, delete).

| Category | Data Structure | Description | Operation | Application |
|---|---|---|---|---|
| Linear | Array | Contiguous memory | Access, Delete, Insert, Reverse, Search | Data storage |
| Temporal | Stack | LIFO (Last-In, First-Out) | Compound (Push, Pop) | Syntax parsing |
| | Queue | FIFO (First-In, First-Out) | Compound | OS management |
| | LRU Cache | Least-recently-used | Cache (Evict, Add) | Web browsers |
| | Priority Queue* | Priority ordering | Compound | Job scheduling |
| Associative | Hashmap | Key-value storage | Compound | Large-scale storage |
| | Trie* | Hierarchical mapping of strings | Compound | Autocomplete |
| | Suffix Tree* | Text indexing via suffixes | Compound | DNA pattern matching |
| | Skip List* | Probabilistic layers for fast search | Compound | Concurrent databases |
| Hierarchical | Binary Search Tree | Hierarchical storage | Pre/In/Post-Order Traversal, Insert, Remove, Compound | Computer networks |
| | Heap* | Complete binary tree with priority ordering | Heapify, Compound | Memory management |
| | Red-Black Tree* | Self-balanced tree | Construct, Compound | Database indexes |
| | B+ Tree* | Multi-way balanced tree | Compound | File systems |
| | K-D Tree* | Hierarchical, spatial partition | Construct, Compound | 3D graphics |
| | K-D Heap* | Hierarchical, complete binary tree, high-dimensional priority | Compound | GPU job scheduling |
| Network | Graph | Many-to-many relationships | Breadth-First Traversal, Depth-First Traversal | Social networks |
| | Disjoint Set Union* | Sets partition & union | Compound (Union, Find) | Physics simulation |
| | Geometric Graph* | Graph modeling spatial data | Construct | Public transportation |
| Hybrid | Bloom Filter* | Probabilistic set and hashmap | Compound | Spam detection |
| | Directed Acyclic Word Graph* | Graph and trie tree | Compound | Compilers |

order traversal of the Fibonacci heap forest, outputting (value,priority) pairs, with nodes at each level sorted by descending priority and ties broken by larger value first.

**Hashmap**  A hashmap is a key-value structure supporting fast access, insertion, and deletion via hashed keys. We test compound insert and delete operations, specifying the hash function and using chaining for collision resolution. The final state is a list of key-value pairs per bucket, preserving insertion order within each chain.

**Trie**  A trie is a tree-based data structure for storing strings, where each node represents a character and paths from the root to leaves represent complete words, where common prefixes are shared. When generating strings, we increase the likelihood of shared prefixes to ensure the resulting trie has meaningful structure. The final state is a pre-order traversal of the trie, where each node's children are visited in lexicographical order to ensure an unambiguous representation.

**Suffix Tree**  A suffix tree is a compressed trie built from all suffixes of a string, where each edge can represent multiple characters and each path from the root corresponds to a substring. We test the

construction of a suffix tree from a given word, appending a terminal character "$" to ensure a unique structure. The final state is a pre-order traversal collecting edge labels, with child edges visited in lexicographical order and "$" taking priority.

**Skip List**   A skip list is a probabilistic data structure composed of multiple layers of linked lists, where higher layers allow "skipping" over elements for faster access. We test compound operations of insert and delete. Insertion begins at the bottom layer, with the element randomly promoted to higher levels; pointers are updated at each level to preserve the structure. To remove ambiguity, promotion probabilities are explicitly specified in the prompts. The final state is represented as a list of lists, each corresponding to a layer of the skip list.

**Binary Search Tree (BST)**   A binary search tree is a hierarchical structure where each node has at most two children: the left holds smaller values, and the right holds larger ones. We test insert, remove, tree traversals (pre-order, in-order, post-order), depth computation, and compound insert-remove operations. Inputs are guaranteed to contain no duplicates to ensure unique outputs. The final state for traversal tasks is a list of elements in the specified order, while for insert, remove, and compound tasks, it includes both pre-order and post-order traversals.

**Heap**   A heap is a complete binary tree that satisfies the min-heap property, where each parent node is less than or equal to its children. We test both heapify and compound insert-delete operations using an array-based heap. Comparisons follow min-heap ordering, with ties broken by preferring the left child. The final state is the array representation of the heap.

**Red-Black (RB) Tree**   A red-black tree is a self-balancing binary search tree where each node is colored red or black and must satisfy specific balance rules: no two consecutive red nodes are allowed, and all root-to-leaf paths must have the same number of black nodes. We test both construction and compound (insert, delete) operations. The final state is a pre-order traversal of the nodes, represented as tuples (value, color).

**B+ Tree**   A B+ tree is a multi-way search tree used in databases and filesystems, where values are stored in leaf nodes and internal nodes serve as routing indexes. Leaf nodes are linked for efficient range queries. We specify splitting and merging rules to ensure unambiguous, balanced updates during compound insert and delete operations. The final state is a pre-order traversal of nodes, with keys in each node sorted in ascending order.

**K-D Tree**   A K-D (k-dimensional) tree recursively partitions space by alternating the splitting axis at each level. Each node represents a point and divides the space into two halves based on a chosen coordinate. It is commonly used for spatial indexing, range queries, and nearest neighbor search. We test the construction of K-D trees across different dimensionalities, specifying the axis splitting sequence and tie-breaking rules (e.g., median selection for even-sized splits) to ensure consistency. The final state is a pre-order traversal of the tree.

**K-D Heap**   A K-D heap maintains heap order based on a $k$-dimensional priority with a comparison metric, enabling efficient access to extremal points in multidimensional datasets. We test compound operations of insert and delete across different dimensionalities. We specify an array-based heap implementation, with comparisons based on Euclidean distance and tie-breaking rules that prefer the left child in case of a tie. The final state is a list of vectors representing the contents of the min-heap.

**Graph**   A graph is a collection of nodes connected by edges, which can be directed or undirected, and is used to model networks, dependencies, and paths. We define graphs using edge list statements and test both breadth-first and depth-first traversals from a given source node, visiting neighbors in ascending order. Node values are unique to ensure consistent outputs. The final state is the list of nodes visited during the traversal.

**Disjoint Set Union (DSU)**   A disjoint set union maintains a partition of elements into disjoint subsets, supporting efficient merges and membership queries. Internally, it forms a forest where each node points to a representative root. We test two operations: a sequence of unions between subsets, followed by queries for each element's representative. To ensure consistency, we specify that

lower-rank roots are always attached to higher-rank ones. The final state lists the representative root of each input element in its original order.

**Geometric (Geom) Graph**  Geometric graphs are graphs with nodes embedded in geometric space, typically Euclidean, where edges are formed based on spatial relationships such as proximity. They are widely used in robotics, computer graphics, and sensor networks where spatial structure is essential. We compute the Euclidean distance between each pair of points and add an edge if the distance is below a given threshold, assigning the edge a weight equal to that distance. The final state is a breadth-first traversal from a specified source node, exploring all neighbors at each level before proceeding. We specify the order of search based on the edge weights.

**Bloom Filter**  A (counting) Bloom filter is a compact, probabilistic data structure for set membership testing, guaranteeing no false negatives and allowing a tunable false positive rate. It uses multiple hash functions to map each element to several positions in a counter array, incrementing or decrementing counts. We test on compound operations of insert and delete. We specify the hash functions used in the prompt to avoid ambiguity. The final state is the array of counters representing the Bloom filter.

**Directed Acyclic Word Graph (DAWG)**  A Directed Acyclic Word Graph (DAWG) is a compressed data structure for storing a set of words, sharing both prefixes and suffixes. Nodes indicate whether they mark the end of a word, and edges are labeled with characters. Unlike a trie, a DAWG merges equivalent subtrees to reduce redundancy, making it well-suited for large static dictionaries and lexicon lookups. We test compound operations of insert and delete, specifying that merging should occur at the final step along with the merging rules. To ensure a meaningful structure, we increase the likelihood of generating words with shared prefixes. The final state is a breadth-first traversal from the root (an empty string), where each node is recorded by the prefix it represents and whether it marks the end of a word.

### A.3    EXAMPLES OF PROMPTING STRATEGIES

In this section, we illustrate prompting methods using compound operations of QUEUE as an example.

**Stepwise**  This method explicitly adds a `steps` attribute in the JSON schema of Structured Output, guiding the model to produce operations in a sequential and interpretable manner. Below is an example schema for an array task:

```
class Step(BaseModel):
    explanation: str
    output: str
class ArraySchema(BaseModel):
    steps: list[Step]
    final_answer: int
```

---

**_Stepwise_ prompting on compound operations of QUEUE.**

A queue is a data structure in which items are added at one end and removed from the other, maintaining a first-in, first-out (FIFO) order. You should create a queue. There are two types of operations: 1. (enqueue, k) means an element k is appended to the queue as the last element. 2. (dequeue) means the first element of the queue is deleted. You are given an empty queue initially.

**Q**: What is the final queue, when performing the following operations:

• (enqueue, 49)

• (dequeue)

• (enqueue, 86)

• (enqueue, 52)

Answer the question in 8000 tokens.

---

**0-CoT**   This method appends the phrase "Let's think step by step" to the prompt to encourage reasoning without providing exemplars.

---

**0-CoT prompting on compound operations of QUEUE.**

A queue is a data structure in which items are added at one end and removed from the other, maintaining a first-in, first-out (FIFO) order. You should create a queue. There are two types of operations: 1. (enqueue, k) means an element k is appended to the queue as the last element. 2. (dequeue) means the first element of the queue is deleted.

You are given an empty queue initially.

**Q**: What is the final queue, when performing the following operations:

- (enqueue, 49)
- (dequeue)
- (enqueue, 86)
- (enqueue, 52)

Let's think step by step. Answer the question in 8000 tokens.

---

**CoT**   This strategy provides a single example that includes both intermediate reasoning steps and the final answer.

---

**CoT prompting on compound operations of QUEUE.**

A queue is a data structure in which items are added at one end and removed from the other, maintaining a first-in, first-out (FIFO) order. You should create a queue. There are two types of operations: 1. (enqueue, k) means an element k is appended to the queue as the last element. 2. (dequeue) means the first element of the queue is deleted. You are given an empty queue initially.
Q: What is the final queue, when performing the following operations:
- (enqueue, 21)
- (enqueue, 3)
- (dequeue)
- (dequeue)
- (enqueue, 48)
A: Initially, the queue is []. After (enqueue, 21), it becomes [21]. After (enqueue, 3), it becomes [21, 3]. After (dequeue), it becomes [3]. After (dequeue), it becomes []. After (enqueue, 48), it becomes [48]. The final queue is [48].
**Q**: What is the final queue, when performing the following operations:
- (enqueue, 49)
- (dequeue)
- (enqueue, 86)
- (enqueue, 52)
Answer the question in 8000 tokens.

---

**3-shot**   This strategy provides three input-output examples to guide the model through pattern matching and demonstration.

---

**_3-shot_ prompting on compound operations of QUEUE.**

A queue is a data structure in which items are added at one end and removed from the other, maintaining a first-in, first-out (FIFO) order. You should create a queue. There are two types of operations: 1. (enqueue, k) means an element k is appended to the queue as the last element. 2. (dequeue) means the first element of the queue is deleted. You are given an empty queue initially.

Q: What is the final queue, when performing the following operations:

- (enqueue, 21)

- (enqueue, 3)

- (dequeue)

- (dequeue)

- (enqueue, 48)

A: The final queue is [48]. (... Example 2...) (... Example 3...)

**Q**: What is the final queue, when performing the following operations:

- (enqueue, 49)

- (dequeue)

- (enqueue, 86)

- (enqueue, 52)

Answer the question in 8000 tokens.

---

**None**  This method adds the instruction "No additional text needed" to prompt concise, direct answers that fit within the token limit and conform to the structured output format.

---

**none prompting on compound operations of QUEUE.**

A queue is a data structure in which items are added at one end and removed from the other, maintaining a first-in, first-out (FIFO) order. You should create a queue. There are two types of operations: 1. (enqueue, k) means an element k is appended to the queue as the last element. 2. (dequeue) means the first element of the queue is deleted. You are given an empty queue initially.

**Q**: What is the final queue, when performing the following operations:

- (enqueue, 49)

- (dequeue)

- (enqueue, 86)

- (enqueue, 52)

No additional text needed. Answer the question in 8000 tokens.

---

### A.4 ACCURACY BY TASK AND LENGTH LEVEL ACROSS ALL MODELS

In this section, we provide supplementary accuracy tables for all models in `DSR-Bench`, broken down by task and length level. Table 11 summarizes the accuracy of instruction-tuned models on a subset of basic data structures across different length levels. Table 12 presents the accuracy of reasoning models on selected data structures from the `DSR-Bench-challenge` suite. For detailed per-model results across all tasks and length levels, see Table 13 (GPT-5), Table 14 (o4-mini), Table 15 (Gemini-2.5-Pro), Table 16 (Claude-3.7-Sonnet), Table 17 (DeepSeek-R1), Table 18 (GPT-4.1), Table 19 (Gemini-2.0-Flash), Table 20 (Claude-3.5-Sonnet), Table 21 (DeepSeek-V3), and Table 22 (Llama-3.3).

Table 11: Average accuracy on basic data structure tasks for instruction-tuned models (3 runs, scaled to [0, 1], rounded to two decimals).

| Category | DS | Length | GPT-4.1 | Gemini-2.0-Flash | Claude-3.5-Sonnet | DeepSeek-V3 | Llama-3.3 |
|---|---|---|---|---|---|---|---|
| Linear | Array | Short | 0.98 | 0.98 | 1.00 | 1.00 | 0.89 |
| | | Medium | 0.95 | 0.92 | 0.96 | 0.97 | 0.70 |
| | | Long | 0.88 | 0.88 | 0.91 | 0.96 | 0.48 |
| Temporal | Queue | Short | 0.97 | 0.67 | 1.00 | 0.84 | 0.58 |
| | | Medium | 0.49 | 0.37 | 1.00 | 0.38 | 0.12 |
| | | Long | 0.18 | 0.03 | 0.98 | 0.07 | 0.06 |
| | Stack | Short | 0.97 | 0.67 | 1.00 | 0.70 | 0.09 |
| | | Medium | 0.49 | 0.37 | 1.00 | 0.49 | 0.04 |
| | | Long | 0.18 | 0.03 | 0.98 | 0.04 | 0.00 |
| Associative | Hashmap | Short | 0.19 | 0.28 | 0.37 | 0.00 | 0.00 |
| | | Medium | 0.00 | 0.01 | 0.10 | 0.00 | 0.00 |
| | | Long | 0.00 | 0.00 | 0.00 | 0.00 | 0.00 |
| Hierarchical | BST | Short | 0.82 | 0.63 | 0.89 | 0.76 | 0.46 |
| | | Medium | 0.56 | 0.37 | 0.66 | 0.55 | 0.31 |
| | | Long | 0.39 | 0.29 | 0.57 | 0.43 | 0.26 |
| Network | Graph | Short | 0.41 | 0.15 | 0.15 | 0.16 | 0.06 |
| | | Medium | 0.05 | 0.02 | 0.02 | 0.02 | 0.01 |
| | | Long | 0.00 | 0.00 | 0.00 | 0.00 | 0.00 |

Table 12: Average accuracy on the `DSR-Bench-challenge` suite for reasoning models (3 runs, scaled to [0, 1], rounded to two decimals).

| Category | Data Structure | Length | o4-mini | Gemini-2.5 Pro | Claude-3.7 Sonnet | Deepseek R1 | GPT-5 (med) |
|---|---|---|---|---|---|---|---|
| Temporal | Priority Queue | Short | 0.89 | 0.89 | 0.70 | 0.92 | 0.84 |
| | | Medium | 0.47 | 0.41 | 0.11 | 0.54 | 0.44 |
| | | Long | 0.30 | 0.23 | 0.04 | 0.48 | 0.28 |
| Associative | Trie | Short | 0.99 | 0.93 | 0.23 | 0.93 | 0.97 |
| | | Medium | 0.73 | 0.56 | 0.00 | 0.50 | 0.94 |
| | | Long | 0.32 | 0.37 | 0.00 | 0.05 | 0.92 |
| | Suffix Tree | Short | 0.96 | 0.96 | 0.96 | 1.00 | 1.00 |
| | | Medium | 0.87 | 0.89 | 0.98 | 0.98 | 1.00 |
| | | Long | 0.37 | 0.86 | 0.79 | 0.90 | 0.95 |
| | Skip List | Short | 0.84 | 0.94 | 0.87 | 0.89 | 0.88 |
| | | Medium | 0.60 | 0.87 | 0.76 | 0.63 | 0.66 |
| | | Long | 0.41 | 0.53 | 0.61 | 0.54 | 0.51 |
| Hierarchical | Heap | Short | 0.61 | 0.57 | 0.70 | 0.63 | 0.78 |
| | | Medium | 0.72 | 0.28 | 0.37 | 0.53 | 0.71 |
| | | Long | 0.71 | 0.29 | 0.13 | 0.27 | 0.71 |
| | Red Black Tree | Short | 0.92 | 0.91 | 0.69 | 0.86 | 0.92 |
| | | Medium | 0.67 | 0.47 | 0.11 | 0.63 | 0.79 |
| | | Long | 0.37 | 0.08 | 0.12 | 0.37 | 0.59 |
| | B+ Tree | Short | 0.99 | 1.00 | 0.70 | 0.49 | 0.97 |
| | | Medium | 0.98 | 0.97 | 0.32 | 0.23 | 0.99 |
| | | Long | 0.94 | 0.94 | 0.13 | 0.21 | 0.98 |
| | K-D Heap | Short | 0.22 | 0.23 | 0.11 | 0.23 | 0.23 |
| | | Medium | 0.09 | 0.06 | 0.03 | 0.08 | 0.08 |
| | | Long | 0.00 | 0.00 | 0.00 | 0.01 | 0.00 |
| | K-D Tree | Short | 0.59 | 0.96 | 0.00 | 1.00 | 0.97 |
| | | Medium | 0.43 | 0.64 | 0.00 | 0.34 | 0.90 |
| | | Long | 0.38 | 0.16 | 0.00 | 0.01 | 0.67 |
| Network | DSU | Short | 1.00 | 1.00 | 1.00 | 1.00 | 1.00 |
| | | Medium | 1.00 | 0.99 | 1.00 | 1.00 | 1.00 |
| | | Long | 0.94 | 0.90 | 0.98 | 0.99 | 1.00 |
| Hybrid | Bloom Filter | Short | 1.00 | 0.94 | 0.44 | 0.99 | 1.00 |
| | | Medium | 0.77 | 0.97 | 0.03 | 0.92 | 0.84 |
| | | Long | 0.07 | 0.66 | 0.00 | 0.31 | 0.47 |
| | DAWG | Short | 0.49 | 0.61 | 0.17 | 0.40 | 1.00 |
| | | Medium | 0.20 | 0.08 | 0.00 | 0.02 | 0.03 |
| | | Long | 0.06 | 0.01 | 0.00 | 0.00 | 0.00 |
| | Geom Graph | Short | 0.37 | 0.19 | 0.07 | 0.36 | 0.21 |
| | | Medium | 0.10 | 0.17 | 0.00 | 0.01 | 0.08 |
| | | Long | 0.00 | 0.02 | 0.00 | 0.01 | 0.19 |

### A.4.1 PERFORMANCE OF GPT-5

Table 13: Mean (± std) accuracy of **GPT-5** on all `DSR-Bench` tasks over three runs. Data structures marked with * are included in the `DSR-Bench-challenge` suite.

| Category | Data Structure | Operation | Short | Medium | Long |
|---|---|---|---|---|---|
| Linear | Array | Access | 1.00 (0.00) | 1.00 (0.00) | 1.00 (0.00) |
| | | Delete | 1.00 (0.00) | 1.00 (0.00) | 1.00 (0.00) |
| | | Insert | 1.00 (0.00) | 1.00 (0.00) | 1.00 (0.00) |
| | | Reverse | 1.00 (0.00) | 1.00 (0.00) | 1.00 (0.00) |
| | | Search | 1.00 (0.00) | 1.00 (0.00) | 1.00 (0.00) |
| Temporal | Stack | Compound | 1.00 (0.00) | 1.00 (0.00) | 1.00 (0.00) |
| | Queue | Compound | 1.00 (0.00) | 1.00 (0.00) | 1.00 (0.00) |
| | LRU Cache | Cache | 1.00 (0.00) | 1.00 (0.00) | 1.00 (0.00) |
| | Priority Queue* | Compound | 0.84 (0.02) | 0.44 (0.02) | 0.28 (0.07) |
| Associative | Hashmap | Compound | 1.00 (0.00) | 0.89 (0.02) | 0.71 (0.10) |
| | Trie* | Compound | 0.97 (0.03) | 0.94 (0.05) | 0.92 (0.07) |
| | Suffix Tree* | Construct | 1.00 (0.00) | 1.00 (0.00) | 0.95 (0.02) |
| | Skip List* | Compound | 0.88 (0.05) | 0.66 (0.08) | 0.51 (0.05) |
| Hierarchical | BST | Insert | 1.00 (0.00) | 1.00 (0.00) | 1.00 (0.00) |
| | | Remove | 0.98 (0.02) | 1.00 (0.00) | 0.98 (0.02) |
| | | In-Order Traversal | 1.00 (0.00) | 1.00 (0.00) | 1.00 (0.00) |
| | | Pre-Order Traversal | 0.98 (0.04) | 1.00 (0.00) | 1.00 (0.00) |
| | | Post-Order Traversal | 1.00 (0.00) | 1.00 (0.00) | 1.00 (0.00) |
| | | Depth | 1.00 (0.00) | 1.00 (0.00) | 1.00 (0.00) |
| | | Compound | 1.00 (0.00) | 1.00 (0.00) | 0.98 (0.02) |
| | Heap* | Compound | 0.51 (0.02) | 0.81 (0.08) | 0.66 (0.12) |
| | | Heapify | 0.33 (0.00) | 0.61 (0.05) | 0.76 (0.10) |
| | RB Tree* | Construct | 0.93 (0.00) | 0.90 (0.03) | 0.68 (0.13) |
| | | Compound | 0.91 (0.04) | 0.67 (0.03) | 0.49 (0.04) |
| | $B^+$ Tree* | Compound | 0.97 (0.00) | 0.99 (0.02) | 0.98 (0.02) |
| | K-D Tree* | Construct | 0.97 (0.03) | 0.90 (0.00) | 0.67 (0.06) |
| | K-D Heap* | Compound | 0.23 (0.00) | 0.08 (0.02) | 0.00 (0.00) |
| Network | Graph | Breadth-First Traversal | 0.92 (0.02) | 0.98 (0.02) | 0.94 (0.02) |
| | | Depth-First Traversal | 0.93 (0.03) | 1.00 (0.00) | 0.96 (0.02) |
| | DSU* | Compound | 1.00 (0.00) | 1.00 (0.00) | 1.00 (0.00) |
| | Geom Graph* | Construct | 0.21 (0.10) | 0.08 (0.04) | 0.19 (0.13) |
| Hybrid | Bloom Filter* | Construct | 1.00 (0.00) | 0.84 (0.07) | 0.47 (0.00) |
| | DAWG* | Compound | 1.00 (0.00) | 0.03 (0.03) | 0.00 (0.00) |

### A.4.2 PERFORMANCE OF O4-MINI

Table 14: Mean (± std) accuracy of **o4-mini** on all `DSR-Bench` tasks over three runs. Data structures marked with * are included in the `DSR-Bench-challenge` suite.

| Category | Data Structure | Operation | Short | Medium | Long |
|---|---|---|---|---|---|
| Linear | Array | Access | 1.00 (0.00) | 1.00 (0.00) | 1.00 (0.00) |
| | | Delete | 1.00 (0.00) | 1.00 (0.00) | 1.00 (0.00) |
| | | Insert | 1.00 (0.00) | 1.00 (0.00) | 1.00 (0.00) |
| | | Reverse | 1.00 (0.00) | 1.00 (0.00) | 1.00 (0.00) |
| | | Search | 1.00 (0.00) | 1.00 (0.00) | 1.00 (0.00) |
| Temporal | Stack | Compound | 1.00 (0.00) | 1.00 (0.00) | 1.00 (0.00) |
| | Queue | Compound | 1.00 (0.00) | 1.00 (0.00) | 1.00 (0.00) |
| | LRU Cache | Cache | 1.00 (0.00) | 1.00 (0.00) | 1.00 (0.00) |
| | Priority Queue* | Compound | 0.89 (0.02) | 0.47 (0.06) | 0.30 (0.03) |
| Associative | Hashmap | Compound | 0.89 (0.04) | 0.37 (0.00) | 0.26 (0.08) |
| | Trie* | Compound | 0.99 (0.02) | 0.73 (0.06) | 0.32 (0.05) |
| | Suffix Tree* | Construct | 0.96 (0.02) | 0.87 (0.03) | 0.37 (0.07) |
| | Skip List* | Compound | 0.84 (0.02) | 0.60 (0.03) | 0.41 (0.02) |
| Hierarchical | BST | Insert | 1.00 (0.00) | 1.00 (0.00) | 0.99 (0.02) |
| | | Remove | 1.00 (0.00) | 1.00 (0.00) | 0.97 (0.03) |
| | | In-Order Traversal | 1.00 (0.00) | 1.00 (0.00) | 1.00 (0.00) |
| | | Pre-Order Traversal | 1.00 (0.00) | 1.00 (0.00) | 1.00 (0.00) |
| | | Post-Order Traversal | 1.00 (0.00) | 1.00 (0.00) | 1.00 (0.00) |
| | | Depth | 1.00 (0.00) | 0.99 (0.02) | 1.00 (0.00) |
| | | Compound | 1.00 (0.00) | 1.00 (0.00) | 0.98 (0.02) |
| | Heap* | Compound | 0.77 (0.06) | 0.86 (0.07) | 0.74 (0.05) |
| | | Heapify | 0.44 (0.05) | 0.58 (0.04) | 0.67 (0.03) |
| | RB Tree* | Construct | 0.90 (0.03) | 0.32 (0.13) | 0.05 (0.02) |
| | | Compound | 0.97 (0.00) | 0.64 (0.04) | 0.26 (0.02) |
| | $B^+$ Tree* | Compound | 0.99 (0.02) | 0.98 (0.04) | 0.94 (0.02) |
| | K-D Tree* | Construct | 0.59 (0.07) | 0.43 (0.06) | 0.38 (0.10) |
| | K-D Heap* | Compound | 0.22 (0.02) | 0.09 (0.02) | 0.00 (0.00) |
| Network | Graph | Breadth-First Traversal | 0.99 (0.02) | 0.97 (0.03) | 0.72 (0.14) |
| | | Depth-First Traversal | 1.00 (0.00) | 0.88 (0.02) | 0.64 (0.10) |
| | DSU* | Compound | 1.00 (0.00) | 1.00 (0.00) | 0.94 (0.04) |
| | Geom Graph* | Construct | 0.83 (0.07) | 0.13 (0.00) | 0.01 (0.02) |
| Hybrid | Bloom Filter* | Compound | 1.00 (0.00) | 0.77 (0.09) | 0.07 (0.00) |
| | DAWG* | Compound | 0.49 (0.10) | 0.20 (0.09) | 0.06 (0.05) |

### A.4.3 PERFORMANCE OF GEMINI-2.5-PRO

Table 15: Mean (± std) accuracy of **Gemini-2.5-Pro** on all `DSR-Bench` tasks over three runs. Data structures marked with * are included in the `DSR-Bench-challenge` suite.

| Category | Data Structure | Operation | Short | Medium | Long |
|---|---|---|---|---|---|
| Linear | Array | Access | 1.00 (0.00) | 1.00 (0.00) | 1.00 (0.00) |
| | | Delete | 1.00 (0.00) | 1.00 (0.00) | 1.00 (0.00) |
| | | Insert | 1.00 (0.00) | 1.00 (0.00) | 1.00 (0.00) |
| | | Reverse | 1.00 (0.00) | 0.98 (0.02) | 0.99 (0.02) |
| | | Search | 1.00 (0.00) | 1.00 (0.00) | 1.00 (0.00) |
| Temporal | Stack | Compound | 1.00 (0.00) | 1.00 (0.00) | 1.00 (0.00) |
| | Queue | Compound | 1.00 (0.00) | 1.00 (0.00) | 1.00 (0.00) |
| | LRU Cache | Cache | 1.00 (0.00) | 1.00 (0.00) | 1.00 (0.00) |
| | Priority Queue* | Compound | 0.89 (0.02) | 0.41 (0.02) | 0.23 (0.05) |
| Associative | Hashmap | Compound | 0.58 (0.02) | 0.16 (0.07) | 0.11 (0.04) |
| | Trie* | Compound | 0.93 (0.03) | 0.56 (0.02) | 0.37 (0.03) |
| | Suffix Tree* | Construct | 0.96 (0.04) | 0.89 (0.02) | 0.86 (0.08) |
| | Skip List* | Compound | 0.94 (0.02) | 0.87 (0.06) | 0.53 (0.03) |
| Hierarchical | BST | Insert | 1.00 (0.00) | 0.94 (0.02) | 0.94 (0.05) |
| | | Remove | 0.87 (0.06) | 0.86 (0.07) | 0.89 (0.02) |
| | | In-Order Traversal | 1.00 (0.00) | 1.00 (0.00) | 1.00 (0.00) |
| | | Pre-Order Traversal | 1.00 (0.00) | 1.00 (0.00) | 1.00 (0.00) |
| | | Post-Order Traversal | 1.00 (0.00) | 0.99 (0.02) | 0.94 (0.04) |
| | | Depth | 1.00 (0.00) | 1.00 (0.00) | 1.00 (0.00) |
| | | Compound | 0.99 (0.02) | 0.99 (0.02) | 0.98 (0.02) |
| | Heap* | Compound | 0.78 (0.04) | 0.49 (0.10) | 0.53 (0.12) |
| | | Heapify | 0.36 (0.02) | 0.06 (0.05) | 0.04 (0.04) |
| | RB Tree* | Construct | 0.91 (0.07) | 0.53 (0.09) | 0.03 (0.03) |
| | | Compound | 0.91 (0.02) | 0.41 (0.07) | 0.13 (0.03) |
| | $B^+$ Tree* | Compound | 1.00 (0.00) | 0.97 (0.03) | 0.94 (0.06) |
| | K-D Tree* | Construct | 0.96 (0.02) | 0.64 (0.13) | 0.16 (0.02) |
| | K-D Heap* | Compound | 0.23 (0.00) | 0.06 (0.02) | 0.00 (0.00) |
| Network | Graph | Breadth-First Traversal | 1.00 (0.00) | 1.00 (0.00) | 0.74 (0.02) |
| | | Depth-First Traversal | 1.00 (0.00) | 0.81 (0.05) | 0.14 (0.02) |
| | DSU* | Compound | 1.00 (0.00) | 0.99 (0.02) | 0.90 (0.05) |
| | Geom Graph* | Construct | 0.19 (0.02) | 0.17 (0.06) | 0.02 (0.02) |
| Hybrid | Bloom Filter* | Construct | 0.94 (0.02) | 0.97 (0.00) | 0.66 (0.02) |
| | DAWG* | Compound | 0.61 (0.05) | 0.08 (0.07) | 0.01 (0.02) |

A.4.4 PERFORMANCE OF CLAUDE-3.7-SONNET

Table 16: Mean (± std) accuracy of **Claude-3.7-Sonnet** on all `DSR-Bench` tasks over three runs. Data structures marked with * are included in the `DSR-Bench-challenge` suite.

| Category | Data Structure | Operation | Short | Medium | Long |
|---|---|---|---|---|---|
| Linear | Array | Access | 1.00 (0.00) | 1.00 (0.00) | 1.00 (0.00) |
| | | Delete | 1.00 (0.00) | 1.00 (0.00) | 1.00 (0.00) |
| | | Insert | 1.00 (0.00) | 1.00 (0.00) | 0.96 (0.02) |
| | | Reverse | 1.00 (0.00) | 0.98 (0.02) | 0.97 (0.00) |
| | | Search | 1.00 (0.00) | 1.00 (0.00) | 1.00 (0.00) |
| Temporal | Stack | Compound | 1.00 (0.00) | 1.00 (0.00) | 1.00 (0.00) |
| | Queue | Compound | 1.00 (0.00) | 0.93 (0.00) | 0.98 (0.02) |
| | LRU Cache | Compound | 1.00 (0.00) | 1.00 (0.00) | 0.98 (0.02) |
| | Priority Queue* | Compound | 0.70 (0.06) | 0.11 (0.02) | 0.04 (0.02) |
| Associative | Hashmap | Compound | 0.71 (0.02) | 0.16 (0.05) | 0.04 (0.05) |
| | Trie* | Compound | 0.94 (0.02) | 0.64 (0.07) | 0.31 (0.10) |
| | Suffix Tree* | Construct | 0.23 (0.00) | 0.00 (0.00) | 0.00 (0.00) |
| | Skip List* | Compound | 0.87 (0.06) | 0.76 (0.05) | 0.61 (0.11) |
| Hierarchical | BST | Insert | 0.96 (0.04) | 0.98 (0.04) | 0.79 (0.02) |
| | | Remove | 0.94 (0.02) | 0.91 (0.02) | 0.92 (0.02) |
| | | In-Order Traversal | 1.00 (0.00) | 1.00 (0.00) | 1.00 (0.00) |
| | | Pre-Order Traversal | 1.00 (0.00) | 0.99 (0.02) | 1.00 (0.00) |
| | | Post-Order Traversal | 1.00 (0.00) | 0.74 (0.02) | 0.93 (0.03) |
| | | Depth | 1.00 (0.00) | 1.00 (0.00) | 0.98 (0.02) |
| | | Compound | 0.90 (0.03) | 0.21 (0.02) | 0.24 (0.05) |
| | Heap* | Compound | 0.70 (0.03) | 0.32 (0.05) | 0.13 (0.03) |
| | | Heapify | 0.89 (0.02) | 0.62 (0.08) | 0.26 (0.02) |
| | RB Tree* | Construct | 0.19 (0.05) | 0.00 (0.00) | 0.00 (0.00) |
| | | Compound | 0.57 (0.03) | 0.03 (0.00) | 0.00 (0.00) |
| | $B^+$ Tree* | Compound | 0.80 (0.00) | 0.18 (0.02) | 0.23 (0.03) |
| | K-D Tree* | Construct | 0.00 (0.00) | 0.00 (0.00) | 0.00 (0.00) |
| | K-D Heap* | Compound | 0.11 (0.02) | 0.03 (0.00) | 0.00 (0.00) |
| Network | Graph | Breadth-First Traversal | 0.40 (0.03) | 0.08 (0.02) | 0.01 (0.02) |
| | | Depth-First Traversal | 0.50 (0.03) | 0.11 (0.02) | 0.00 (0.00) |
| | DSU* | Compound | 0.04 (0.02) | 0.12 (0.04) | 0.00 (0.00) |
| | Geom Graph* | Construct | 0.04 (0.05) | 0.00 (0.00) | 0.00 (0.00) |
| Hybrid | Bloom Filter* | Compound | 0.44 (0.04) | 0.03 (0.00) | 0.00 (0.00) |
| | DAWG* | Compound | 0.17 (0.00) | 0.00 (0.00) | 0.00 (0.00) |

### A.4.5 PERFORMANCE OF DEEPSEEK-R1

Table 17: Mean (± std) accuracy of **DeepSeek-R1** on all `DSR-Bench` tasks over three runs. Data structures marked with * are included in the `DSR-Bench-challenge` suite.

| Category | Data Structure | Operation | Short | Medium | Long |
|---|---|---|---|---|---|
| Linear | Array | Access | 1.00 (0.00) | 0.98 (0.02) | 1.00 (0.00) |
| | | Delete | 0.99 (0.02) | 1.00 (0.00) | 0.98 (0.02) |
| | | Insert | 0.98 (0.02) | 0.99 (0.02) | 0.99 (0.02) |
| | | Reverse | 1.00 (0.00) | 1.00 (0.00) | 1.00 (0.00) |
| | | Search | 1.00 (0.00) | 1.00 (0.00) | 1.00 (0.00) |
| Temporal | Stack | Compound | 1.00 (0.00) | 1.00 (0.00) | 0.94 (0.07) |
| | Queue | Compound | 1.00 (0.00) | 1.00 (0.00) | 0.97 (0.00) |
| | LRU Cache | Compound | 1.00 (0.00) | 1.00 (0.00) | 0.99 (0.01) |
| | Priority Queue* | Compound | 0.92 (0.02) | 0.54 (0.07) | 0.48 (0.05) |
| Associative | Hashmap | Compound | 0.44 (0.05) | 0.01 (0.02) | 0.03 (0.03) |
| | Trie* | Compound | 0.54 (0.24) | 0.32 (0.04) | 0.12 (0.06) |
| | Suffix Tree* | Construct | 0.93 (0.07) | 0.50 (0.07) | 0.05 (0.05) |
| | Skip List* | Compound | 0.89 (0.04) | 0.63 (0.03) | 0.54 (0.02) |
| Hierarchical | BST | Insert | 1.00 (0.00) | 0.98 (0.02) | 0.90 (0.03) |
| | | Remove | 0.98 (0.02) | 0.93 (0.03) | 0.88 (0.05) |
| | | In-Order Traversal | 1.00 (0.00) | 0.99 (0.02) | 1.00 (0.00) |
| | | Pre-Order Traversal | 1.00 (0.00) | 1.00 (0.00) | 1.00 (0.00) |
| | | Post-Order Traversal | 1.00 (0.00) | 1.00 (0.00) | 0.97 (0.00) |
| | | Depth | 1.00 (0.00) | 1.00 (0.00) | 1.00 (0.00) |
| | | Compound | 0.97 (0.03) | 0.84 (0.08) | 0.65 (0.07) |
| | Heap* | Compound | 0.49 (0.04) | 0.23 (0.07) | 0.21 (0.05) |
| | | Heapify | 0.34 (0.02) | 0.16 (0.08) | 0.08 (0.06) |
| | RB Tree* | Construct | 0.88 (0.02) | 0.10 (0.06) | 0.00 (0.00) |
| | | Compound | 0.91 (0.04) | 0.37 (0.10) | 0.03 (0.03) |
| | $B^+$ Tree* | Compound | 0.81 (0.02) | 0.88 (0.04) | 0.70 (0.06) |
| | K-D Tree* | Construct | 1.00 (0.00) | 0.34 (0.05) | 0.01 (0.02) |
| | K-D Heap* | Compound | 0.23 (0.00) | 0.08 (0.02) | 0.01 (0.02) |
| Network | Graph | Breadth-First Traversal | 0.92 (0.02) | 0.90 (0.06) | 0.46 (0.05) |
| | | Depth-First Traversal | 0.80 (0.09) | 0.58 (0.04) | 0.22 (0.02) |
| | DSU* | Compound | 0.64 (0.56) | 0.92 (0.04) | 0.83 (0.07) |
| | Geom Graph* | Construct | 0.99 (0.02) | 0.00 (0.00) | 0.00 (0.00) |
| Hybrid | Bloom Filter* | Compound | 0.99 (0.02) | 0.92 (0.02) | 0.31 (0.02) |
| | DAWG* | Compound | 0.40 (0.12) | 0.02 (0.02) | 0.00 (0.00) |

### A.4.6 PERFORMANCE OF GPT-4.1

Table 18: Mean (± std) accuracy of **GPT-4.1** on all `DSR-Bench` tasks over three runs. Data structures marked with * are included in the `DSR-Bench-challenge` suite.

| Category | Data Structure | Operation | Short | Medium | Long |
|---|---|---|---|---|---|
| Linear | Array | Access | 1.00 (0.00) | 1.00 (0.00) | 1.00 (0.00) |
| | | Delete | 1.00 (0.00) | 1.00 (0.00) | 1.00 (0.00) |
| | | Insert | 0.91 (0.08) | 0.79 (0.02) | 0.54 (0.02) |
| | | Reverse | 0.98 (0.02) | 0.97 (0.00) | 0.86 (0.02) |
| | | Search | 1.00 (0.00) | 1.00 (0.00) | 1.00 (0.00) |
| Temporal | Stack | Compound | 0.97 (0.00) | 0.49 (0.02) | 0.18 (0.04) |
| | Queue | Compound | 0.82 (0.04) | 0.59 (0.04) | 0.19 (0.07) |
| | LRU Cache | Cache | 0.94 (0.02) | 0.80 (0.00) | 0.81 (0.02) |
| | Priority Queue* | Compound | 0.63 (0.03) | 0.10 (0.00) | 0.03 (0.00) |
| Associative | Hashmap | Compound | 0.19 (0.07) | 0.00 (0.00) | 0.00 (0.00) |
| | Trie* | Compound | 0.39 (0.07) | 0.13 (0.03) | 0.01 (0.02) |
| | Suffix Tree* | Construct | 0.00 (0.00) | 0.00 (0.00) | 0.00 (0.00) |
| | Skip List* | Compound | 0.21 (0.02) | 0.00 (0.00) | 0.00 (0.00) |
| Hierarchical | BST | Insert | 0.79 (0.04) | 0.50 (0.03) | 0.14 (0.02) |
| | | Remove | 0.78 (0.04) | 0.58 (0.02) | 0.36 (0.04) |
| | | In-Order Traversal | 1.00 (0.00) | 1.00 (0.00) | 0.94 (0.02) |
| | | Pre-Order Traversal | 1.00 (0.00) | 0.97 (0.00) | 0.98 (0.02) |
| | | Post-Order Traversal | 0.82 (0.02) | 0.51 (0.04) | 0.23 (0.06) |
| | | Depth | 0.30 (0.04) | 0.07 (0.06) | 0.03 (0.03) |
| | | Compound | 0.69 (0.02) | 0.26 (0.02) | 0.03 (0.00) |
| | Heap* | Compound | 0.58 (0.02) | 0.01 (0.02) | 0.00 (0.00) |
| | | Heapify | 0.57 (0.03) | 0.04 (0.02) | 0.00 (0.00) |
| | RB Tree* | Construct | 0.12 (0.02) | 0.00 (0.00) | 0.00 (0.00) |
| | | Compound | 0.31 (0.04) | 0.02 (0.02) | 0.00 (0.00) |
| | B$^+$ Tree* | Compound | 0.27 (0.00) | 0.30 (0.00) | 0.13 (0.00) |
| | K-D Tree* | Construct | 0.00 (0.00) | 0.00 (0.00) | 0.00 (0.00) |
| | K-D Heap* | Compound | 0.10 (0.00) | 0.03 (0.00) | 0.00 (0.00) |
| Network | Graph | Breadth-First Traversal | 0.31 (0.05) | 0.09 (0.02) | 0.00 (0.00) |
| | | Depth-First Traversal | 0.50 (0.03) | 0.00 (0.00) | 0.00 (0.00) |
| | DSU* | Compound | 0.06 (0.02) | 0.00 (0.00) | 0.00 (0.00) |
| | Geom Graph* | Construct | 0.03 (0.00) | 0.00 (0.00) | 0.00 (0.00) |
| Hybrid | Bloom Filter* | Compound | 0.10 (0.00) | 0.03 (0.00) | 0.00 (0.00) |
| | DAWG* | Compound | 0.16 (0.02) | 0.00 (0.00) | 0.00 (0.00) |

### A.4.7 PERFORMANCE OF GEMINI-2.0-FLASH

Table 19: Mean (± std) accuracy of **Gemini-2.0-Flash** on all `DSR-Bench` tasks over three runs. Data structures marked with * are included in the `DSR-Bench-challenge` suite.

| Category | Data Structure | Operation | Short | Medium | Long |
|---|---|---|---|---|---|
| Linear | Array | Access | 1.00 (0.00) | 1.00 (0.00) | 1.00 (0.00) |
| | | Delete | 0.96 (0.02) | 0.87 (0.03) | 0.77 (0.03) |
| | | Insert | 0.99 (0.02) | 0.96 (0.02) | 1.00 (0.00) |
| | | Reverse | 0.96 (0.02) | 0.78 (0.02) | 0.64 (0.04) |
| | | Search | 1.00 (0.00) | 1.00 (0.00) | 0.97 (0.00) |
| Temporal | Stack | Compound | 0.67 (0.00) | 0.37 (0.00) | 0.03 (0.00) |
| | Queue | Compound | 0.87 (0.00) | 0.33 (0.00) | 0.10 (0.00) |
| | LRU Cache | Cache | 0.93 (0.00) | 0.86 (0.02) | 0.56 (0.02) |
| | Priority Queue* | Compound | 0.38 (0.02) | 0.10 (0.00) | 0.01 (0.02) |
| Associative | Hashmap | Compound | 0.28 (0.05) | 0.01 (0.02) | 0.00 (0.00) |
| | Trie* | Compound | 0.31 (0.02) | 0.18 (0.02) | 0.03 (0.00) |
| | Suffix Tree* | Construct | 0.00 (0.00) | 0.02 (0.02) | 0.00 (0.00) |
| | Skip List* | Compound | 0.16 (0.02) | 0.00 (0.00) | 0.03 (0.00) |
| Hierarchical | BST | Insert | 0.31 (0.02) | 0.27 (0.03) | 0.06 (0.04) |
| | | Remove | 0.63 (0.09) | 0.33 (0.03) | 0.13 (0.03) |
| | | In-Order Traversal | 0.87 (0.00) | 0.66 (0.02) | 0.71 (0.04) |
| | | Pre-Order Traversal | 1.00 (0.00) | 1.00 (0.00) | 0.93 (0.00) |
| | | Post-Order Traversal | 0.63 (0.00) | 0.17 (0.00) | 0.10 (0.00) |
| | | Depth | 0.13 (0.09) | 0.03 (0.00) | 0.00 (0.00) |
| | | Compound | 0.51 (0.05) | 0.12 (0.04) | 0.10 (0.00) |
| | Heap* | Compound | 0.32 (0.05) | 0.03 (0.00) | 0.02 (0.02) |
| | | Heapify | 0.23 (0.06) | 0.00 (0.00) | 0.00 (0.00) |
| | RB Tree* | Construct | 0.08 (0.02) | 0.00 (0.00) | 0.00 (0.00) |
| | | Compound | 0.43 (0.03) | 0.07 (0.00) | 0.00 (0.00) |
| | $B^+$ Tree* | Compound | 0.17 (0.00) | 0.13 (0.03) | 0.06 (0.05) |
| | K-D Tree* | Construct | 0.02 (0.02) | 0.00 (0.00) | 0.00 (0.00) |
| | K-D Heap* | Compound | 0.10 (0.00) | 0.03 (0.00) | 0.00 (0.00) |
| Network | Graph | Breadth-First Traversal | 0.10 (0.00) | 0.03 (0.00) | 0.00 (0.00) |
| | | Depth-First Traversal | 0.19 (0.02) | 0.00 (0.00) | 0.00 (0.00) |
| | DSU* | Compound | 0.01 (0.02) | 0.00 (0.00) | 0.00 (0.00) |
| | Geom Graph* | Construct | 0.07 (0.03) | 0.00 (0.00) | 0.00 (0.00) |
| Hybrid | Bloom Filter* | Compound | 0.10 (0.00) | 0.03 (0.00) | 0.00 (0.00) |
| | DAWG* | Compound | 0.18 (0.02) | 0.00 (0.00) | 0.00 (0.00) |

### A.4.8 Performance of Claude-3.5-Sonnet

Table 20: Mean (± std) accuracy of **Claude-3.5-Sonnet** on all `DSR-Bench` tasks over three runs. Data structures marked with * are included in the `DSR-Bench-challenge` suite.

| Category | Data Structure | Operation | Short | Medium | Long |
|---|---|---|---|---|---|
| Linear | Array | Access | 1.00 (0.00) | 1.00 (0.00) | 1.00 (0.00) |
| | | Delete | 1.00 (0.00) | 1.00 (0.00) | 0.93 (0.00) |
| | | Insert | 1.00 (0.00) | 0.90 (0.00) | 0.90 (0.00) |
| | | Reverse | 1.00 (0.00) | 0.88 (0.02) | 0.72 (0.05) |
| | | Search | 1.00 (0.00) | 1.00 (0.00) | 1.00 (0.00) |
| Temporal | Stack | Compound | 1.00 (0.00) | 1.00 (0.00) | 0.98 (0.04) |
| | Queue | Compound | 0.87 (0.00) | 0.67 (0.03) | 0.79 (0.02) |
| | LRU Cache | Cache | 0.99 (0.02) | 0.90 (0.07) | 0.58 (0.13) |
| | Priority Queue* | Compound | 0.63 (0.00) | 0.27 (0.03) | 0.09 (0.02) |
| Associative | Hashmap | Compound | 0.37 (0.03) | 0.10 (0.00) | 0.00 (0.00) |
| | Trie* | Compound | 0.89 (0.04) | 0.50 (0.03) | 0.07 (0.00) |
| | Suffix Tree* | Construct | 0.21 (0.02) | 0.03 (0.00) | 0.00 (0.00) |
| | Skip List* | Compound | 0.77 (0.06) | 0.30 (0.07) | 0.20 (0.07) |
| Hierarchical | BST | Insert | 0.80 (0.06) | 0.50 (0.06) | 0.51 (0.05) |
| | | Remove | 0.96 (0.04) | 0.87 (0.00) | 0.77 (0.00) |
| | | In-Order Traversal | 0.97 (0.03) | 0.94 (0.02) | 0.94 (0.02) |
| | | Pre-Order Traversal | 1.00 (0.00) | 1.00 (0.00) | 0.99 (0.02) |
| | | Post-Order Traversal | 1.00 (0.00) | 0.69 (0.08) | 0.54 (0.02) |
| | | Depth | 1.00 (0.00) | 0.96 (0.02) | 0.78 (0.02) |
| | | Compound | 0.77 (0.07) | 0.28 (0.05) | 0.09 (0.02) |
| | Heap* | Compound | 0.78 (0.04) | 0.13 (0.00) | 0.11 (0.02) |
| | | Heapify | 0.53 (0.12) | 0.08 (0.04) | 0.00 (0.00) |
| | RB Tree* | Construct | 0.13 (0.00) | 0.00 (0.00) | 0.00 (0.00) |
| | | Compound | 0.44 (0.02) | 0.03 (0.00) | 0.00 (0.00) |
| | $B^+$ Tree* | Compound | 0.40 (0.00) | 0.28 (0.08) | 0.02 (0.02) |
| | K-D Tree* | Construct | 0.09 (0.02) | 0.00 (0.00) | 0.00 (0.00) |
| | K-D Heap* | Compound | 0.13 (0.00) | 0.02 (0.02) | 0.00 (0.00) |
| Network | Graph | Breadth-First Traversal | 0.17 (0.03) | 0.02 (0.02) | 0.00 (0.00) |
| | | Depth-First Traversal | 0.13 (0.03) | 0.02 (0.02) | 0.00 (0.00) |
| | DSU* | Compound | 0.07 (0.03) | 0.00 (0.00) | 0.00 (0.00) |
| | Geom Graph* | Construct | 0.10 (0.00) | 0.00 (0.00) | 0.00 (0.00) |
| Hybrid | Bloom Filter* | Compound | 0.10 (0.00) | 0.03 (0.00) | 0.00 (0.00) |
| | DAWG* | Compound | 0.20 (0.00) | 0.00 (0.00) | 0.00 (0.00) |

### A.4.9 PERFORMANCE OF DEEPSEEK-V3

Table 21: Mean (± std) accuracy of **DeepSeek-V3** on all `DSR-Bench` tasks over three runs. Data structures marked with * are included in the `DSR-Bench-challenge` suite.

| Category | Data Structure | Operation | Short | Medium | Long |
|---|---|---|---|---|---|
| Linear | Array | Access | 1.00 (0.00) | 0.97 (0.00) | 0.97 (0.00) |
| | | Delete | 1.00 (0.00) | 1.00 (0.00) | 1.00 (0.00) |
| | | Insert | 1.00 (0.00) | 1.00 (0.00) | 1.00 (0.00) |
| | | Reverse | 1.00 (0.00) | 0.92 (0.02) | 0.92 (0.02) |
| | | Search | 1.00 (0.00) | 0.97 (0.00) | 0.93 (0.00) |
| Temporal | Stack | Compound | 0.70 (0.03) | 0.49 (0.02) | 0.04 (0.02) |
| | Queue | Compound | 0.84 (0.02) | 0.38 (0.02) | 0.07 (0.03) |
| | LRU Cache | Compound | 0.94 (0.02) | 0.77 (0.06) | 0.76 (0.02) |
| | Priority Queue* | Compound | 0.53 (0.03) | 0.06 (0.04) | 0.00 (0.00) |
| Associative | Hashmap | Compound | 0.04 (0.02) | 0.00 (0.00) | 0.00 (0.00) |
| | Trie* | Compound | 0.00 (0.00) | 0.00 (0.00) | 0.00 (0.00) |
| | Suffix Tree* | Construct | 0.06 (0.02) | 0.00 (0.00) | 0.00 (0.00) |
| | Skip List* | Compound | 0.06 (0.02) | 0.00 (0.00) | 0.00 (0.00) |
| Hierarchical | BST | Insert | 0.93 (0.03) | 0.62 (0.02) | 0.46 (0.05) |
| | | Remove | 0.84 (0.04) | 0.80 (0.03) | 0.66 (0.02) |
| | | In-Order Traversal | 0.97 (0.00) | 1.00 (0.00) | 1.00 (0.00) |
| | | Pre-Order Traversal | 1.00 (0.00) | 1.00 (0.00) | 1.00 (0.00) |
| | | Post-Order Traversal | 0.82 (0.02) | 0.53 (0.03) | 0.20 (0.03) |
| | | Depth | 0.67 (0.03) | 0.24 (0.02) | 0.07 (0.03) |
| | | Compound | 0.68 (0.02) | 0.12 (0.04) | 0.06 (0.02) |
| | Heap* | Compound | 0.23 (0.00) | 0.00 (0.00) | 0.00 (0.00) |
| | | Heapify | 0.59 (0.02) | 0.06 (0.02) | 0.00 (0.00) |
| | RB Tree* | Construct | 0.09 (0.02) | 0.00 (0.00) | 0.00 (0.00) |
| | | Compound | 0.30 (0.03) | 0.00 (0.00) | 0.00 (0.00) |
| | $B^+$ Tree* | Compound | 0.14 (0.05) | 0.10 (0.03) | 0.00 (0.00) |
| | K-D Tree* | Construct | 0.00 (0.00) | 0.00 (0.00) | 0.00 (0.00) |
| | K-D Heap* | Compound | 0.07 (0.00) | 0.03 (0.00) | 0.00 (0.00) |
| Network | Graph | Breadth-First Traversal | 0.29 (0.05) | 0.04 (0.02) | 0.00 (0.00) |
| | | Depth-First Traversal | 0.22 (0.02) | 0.03 (0.00) | 0.00 (0.00) |
| | DSU* | Compound | 0.03 (0.00) | 0.00 (0.00) | 0.00 (0.00) |
| | Geom Graph* | Construct | 0.06 (0.02) | 0.00 (0.00) | 0.00 (0.00) |
| Hybrid | Bloom Filter* | Compound | 0.10 (0.00) | 0.03 (0.00) | 0.00 (0.00) |
| | DAWG* | Compound | 0.17 (0.00) | 0.00 (0.00) | 0.00 (0.00) |

### A.4.10  PERFORMANCE OF LLAMA-3.3

Table 22: Mean (± std) accuracy of **Llama-3.3** on all `DSR-Bench` tasks over three runs. Data structures marked with * are included in the `DSR-Bench-challenge` suite.

| Category | Data Structure | Operation | Short | Medium | Long |
|---|---|---|---|---|---|
| Linear | Array | Access | 1.00 (0.00) | 0.56 (0.04) | 0.38 (0.02) |
| | | Delete | 0.81 (0.08) | 0.68 (0.04) | 0.44 (0.05) |
| | | Insert | 0.76 (0.04) | 0.78 (0.02) | 0.29 (0.07) |
| | | Reverse | 0.91 (0.02) | 0.56 (0.02) | 0.34 (0.07) |
| | | Search | 0.97 (0.00) | 0.90 (0.00) | 0.93 (0.00) |
| Temporal | Stack | Compound | 0.09 (0.02) | 0.04 (0.05) | 0.00 (0.00) |
| | Queue | Compound | 0.58 (0.11) | 0.12 (0.02) | 0.06 (0.02) |
| | LRU Cache | Compound | 0.74 (0.08) | 0.44 (0.11) | 0.31 (0.11) |
| | Priority Queue* | Compound | 0.21 (0.05) | 0.01 (0.02) | 0.02 (0.02) |
| Associative | Hashmap | Compound | 0.00 (0.00) | 0.00 (0.00) | 0.00 (0.00) |
| | Trie* | Compound | 0.01 (0.02) | 0.00 (0.00) | 0.00 (0.00) |
| | Suffix Tree* | Construct | 0.00 (0.00) | 0.00 (0.00) | 0.00 (0.00) |
| | Skip List* | Compound | 0.03 (0.00) | 0.00 (0.00) | 0.00 (0.00) |
| Hierarchical | BST | Insert | 0.37 (0.00) | 0.14 (0.02) | 0.03 (0.00) |
| | | Remove | 0.49 (0.04) | 0.30 (0.03) | 0.14 (0.04) |
| | | In-Order Traversal | 0.60 (0.06) | 0.61 (0.08) | 0.61 (0.04) |
| | | Pre-Order Traversal | 0.86 (0.11) | 0.81 (0.08) | 0.78 (0.11) |
| | | Post-Order Traversal | 0.31 (0.04) | 0.04 (0.02) | 0.00 (0.00) |
| | | Depth | 0.70 (0.00) | 0.38 (0.07) | 0.13 (0.06) |
| | | Compound | 0.26 (0.02) | 0.01 (0.02) | 0.00 (0.00) |
| | Heap* | Compound | 0.17 (0.03) | 0.03 (0.00) | 0.00 (0.00) |
| | | Heapify | 0.24 (0.05) | 0.00 (0.00) | 0.00 (0.00) |
| | RB Tree* | Construct | 0.00 (0.00) | 0.00 (0.00) | 0.00 (0.00) |
| | | Compound | 0.31 (0.02) | 0.00 (0.00) | 0.00 (0.00) |
| | $B^+$ Tree* | Compound | 0.02 (0.04) | 0.00 (0.00) | 0.00 (0.00) |
| | K-D Tree* | Construct | 0.00 (0.00) | 0.00 (0.00) | 0.00 (0.00) |
| | K-D Heap* | Compound | 0.02 (0.02) | 0.02 (0.02) | 0.00 (0.00) |
| Network | Graph | Breadth-First Traversal | 0.07 (0.06) | 0.00 (0.00) | 0.00 (0.00) |
| | | Depth-First Traversal | 0.04 (0.05) | 0.01 (0.02) | 0.00 (0.00) |
| | DSU* | Compound | 0.00 (0.00) | 0.00 (0.00) | 0.00 (0.00) |
| | Geom Graph* | Construct | 0.07 (0.03) | 0.00 (0.00) | 0.00 (0.00) |
| Hybrid | Bloom Filter* | Compound | 0.00 (0.00) | 0.07 (0.00) | 0.00 (0.00) |
| | DAWG* | Compound | 0.02 (0.02) | 0.00 (0.00) | 0.00 (0.00) |

## A.5  ACCURACY BY PROMPTING METHODS ACROSS INSTRUCTION-TUNED MODELS

This section presents additional accuracy tables for tasks in `DSR-Bench`, evaluating each instruction-tuned model across five prompting methods: **Stepwise**, **0-CoT**, **CoT**, **3-shot**, and **None**. The results are shown in Table 23 (GPT-4.1), Table 24 (Gemini-2.0-Flash), Table 25 (Claude-3.5-Sonnet), Table 26 (DeepSeek-V3), and Table 27 (Llama-3.3).

Table 23: Mean (± std) accuracy of **GPT-4.1** across prompting methods over three runs.

| Data structure | Task | Stepwise | 0-CoT | CoT | 3-shot | None |
|---|---|---|---|---|---|---|
| Array | Access | 1.00 (0.00) | 1.00 (0.00) | 1.00 (0.00) | 1.00 (0.00) | 1.00 (0.00) |
| | Delete | 1.00 (0.00) | 1.00 (0.00) | 1.00 (0.00) | 1.00 (0.00) | 1.00 (0.00) |
| | Insert | 1.00 (0.00) | 1.00 (0.00) | 1.00 (0.00) | 1.00 (0.00) | 0.91 (0.08) |
| | Reverse | 1.00 (0.00) | 1.00 (0.00) | 1.00 (0.00) | 0.99 (0.02) | 0.98 (0.02) |
| | Search | 1.00 (0.00) | 1.00 (0.00) | 1.00 (0.00) | 1.00 (0.00) | 1.00 (0.00) |
| Queue | Compound | 1.00 (0.00) | 1.00 (0.00) | 1.00 (0.00) | 1.00 (0.00) | 0.82 (0.04) |
| Stack | Compound | 1.00 (0.00) | 1.00 (0.00) | 1.00 (0.00) | 1.00 (0.00) | 0.97 (0.00) |
| LRU Cache | Cache | 1.00 (0.00) | 1.00 (0.00) | 1.00 (0.00) | 1.00 (0.00) | 0.94 (0.02) |
| Priority Queue | Compound | 0.94 (0.04) | 0.99 (0.01) | 0.91 (0.02) | 0.94 (0.02) | 0.63 (0.03) |
| Hashmap | Compound | 0.96 (0.02) | 0.99 (0.01) | 1.00 (0.00) | 1.00 (0.00) | 0.19 (0.07) |
| Trie | Compound | 0.82 (0.02) | 0.98 (0.00) | 0.77 (0.07) | 0.68 (0.02) | 0.39 (0.07) |
| Suffix Tree | Construct | 0.49 (0.07) | 0.87 (0.01) | 0.69 (0.04) | 0.28 (0.08) | 0.00 (0.00) |
| Skip List | Compound | 0.77 (0.03) | 0.94 (0.01) | 0.56 (0.10) | 0.84 (0.02) | 0.21 (0.02) |
| BST | Insert | 0.99 (0.02) | 1.00 (0.00) | 0.97 (0.00) | 0.99 (0.02) | 0.79 (0.04) |
| | Remove | 0.99 (0.02) | 1.00 (0.00) | 0.99 (0.02) | 1.00 (0.00) | 0.78 (0.04) |
| | In-Order Traversal | 0.98 (0.02) | 1.00 (0.00) | 1.00 (0.00) | 0.98 (0.02) | 1.00 (0.00) |
| | Pre-Order Traversal | 1.00 (0.00) | 1.00 (0.00) | 1.00 (0.00) | 1.00 (0.00) | 1.00 (0.00) |
| | Post-Order Traversal | 1.00 (0.00) | 1.00 (0.00) | 1.00 (0.00) | 1.00 (0.00) | 0.82 (0.02) |
| | Depth | 1.00 (0.00) | 0.99 (0.02) | 1.00 (0.00) | 1.00 (0.00) | 0.30 (0.04) |
| | Compound | 1.00 (0.00) | 1.00 (0.00) | 0.98 (0.02) | 1.00 (0.00) | 0.69 (0.02) |
| Heap | Compound | 0.77 (0.07) | 0.96 (0.01) | 0.87 (0.06) | 0.78 (0.14) | 0.58 (0.02) |
| | Heapify | 0.99 (0.02) | 1.00 (0.01) | 0.96 (0.04) | 0.93 (0.07) | 0.57 (0.03) |
| RB Tree | Construct | 0.40 (0.13) | 0.91 (0.02) | 0.40 (0.12) | 0.38 (0.05) | 0.12 (0.02) |
| | Compound | 0.77 (0.03) | 0.96 (0.01) | 0.37 (0.12) | 0.70 (0.07) | 0.31 (0.04) |
| B+ Tree | Compound | 0.71 (0.05) | 0.93 (0.01) | 0.77 (0.03) | 0.60 (0.06) | 0.27 (0.00) |
| Graph | Breadth-First Traversal | 0.90 (0.03) | 0.94 (0.02) | 0.83 (0.07) | 0.82 (0.07) | 0.31 (0.05) |
| | Depth-First Traversal | 0.86 (0.05) | 0.92 (0.02) | 0.83 (0.03) | 0.80 (0.03) | 0.50 (0.03) |
| DSU | Compound | 0.67 (0.03) | 0.93 (0.02) | 0.67 (0.07) | 0.62 (0.05) | 0.06 (0.02) |
| Bloom Filter | Compound | 0.36 (0.07) | 0.91 (0.02) | 0.26 (0.08) | 0.42 (0.07) | 0.10 (0.00) |
| DAWG | Compound | 0.20 (0.00) | 0.79 (0.01) | 0.21 (0.02) | 0.20 (0.00) | 0.16 (0.02) |

Table 24: Mean (± std) accuracy of **Gemini-2.0-Flash** across prompting methods over three runs.

| Data structure | Task | Stepwise | 0-CoT | CoT | 3-shot | None |
|---|---|---|---|---|---|---|
| Array | Access | 1.00 (0.00) | 1.00 (0.00) | 1.00 (0.00) | 1.00 (0.00) | 1.00 (0.00) |
| | Delete | 1.00 (0.00) | 1.00 (0.00) | 1.00 (0.00) | 1.00 (0.00) | 0.96 (0.02) |
| | Insert | 1.00 (0.00) | 1.00 (0.00) | 1.00 (0.00) | 1.00 (0.00) | 0.99 (0.02) |
| | Reverse | 0.67 (0.00) | 0.78 (0.19) | 0.56 (0.19) | 1.00 (0.00) | 0.96 (0.02) |
| | Search | 1.00 (0.00) | 1.00 (0.00) | 1.00 (0.00) | 1.00 (0.00) | 1.00 (0.00) |
| Queue | Compound | 0.91 (0.05) | 0.93 (0.03) | 0.94 (0.02) | 0.94 (0.02) | 0.87 (0.00) |
| Stack | Compound | 0.93 (0.07) | 0.92 (0.04) | 0.73 (0.09) | 0.97 (0.03) | 0.67 (0.00) |
| LRU Cache | Cache | 0.97 (0.00) | 0.96 (0.04) | 0.82 (0.04) | 0.87 (0.06) | 0.93 (0.00) |
| Priority Queue | Compound | 0.53 (0.12) | 0.62 (0.12) | 0.53 (0.03) | 0.72 (0.05) | 0.38 (0.02) |
| Hashmap | Compound | 0.42 (0.08) | 0.56 (0.11) | 0.67 (0.09) | 0.63 (0.07) | 0.28 (0.05) |
| Trie | Compound | 0.26 (0.04) | 0.27 (0.12) | 0.19 (0.07) | 0.33 (0.03) | 0.31 (0.02) |
| Suffix Tree | Construct | 0.13 (0.00) | 0.11 (0.05) | 0.22 (0.07) | 0.18 (0.05) | 0.00 (0.00) |
| Skip List | Compound | 0.18 (0.04) | 0.31 (0.08) | 0.27 (0.03) | 0.31 (0.08) | 0.16 (0.02) |
| BST | Insert | 0.62 (0.02) | 0.58 (0.13) | 0.66 (0.02) | 0.64 (0.08) | 0.31 (0.02) |
| | Remove | 0.64 (0.12) | 0.67 (0.07) | 0.73 (0.03) | 0.69 (0.08) | 0.63 (0.09) |
| | In-Order Traversal | 0.87 (0.03) | 0.86 (0.02) | 0.92 (0.02) | 0.80 (0.03) | 0.87 (0.00) |
| | Pre-Order Traversal | 1.00 (0.00) | 1.00 (0.00) | 1.00 (0.00) | 1.00 (0.00) | 1.00 (0.00) |
| | Post-Order Traversal | 0.86 (0.05) | 0.78 (0.04) | 0.86 (0.04) | 0.84 (0.08) | 0.63 (0.00) |
| | Depth | 1.00 (0.00) | 1.00 (0.00) | 1.00 (0.00) | 0.86 (0.02) | 0.13 (0.09) |
| | Compound | 0.66 (0.08) | 0.74 (0.08) | 0.80 (0.03) | 0.69 (0.08) | 0.51 (0.05) |
| Heap | Compound | 0.40 (0.09) | 0.37 (0.07) | 0.44 (0.13) | 0.39 (0.05) | 0.32 (0.05) |
| | Heapify | 0.51 (0.11) | 0.61 (0.05) | 0.43 (0.09) | 0.54 (0.08) | 0.23 (0.06) |
| RB Tree | Construct | 0.08 (0.07) | 0.07 (0.03) | 0.04 (0.02) | 0.03 (0.00) | 0.08 (0.02) |
| | Compound | 0.41 (0.08) | 0.44 (0.02) | 0.28 (0.04) | 0.31 (0.02) | 0.43 (0.03) |
| B+ Tree | Compound | 0.33 (0.09) | 0.39 (0.08) | 0.47 (0.03) | 0.50 (0.09) | 0.17 (0.00) |
| Graph | Breadth-First Traversal | 0.17 (0.03) | 0.24 (0.10) | 0.36 (0.07) | 0.20 (0.03) | 0.10 (0.00) |
| | Depth-First Traversal | 0.20 (0.09) | 0.27 (0.12) | 0.30 (0.03) | 0.08 (0.02) | 0.19 (0.02) |
| DSU | Compound | 0.29 (0.02) | 0.20 (0.00) | 0.17 (0.09) | 0.12 (0.02) | 0.01 (0.02) |
| Bloom Filter | Compound | 0.33 (0.07) | 0.20 (0.00) | 0.12 (0.04) | 0.29 (0.08) | 0.10 (0.00) |
| DAWG | Compound | 0.16 (0.02) | 0.18 (0.02) | 0.13 (0.00) | 0.14 (0.02) | 0.18 (0.02) |

Table 25: Mean (± std) accuracy of **Claude-3.5-Sonnet** across prompting methods over three runs.

| Data structure | Task | Stepwise | 0-CoT | CoT | 3-shot | None |
|---|---|---|---|---|---|---|
| Array | Access | 1.00 (0.00) | 1.00 (0.00) | 1.00 (0.00) | 1.00 (0.00) | 1.00 (0.00) |
| | Delete | 1.00 (0.00) | 1.00 (0.00) | 1.00 (0.00) | 1.00 (0.00) | 1.00 (0.00) |
| | Insert | 1.00 (0.00) | 1.00 (0.00) | 1.00 (0.00) | 1.00 (0.00) | 1.00 (0.00) |
| | Reverse | 1.00 (0.00) | 0.99 (0.02) | 0.96 (0.04) | 0.99 (0.02) | 1.00 (0.00) |
| | Search | 1.00 (0.00) | 1.00 (0.00) | 1.00 (0.00) | 1.00 (0.00) | 1.00 (0.00) |
| Queue | Compound | 1.00 (0.00) | 1.00 (0.00) | 1.00 (0.00) | 1.00 (0.00) | 0.87 (0.00) |
| Stack | Compound | 1.00 (0.00) | 1.00 (0.00) | 1.00 (0.00) | 1.00 (0.00) | 1.00 (0.00) |
| LRU Cache | Cache | 1.00 (0.00) | 1.00 (0.00) | 1.00 (0.00) | 1.00 (0.00) | 0.99 (0.02) |
| Priority Queue | Compound | 0.94 (0.02) | 0.96 (0.02) | 0.90 (0.03) | 0.94 (0.02) | 0.63 (0.00) |
| Hashmap | Compound | 0.89 (0.02) | 0.81 (0.04) | 1.00 (0.00) | 1.00 (0.00) | 0.37 (0.03) |
| Trie | Compound | 0.02 (0.02) | 0.11 (0.07) | 0.00 (0.00) | 0.11 (0.02) | 0.89 (0.04) |
| Suffix Tree | Construct | 0.29 (0.08) | 0.24 (0.07) | 0.56 (0.08) | 0.27 (0.03) | 0.21 (0.02) |
| Skip List | Compound | 0.82 (0.02) | 0.77 (0.03) | 0.64 (0.04) | 0.61 (0.02) | 0.77 (0.06) |
| BST | Insert | 1.00 (0.00) | 1.00 (0.00) | 0.96 (0.04) | 0.92 (0.05) | 0.80 (0.06) |
| | Remove | 1.00 (0.00) | 1.00 (0.00) | 0.97 (0.00) | 0.98 (0.02) | 0.96 (0.04) |
| | In-Order Traversal | 0.88 (0.05) | 0.97 (0.06) | 1.00 (0.00) | 0.98 (0.02) | 0.97 (0.03) |
| | Pre-Order Traversal | 1.00 (0.00) | 1.00 (0.00) | 1.00 (0.00) | 1.00 (0.00) | 1.00 (0.00) |
| | Post-Order Traversal | 0.99 (0.02) | 0.99 (0.02) | 1.00 (0.00) | 1.00 (0.00) | 1.00 (0.00) |
| | Depth | 1.00 (0.00) | 0.99 (0.02) | 1.00 (0.00) | 1.00 (0.00) | 1.00 (0.00) |
| | Compound | 0.89 (0.05) | 0.88 (0.05) | 0.88 (0.04) | 0.86 (0.05) | 0.77 (0.07) |
| Heap | Compound | 0.69 (0.02) | 0.69 (0.02) | 0.66 (0.04) | 0.68 (0.02) | 0.78 (0.04) |
| | Heapify | 0.73 (0.00) | 0.69 (0.04) | 0.33 (0.06) | 0.80 (0.07) | 0.53 (0.12) |
| RB Tree | Construct | 0.11 (0.08) | 0.08 (0.05) | 0.19 (0.07) | 0.21 (0.02) | 0.13 (0.00) |
| | Compound | 0.57 (0.00) | 0.60 (0.03) | 0.03 (0.00) | 0.13 (0.03) | 0.44 (0.02) |
| B+ Tree | Compound | 0.61 (0.02) | 0.69 (0.04) | 0.67 (0.03) | 0.44 (0.05) | 0.40 (0.02) |
| Graph | Breadth-First Traversal | 0.30 (0.09) | 0.32 (0.04) | 0.52 (0.02) | 0.26 (0.04) | 0.17 (0.03) |
| | Depth-First Traversal | 0.30 (0.09) | 0.24 (0.04) | 0.26 (0.05) | 0.23 (0.03) | 0.13 (0.03) |
| DSU | Compound | 0.53 (0.07) | 0.49 (0.05) | 0.76 (0.08) | 0.53 (0.09) | 0.07 (0.03) |
| Bloom Filter | Compound | 0.12 (0.04) | 0.12 (0.02) | 0.10 (0.00) | 0.10 (0.00) | 0.10 (0.00) |
| DAWG | Compound | 0.17 (0.06) | 0.19 (0.02) | 0.18 (0.02) | 0.19 (0.02) | 0.20 (0.00) |

Table 26: Mean (± std) accuracy of **DeepSeek-V3** across prompting methods over three runs.

| Data structure | Task | Stepwise | 0-CoT | CoT | 3-shot | None |
|---|---|---|---|---|---|---|
| Array | Access | 1.00 (0.00) | 1.00 (0.00) | 1.00 (0.00) | 1.00 (0.00) | 1.00 (0.00) |
| | Delete | 1.00 (0.00) | 1.00 (0.00) | 1.00 (0.00) | 1.00 (0.00) | 1.00 (0.00) |
| | Insert | 1.00 (0.00) | 1.00 (0.00) | 1.00 (0.00) | 1.00 (0.00) | 1.00 (0.00) |
| | Reverse | 1.00 (0.00) | 1.00 (0.00) | 0.99 (0.00) | 1.00 (0.00) | 1.00 (0.00) |
| | Search | 1.00 (0.00) | 1.00 (0.00) | 1.00 (0.00) | 1.00 (0.00) | 1.00 (0.00) |
| Queue | Compound | 1.00 (0.00) | 1.00 (0.00) | 1.00 (0.00) | 1.00 (0.00) | 0.84 (0.02) |
| Stack | Compound | 1.00 (0.00) | 1.00 (0.00) | 1.00 (0.00) | 1.00 (0.00) | 0.70 (0.03) |
| LRU Cache | Cache | 0.93 (0.00) | 0.97 (0.00) | 0.98 (0.02) | 0.90 (0.00) | 0.94 (0.02) |
| Priority Queue | Compound | 0.86 (0.04) | 0.79 (0.05) | 0.84 (0.02) | 0.82 (0.02) | 0.53 (0.03) |
| Hashmap | Compound | 1.00 (0.00) | 1.00 (0.00) | 0.90 (0.03) | 1.00 (0.00) | 0.04 (0.02) |
| Trie | Compound | 0.00 (0.00) | 0.00 (0.00) | 0.63 (0.02) | 0.64 (0.02) | 0.00 (0.00) |
| Suffix Tree | Construct | 0.19 (0.02) | 0.18 (0.02) | 0.39 (0.04) | 0.19 (0.02) | 0.06 (0.02) |
| Skip List | Compound | 0.08 (0.02) | 0.06 (0.02) | 0.19 (0.04) | 0.08 (0.02) | 0.06 (0.02) |
| BST | Insert | 0.94 (0.02) | 0.98 (0.02) | 0.91 (0.02) | 0.87 (0.06) | 0.93 (0.03) |
| | Remove | 0.92 (0.04) | 0.70 (0.03) | 0.84 (0.04) | 0.82 (0.05) | 0.84 (0.04) |
| | In-Order Traversal | 0.94 (0.04) | 0.92 (0.02) | 0.94 (0.04) | 0.93 (0.00) | 0.97 (0.00) |
| | Pre-Order Traversal | 1.00 (0.00) | 1.00 (0.00) | 1.00 (0.00) | 1.00 (0.00) | 1.00 (0.00) |
| | Post-Order Traversal | 1.00 (0.00) | 1.00 (0.00) | 1.00 (0.00) | 0.99 (0.02) | 0.82 (0.02) |
| | Depth | 0.67 (0.03) | 1.00 (0.00) | 1.00 (0.00) | 1.00 (0.00) | 0.82 (0.31) |
| | Compound | 0.81 (0.02) | 0.77 (0.03) | 0.96 (0.02) | 0.71 (0.04) | 0.68 (0.02) |
| RB Tree | Construct | 0.04 (0.04) | 0.02 (0.04) | 0.06 (0.02) | 0.07 (0.06) | 0.09 (0.02) |
| | Compound | 0.63 (0.03) | 0.67 (0.03) | 0.12 (0.02) | 0.59 (0.08) | 0.30 (0.03) |
| B+ Tree | Compound | 0.71 (0.04) | 0.66 (0.02) | 0.44 (0.08) | 0.38 (0.02) | 0.14 (0.05) |
| Heap | Compound | 0.83 (0.03) | 0.87 (0.03) | 0.87 (0.03) | 0.89 (0.05) | 0.23 (0.00) |
| | Heapify | 0.83 (0.06) | 0.83 (0.03) | 0.57 (0.03) | 0.81 (0.02) | 0.59 (0.02) |
| Graph | Breadth-First Traversal | 0.51 (0.02) | 0.51 (0.04) | 0.29 (0.08) | 0.00 (0.00) | 0.29 (0.05) |
| | Depth-First Traversal | 0.23 (0.03) | 0.39 (0.05) | 0.36 (0.05) | 0.00 (0.00) | 0.22 (0.02) |
| DSU | Compound | 0.34 (0.04) | 0.41 (0.05) | 0.30 (0.03) | 0.16 (0.02) | 0.03 (0.00) |
| Bloom Filter | Compound | 0.10 (0.00) | 0.10 (0.00) | 0.10 (0.00) | 0.10 (0.00) | 0.10 (0.00) |
| DAWG | Compound | 0.18 (0.02) | 0.18 (0.02) | 0.17 (0.00) | 0.18 (0.02) | 0.17 (0.00) |

Table 27: Mean (± std) accuracy of **Llama-3.3** across prompting methods over three runs.

| Data structure | Task | Stepwise | 0-CoT | CoT | 3-shot | None |
|---|---|---|---|---|---|---|
| Array | Access | 0.98 (0.02) | 1.00 (0.00) | 0.97 (0.03) | 0.99 (0.02) | 0.98 (0.02) |
| | Delete | 0.92 (0.05) | 0.96 (0.02) | 0.78 (0.05) | 0.82 (0.08) | 0.81 (0.08) |
| | Insert | 0.78 (0.02) | 0.88 (0.07) | 0.81 (0.04) | 0.89 (0.10) | 0.76 (0.04) |
| | Reverse | 0.96 (0.02) | 0.87 (0.06) | 0.50 (0.07) | 0.74 (0.02) | 0.91 (0.02) |
| | Search | 0.99 (0.02) | 0.99 (0.02) | 0.96 (0.02) | 0.98 (0.02) | 1.00 (0.00) |
| Skip List | Compound | 0.14 (0.04) | 0.10 (0.03) | 0.18 (0.02) | 0.20 (0.07) | 0.03 (0.00) |
| Queue | Compound | 0.93 (0.03) | 0.94 (0.05) | 0.94 (0.02) | 0.81 (0.05) | 0.58 (0.11) |
| Stack | Compound | 0.88 (0.02) | 0.86 (0.13) | 0.88 (0.10) | 0.68 (0.07) | 0.09 (0.02) |
| LRU Cache | Compound | 0.89 (0.02) | 0.92 (0.05) | 0.98 (0.04) | 0.82 (0.07) | 0.74 (0.08) |
| Priority Queue | Compound | 0.73 (0.03) | 0.73 (0.10) | 0.69 (0.02) | 0.76 (0.04) | 0.21 (0.05) |
| Hashmap | Compound | 0.39 (0.16) | 0.23 (0.07) | 0.32 (0.04) | 0.13 (0.00) | 0.00 (0.00) |
| Trie | Compound | 0.01 (0.02) | 0.00 (0.00) | 0.10 (0.09) | 0.14 (0.10) | 0.01 (0.02) |
| Suffix Tree | Construct | 0.01 (0.02) | 0.01 (0.02) | 0.07 (0.03) | 0.00 (0.00) | 0.00 (0.00) |
| BST | Insert | 0.53 (0.07) | 0.52 (0.16) | 0.38 (0.05) | 0.37 (0.09) | 0.37 (0.00) |
| | Remove | 0.57 (0.03) | 0.57 (0.15) | 0.40 (0.25) | 0.60 (0.06) | 0.49 (0.04) |
| | In-Order Traversal | 0.56 (0.08) | 0.51 (0.13) | 0.69 (0.02) | 0.60 (0.12) | 0.60 (0.06) |
| | Pre-Order Traversal | 0.60 (0.15) | 0.78 (0.10) | 0.67 (0.03) | 0.82 (0.07) | 0.86 (0.11) |
| | Post-Order Traversal | 0.54 (0.26) | 0.61 (0.05) | 0.90 (0.03) | 0.71 (0.04) | 0.31 (0.04) |
| | Depth | 0.70 (0.00) | 0.83 (0.03) | 0.82 (0.08) | 0.99 (0.02) | 0.93 (0.03) |
| | Compound | 0.54 (0.16) | 0.51 (0.05) | 0.50 (0.09) | 0.49 (0.10) | 0.27 (0.15) |
| Heap | Compound | 0.33 (0.06) | 0.33 (0.03) | 0.32 (0.05) | 0.41 (0.11) | 0.17 (0.03) |
| | Heapify | 0.29 (0.02) | 0.23 (0.12) | 0.08 (0.02) | 0.18 (0.05) | 0.24 (0.05) |
| RB Tree | Construct | 0.01 (0.02) | 0.01 (0.02) | 0.01 (0.02) | 0.07 (0.03) | 0.00 (0.00) |
| | Compound | 0.36 (0.04) | 0.30 (0.10) | 0.03 (0.00) | 0.28 (0.07) | 0.31 (0.02) |
| B+ Tree | Compound | 0.17 (0.03) | 0.18 (0.02) | 0.50 (0.00) | 0.23 (0.03) | 0.02 (0.04) |
| Graph | Breadth-First Traversal | 0.12 (0.02) | 0.12 (0.07) | 0.28 (0.08) | 0.06 (0.07) | 0.04 (0.05) |
| | Depth-First Traversal | 0.12 (0.08) | 0.14 (0.02) | 0.09 (0.05) | 0.10 (0.07) | 0.07 (0.06) |
| DSU | Construct | 0.08 (0.05) | 0.04 (0.02) | 0.37 (0.15) | 0.07 (0.00) | 0.00 (0.00) |
| Bloom Filter | Compound | 0.01 (0.02) | 0.01 (0.02) | 0.00 (0.00) | 0.01 (0.02) | 0.00 (0.00) |
| DAWG | Compound | 0.03 (0.03) | 0.02 (0.02) | 0.06 (0.02) | 0.07 (0.06) | 0.02 (0.02) |

## A.6 ADDITIONAL ANALYSIS ON CoT PROMPTING

In Section 4.1.1, we analyzed various prompting strategies with instruction-tuned models. We further examined the CoT method by inspecting its reasoning outputs, which offers insights we hope will benefit practitioners and researchers:

- **CoT offers limited benefits for well-known tasks.** For familiar problems that models are likely to have encountered during pretraining (such as ARRAY, QUEUE, and BINARY SEARCH TREE), CoT prompting yields only marginal gains. This suggests that models already possess internalized procedures for these tasks and can execute them reliably without additional reasoning steps. Simpler prompts can be more cost efficient and effective.
- **Without careful design, CoT can hurt performance.** In more complex scenarios, such as the HASHMAP compound task, using CoT with structured JSON-style reasoning actually degraded accuracy. When the reasoning format was changed to natural language, performance recovered to match the baseline (0-CoT). This highlights that the effectiveness of CoT is highly sensitive to prompt design, and poorly chosen reasoning formats may introduce unnecessary complexity rather than aiding problem solving.

## A.7 THE SPATIAL SUITE SUPPLEMENTARY MATERIALS

Figure 3 presents example illustrations of the non-uniform input distributions: circles, moons, and blobs, which were adopted from scikit-learn. These synthetic patterns are used to evaluate whether models can adapt to irregular and non-uniform spatial distributions, an essential aspect of real-world data, as discussed in Section 4.2.

We observe a performance drop in KD-TREE tasks when the input data is non-uniform. At the problem level, uniform and non-uniform KD-TREE tasks appear equally difficult for a human reasoner. Across 30 questions per group, duplicated indices, which trigger tie-breaking, occur at similar rates on both axes (x: 32 vs. 33; y: 26 vs. 28), and the total number of median operations is comparable

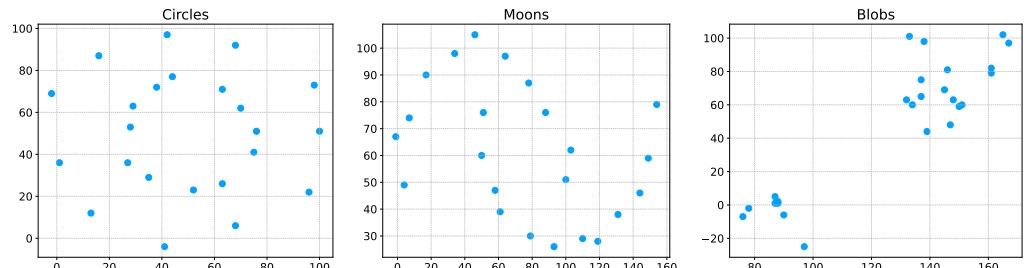

Figure 3: Example K-D Tree instances from three non-uniform distributions.

(156 vs. 145). This suggests the surface-level difficulty is well matched across distributions. A closer inspection of errors reveals two main causes:

**Non-uniform data often increases ambiguity near the global median.** Non-uniform distributions often produce clusters of similar or identical coordinates near the global median, making it harder for the model to select the correct split. Since the root node is determined by a median over the entire dataset, this ambiguity leads to more errors. Supporting this, we observed 13 out of 30 correct root nodes on uniform data, but only 5 out of 30 on circle (non-uniform) data.

**Even simple tasks, such as median calculation, can be challenging for LLMs.** On non-uniform data, models exhibited 28.6% more cases of violating the prompt's "latter-median" rule by instead choosing the "former-median," and 50% more instances of selecting a completely incorrect median (neither the former nor latter). Additionally, while no axis confusion errors occurred on uniform data, the model made three such errors on non-uniform data, mistakenly selecting the wrong axis before performing median calculation.

## A.8 THE NATURAL SUITE SUPPLEMENTARY MATERIALS

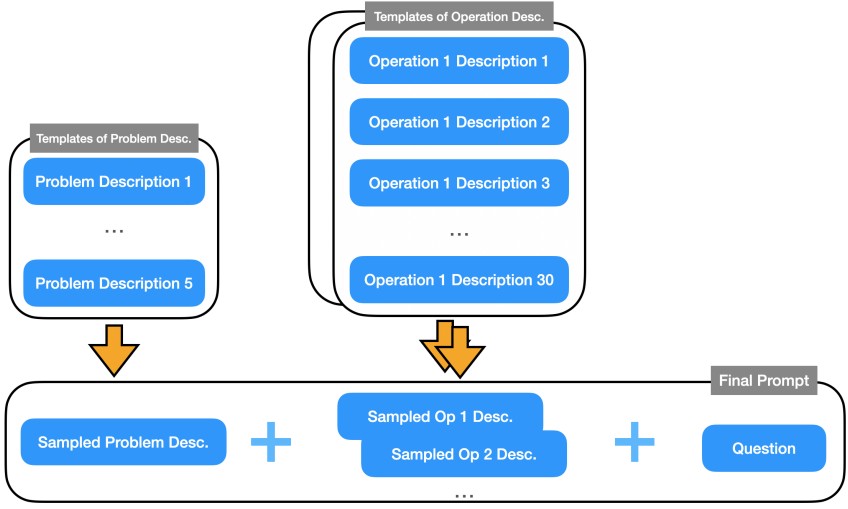

Figure 4: The pipeline for generating natural language prompts.

Figure 4 illustrates our generation process for natural language prompts in DSR-Bench-natural. For each data structure task, we begin by manually writing an initial scenario narrative (see On a sunny afternoon... arrived in the queue" in Section A.8). This narrative is then paraphrased into five variants using GPT-4o. For each supported operation type (e.g., enqueue, dequeue), we generate 30 paraphrased operation templates in a similar manner (e.g., With money in hand, Yuki Lopez stepped

into the queue." corresponds to "enqueue Yuki Lopez" in Section A.8). In total, each task is backed by a pool of five scenario descriptions and 30 templates per operation type.

We use the same input distribution as in the `DSR-Bench`, replacing synthetic values (e.g. "enqueue 5") with realistic values (e.g. "enqueue Yuki Lopez"). During prompt construction, we randomly sample one scenario description. For each data structure operation, we instantiate it with a synthetically generated value (e.g., a name) and sample one operation template to form a complete instance. All templates were reviewed by three human annotators to ensure clarity and unambiguous solvability.

**Queue (children buying ice cream)**   We construct a real-world scenario that implicitly models a QUEUE: children lining up to buy ice cream from a truck. The enqueue operation corresponds to a child joining the end of the line, while the dequeue operation represents a child being served from the front. The scenario explicitly enforces the FIFO discipline by stating that no skipping is allowed.

---

**Example prompt from the natural language extension for QUEUE.**

On a sunny afternoon in the neighborhood park, an ice cream truck rolled in, its cheerful tune drawing children from all directions. The children began to form a line. Each child joined at the end while the vendor served at the front. Coins jingled in pockets while the children eagerly discussed the different flavors. The children were served in the order they had arrived in the queue.

- Fatima Singh ran over from the swings and joined the ice cream line.
- With money in hand, Yuki Lopez stepped into the queue.
- Haruto Sanchez spotted the growing line and quietly joined.
- A cone was handed over, and the line moved on.
- After hearing about the ice cream truck from Fatima Singh, Isabella Miller decided to line up too.
- One more excited customer walked off with a cone in hand.
- Carlos Martinez joined the queue after Yuki Lopez mentioned how good the ice cream looked.

**Q**: What is the order of the remaining kids in line? Your answer should be a list of names. Answer the question in 8000 tokens.

---

**BST (clinic appointments)**   We construct a scenario in which a clinic uses a BST to store patient appointments, ordered by appointment time and tie-broken by the patient's name. The insert operation adds a (name, time) appointment to the tree, while the delete operation corresponds to a patient canceling their appointment. To retrieve all records, a pre-order traversal of the tree is performed.

---

**Example prompt from the natural language extension for BST.**

A local clinic uses an appointment management system which maintains a binary search tree to store appointments. Each appointment (name, appointment time) is a tuple of two strings, e.g. ('Alice Baker', '10:30'), and is represented by a node in the tree. Order is maintained by appointment time. The alphabetical order of the patients' names is used to break ties. During data retrieval (i.e. to print out all of the appointments), a pre-order traversal is used, starting from the root node. Initially the tree is empty.

- Hassan Chen joined the list at 09:22 successfully.
- Knowing Hassan Chen has booked, Amelia Martinez was placed at 13:11.
- Hassan Chen hesitated a lot but still decided to cancel.
- As recommended by a friend Harper Young, Lucas Fernandez quickly booked at 09:18.
- Harper Young was scheduled for 15:48, slightly earlier than 16:01.

**Q**: What is the pre-order traversal of the appointment schedule following the binary search tree? Your answer should be a list of (name, appointment time) in the format of a tuple of two strings. Answer the question in 8000 tokens.

---

**Graph (galaxy traveling)**   We create a scenario set in a galaxy where planets are connected by space tunnels. The task is to navigate a starship to visit as many planets as possible using depth-first search, starting from a given planet and visiting neighbors in lexicographical order.

---

**Example prompt from the natural language extension for GRAPH.**

You pilot a Star Courier through a galaxy of planets. Your job is to travel to as many planets as possible via bidirectional space tunnels, starting from a source planet. The courier computes its route with depth-first search, and whenever multiple unvisited neighbors are available it selects the neighbor with the alphabetically earliest planet name.

- Star maps show a space tunnel running between Triton and Pulsar.
- A space tunnel links Ganymede and Wraith.
- The tunnel from Triton to Ganymede is well-known for its convenience.
- There's a tunnel between Fenrir and Vega.
- The tunnel linking Fenrir and Pulsar is a crucial route for all space dwellers.
- Long-range scans confirm a navigable tunnel between Triton and Orion.
- Though Vega is nearby, the space team decides to connect Nereus and Pulsar via a tunnel.
- Vega and Orion are part of the same local cluster, connected by a space tunnel.
- Vega and Ymir are directly linked by a tunnel monitored by the space police.
- The ancient network includes a direct tunnel between Wraith and Pulsar.
- Only Pulsar and Orion are reachable via this tunnel — not Nereus.

**Q**: What is the full DFS traversal order (as a list of planet names) starting from Fenrir? Answer the question in 8000 tokens.

---

### A.9   THE `CODE` SUITE SUPPLEMENTARY MATERIALS

Our benchmark is designed to evaluate *general reasoning* in LLMs without relying on external tools such as code interpreters or formal solvers. The focus is on the underlying reasoning ability that drives broad problem-solving, beyond coding or domains where code is directly applicable. This follows recent evaluation practices, such as Gemini-Deep-Think and OpenAI's IMO assessments (Luong & Lockhart, 2025; Wei, 2025), which deliberately disallowed code or tool use (e.g., Lean) to test end-to-end reasoning. By using data structures as controlled, interpretable settings for structural reasoning—a core component of general reasoning—`DSR-Bench` differs from code-synthesis benchmarks that emphasise syntax or API knowledge and are often vulnerable to data leakage.

To additionally probe how code generation may support structural reasoning, we conducted an ablation across six models, including reasoning models with competitive performance on code-synthesis benchmarks. We cover seven data structures of varying difficulty (ARRAY, QUEUE, HASHMAP, HEAP, DAWG, GEOM GRAPH, GRAPH-NATURAL) and average results over three runs. Three code-generation modes were tested, with prompts shown below:

- *CodeOnly*: "Your answer should be a Python function 'def solution()' that takes no inputs, solves the problem, and returns the final solution in the expected format. Output the code only."
- *CodeEnforce*: "You should write code to solve the problem, then reason through its execution to explain what the output would be. Your final answer should be the output itself."
- *CodeMaybe*: "You can write code to help solve the problem, or solve it directly. If you use code, reason through its execution to explain what the output would be. Your final answer should be the output itself."

While GPT-4.1 and o4-mini generally perform well, we observe that Claude models also perform competitively, particularly in spatial data structure tasks such as GEOM GRAPH. The only exception is GRAPH-NATURAL, suggesting that Claude models struggle with handling natural language ambiguity. On the other hand, Gemini models tend to return final answers directly rather than code in *CodeOnly* mode, despite explicit instructions ("Your answer should be a Python function... Output the code only."). This indicates weaker instruction-following in code generation compared to other models and

Table 28: Individual scores with code generation by GPT-4.1 and o4-mini.

| | GPT-4.1 | | | | o4-mini | | | |
|---|---|---|---|---|---|---|---|---|
| **Structure** | None | CodeMaybe | CodeEnforce | CodeOnly | None | CodeMaybe | CodeEnforce | CodeOnly |
| Array | 1.00 | 1.00 | 1.00 | 1.00 | 1.00 | 1.00 | 1.00 | 1.00 |
| Queue | 0.82 | 0.84 | 0.89 | 0.98 | 1.00 | 1.00 | 1.00 | 1.00 |
| Hashmap | 0.19 | 0.11 | 0.09 | 1.00 | 0.89 | 0.89 | 0.86 | 0.94 |
| Heap | 0.58 | 0.53 | 0.53 | 1.00 | 0.44 | 0.73 | 0.70 | 0.53 |
| DAWG | 0.16 | 0.17 | 0.11 | 0.90 | 0.49 | 0.46 | 0.30 | 0.56 |
| Geom Graph | 0.03 | 0.02 | 0.04 | 0.93 | 0.83 | 0.98 | 0.97 | 0.99 |
| Graph-Natural | 0.01 | 0.00 | 0.00 | 0.86 | 0.43 | 0.26 | 0.39 | 0.69 |

Table 29: Individual scores with code generation by Gemini-2.0-Flash.

| **Structure** | None | CodeMaybe | CodeEnforce | CodeOnly | Accuracy | Frequency |
|---|---|---|---|---|---|---|
| Array | 1.00 | 1.00 | 1.00 | 0.01 | 1.00 | 0.01 |
| Queue | 0.87 | 0.86 | 0.87 | 0.13 | 0.93 | 0.14 |
| Hashmap | 0.28 | 0.49 | 0.49 | 0.62 | 0.93 | 0.67 |
| Heap | 0.23 | 0.28 | 0.34 | 0.83 | 1.00 | 0.83 |
| DAWG | 0.18 | 0.20 | 0.17 | 0.00 | 0.00 | 0.01 |
| Geom Graph | 0.07 | 0.03 | 0.03 | 0.68 | 0.76 | 0.90 |
| Graph-Natural | 0.00 | 0.00 | 0.00 | 0.82 | 0.82 | 1.00 |

contributes to their low performance. We investigated this further and presented a new table showing (i) task accuracy when code is written (Code Accuracy) and (ii) the code-writing frequency of Gemini models (Code Frequency). We observe that Gemini models also perform on par when code is written.

**Models cannot reason over the code they write.** Performance in *CodeMaybe* and *CodeEnforce* is on par with our original setup, indicating that writing code does not improve reasoning when models must internally simulate it. This reinforces our central claim: LLMs still struggle with structural reasoning, even when guided by their own code.

**Code helps when tasks align with memorized patterns.** In *CodeOnly*, models perform well on GEOM GRAPH (k-dimensional graphs embedded in geometric space, which is a standard data structure widely used in computer graphics), whose code implementation is more available online. However, they struggle with the less familiar DAWG" (directed acyclic word graph), where we define it with customized constraints to enforce a unique output. This suggests that performance may reflect memorization rather than true reasoning.

**CodeOnly falters on natural language task variants.** In GRAPH-NATURAL, models rely on brittle pattern matching (e.g., mapping "A space tunnel links planet1 and planet2" to "G.add_edge(planet1, planet2)"), but often fail to cover all phrasing variations or misinterpret descriptions (e.g., missing "Couriers frequently travel the tunnel that connects planet1 to planet2"). This highlights a key limitation that models still struggle to apply structural reasoning to ambiguous natural language scenarios, even with external tools.

**Ablation with ChatGPT web UI.** Our benchmark was evaluated via APIs. In contrast, the ChatGPT web UI has a built-in code interpreter that executes code automatically when needed. We tested o4-mini on ChatGPT with DAWG+*CodeMaybe*. The model scored 0.70, generating and executing code in 20 out of 30 cases. We observed that the model typically uses code for complex algorithmic components, while relying on natural language reasoning for the rest.

A.10  LEVENSHTEIN DISTANCE: AN AUXILIARY METRIC FOR DSR-Bench

In addition to the binary (0/1) accuracy reported in the main text, DSR-Bench includes an optional evaluation metric based on *Levenshtein distance*, which measures the minimum number of single-character insertions, deletions, or substitutions needed to transform one string into another. As a

Table 30: Individual scores with code generation by Gemini-2.5-Pro.

| Structure | None | CodeMaybe | CodeEnforce | CodeOnly | Accuracy | Frequency |
|---|---|---|---|---|---|---|
| Array | 1.00 | 1.00 | 1.00 | 0.83 | 1.00 | 0.83 |
| Queue | 1.00 | 1.00 | 1.00 | 0.99 | 1.00 | 0.99 |
| Hashmap | 0.58 | 0.17 | 0.13 | 0.57 | 0.95 | 0.60 |
| Heap | 0.36 | 0.37 | 0.50 | 0.37 | 0.42 | 0.87 |
| DAWG | 0.61 | 0.47 | 0.45 | 0.33 | 0.90 | 0.37 |
| Geom Graph | 0.19 | 0.77 | 0.53 | 0.64 | 0.98 | 0.65 |
| Graph-Natural | 0.12 | 0.10 | 0.09 | 0.24 | 0.56 | 0.43 |

Table 31: Individual scores with code generation by Claude-3.5-Sonnet and Claude-3.7-Sonnet.

| Structure | Claude-3.5-Sonnet | | | | Claude-3.7-Sonnet | | | |
|---|---|---|---|---|---|---|---|---|
| | None | CodeMaybe | CodeEnforce | CodeOnly | None | CodeMaybe | CodeEnforce | CodeOnly |
| Array | 1.00 | 1.00 | 1.00 | 1.00 | 1.00 | 1.00 | 1.00 | 1.00 |
| Queue | 0.87 | 0.86 | 0.84 | 1.00 | 1.00 | 1.00 | 1.00 | 1.00 |
| Hashmap | 0.37 | 0.27 | 0.27 | 1.00 | 0.71 | 0.73 | 0.65 | 1.00 |
| Heap | 0.53 | 0.63 | 0.63 | 0.93 | 0.89 | 0.85 | 0.82 | 0.98 |
| DAWG | 0.20 | 0.17 | 0.13 | 0.20 | 0.17 | 0.18 | 0.15 | 0.89 |
| Geom Graph | 0.10 | 0.09 | 0.10 | 0.98 | 0.04 | 0.23 | 0.20 | 0.96 |
| Graph-Natural | 0.00 | 0.00 | 0.00 | 0.10 | 0.01 | 0.02 | 0.01 | 0.23 |

continuous metric, it captures degrees of error that binary accuracy flattens. For instance, given the correct output [1,3,6], the prediction [3,1,6] is clearly closer than [0,0,0], and Levenshtein distance reflects that nuance.

However, this granularity can also blur important semantic distinctions. A syntactically well-formed but semantically incorrect output may still receive a high score, especially when the expected answer is long or formatted. When averaged over 30 test cases, models with large gaps in binary accuracy can appear deceptively similar under Levenshtein. For example, the SKIP LIST and DSU compound tasks yield the same Levenshtein score (0.75), despite the former achieving more than triple the binary accuracy (Table 32).

Output length further complicates cross-task comparison. Tasks with short, single-token outputs (e.g., BINARY SEARCH TREE depth) tend to show similar binary and Levenshtein scores (e.g., 0.66), while longer, multi-token outputs (e.g., GRAPH BFS) inflate Levenshtein scores (0.82) even when binary accuracy remains low (0.31). Relying on Levenshtein distance alone may thus give a misleading impression—for example, that the model performs well on BFS but poorly on tree depth—when binary accuracy indicates the opposite.

Table 32: Mean (± std) binary accuracy vs. Levenshtein distance scores across tasks using GPT-4.1.

| Data structure | Operation | Binary | Levenshtein |
|---|---|---|---|
| Array | Access | 1.00 (0.00) | 1.00 (0.00) |
| Queue | Compound | 0.82 (0.04) | 0.96 (0.01) |
| Stack | Compound | 0.97 (0.00) | 0.99 (0.00) |
| LRU Cache | Cache | 0.94 (0.02) | 1.00 (0.00) |
| Priority Queue | Compound | 0.63 (0.03) | 0.89 (0.01) |
| Hashmap | Compound | 0.19 (0.07) | 0.75 (0.02) |
| Trie | Compound | 0.39 (0.07) | 0.90 (0.02) |
| Suffix Tree | Construct | 0.00 (0.00) | 0.49 (0.01) |
| Skip List | Compound | 0.21 (0.02) | 0.75 (0.01) |
| BST | Insert | 0.79 (0.04) | 0.97 (0.01) |
|  | Remove | 0.78 (0.04) | 0.95 (0.01) |
|  | Post-Order Traversal | 0.82 (0.02) | 0.94 (0.01) |
|  | Depth | 0.66 (0.05) | 0.66 (0.05) |
|  | Compound | 0.69 (0.02) | 0.88 (0.01) |
| Heap | Compound | 0.58 (0.02) | 0.87 (0.01) |
|  | Heapify | 0.57 (0.03) | 0.94 (0.01) |
| RB Tree | Construct | 0.12 (0.02) | 0.87 (0.00) |
|  | Compound | 0.31 (0.04) | 0.88 (0.02) |
| B+ Tree | Compound | 0.27 (0.00) | 0.79 (0.00) |
| Graph | Breadth-First Traversal | 0.31 (0.05) | 0.82 (0.01) |
|  | Depth-First Traversal | 0.50 (0.03) | 0.85 (0.01) |
| DSU | Compound | 0.06 (0.02) | 0.75 (0.01) |
| Bloom Filter | Compound | 0.10 (0.00) | 0.84 (0.01) |
| DAWG | Compound | 0.16 (0.02) | 0.74 (0.02) |

For these reasons, we report all results using binary accuracy and relegate Levenshtein evaluation to the toolkit. The implementation remains publicly available, as the metric can still offer a useful secondary perspective, particularly when comparing models on the same task and output length.

### A.11 FAILURE RATES OF JSON PARSING VIA STRUCTURED OUTPUT

Table 33: Failure rates of JSON parsing across models and data structures.

| Model | Array | Priority Queue | Hashmap | RB Tree | Geom Graph | Bloom Filter |
|---|---|---|---|---|---|---|
| Llama3.3 | 0/30 | 0/30 | 0/30 | 0/30 | 0/30 | 0/30 |
| GPT-4.1 | 0/30 | 0/30 | 0/30 | 0/30 | 0/30 | 0/30 |
| DeepSeek-Chat | 0/30 | 0/30 | 0/30 | 0/30 | 0/30 | 0/30 |
| DeepSeek-R1 | 0/30 | 0/30 | 0/30 | 0/30 | 0/30 | 0/30 |
| o4-mini | 0/30 | 0/30 | 0/30 | 0/30 | 0/30 | 0/30 |
| Claude-3.5-Sonnet | 0/30 | 0/30 | 0/30 | 0/30 | 0/30 | 0/30 |
| Claude-3.7-Sonnet | 0/30 | 0/30 | 0/30 | 0/30 | 0/30 | 0/30 |
| Gemini-2.0-Flash | 0/30 | 0/30 | 0/30 | 0/30 | 0/30 | 0/30 |
| Gemini-2.5-Pro | 0/30 | 0/30 | 0/30 | 0/30 | 0/30 | 0/30 |

### A.12 ABLATION ON PARAPHRASED PROMPT TEMPLATES

In the `natural` suite, we use 5 paraphrased templates for problem statements and 30 for operations (e.g., "A space tunnel connects planet1 and planet2") to smooth out ambiguity in natural language descriptions. However, in the `main` and `challenge` suites, we use formalized prompt templates to enable large-scale evaluation. We conducted an ablation to test how prompt template variations affect performance. Prompting GPT-4o with "Paraphrase the following description," we generated five paraphrases and found that overall performance trends remained consistent.

Table 34: Performance on default and paraphrased prompt templates.

| Task | o4-mini | | GPT-4.1 | |
|---|---|---|---|---|
| | Default | Paraphrased | Default | Paraphrased |
| Priority Queue | 0.99 | 0.84 | 0.39 | 0.36 |
| Trie Tree | 0.89 | 0.90 | 0.63 | 0.61 |
| Geom Graph | 0.37 | 0.37 | 0.20 | 0.14 |

### A.13 PRELIMINARY STUDY ON STRATEGY ADAPTATION WITH DSR-BENCH

In Section 5, we note a future direction is to study whether models can select appropriate data structures given task requirements and switch dynamically. DSR-Bench already implicitly evaluates this adaptivity by not predefining strategies in prompts, requiring models to adjust their reasoning to the task. This emerges in (i) contrasting but related structure pairs (e.g., BFS/DFS, queue/priority queue), (ii) length generalization that demands adaptive complexity management, (iii) distribution shifts (Section 4.2) requiring robustness across input patterns, and (iv) natural language scenarios that demand transferring strategies across contexts (Section 4.3).

Beyond implicit evaluation, DSR-Bench's modular design—data generation, prompt templates, automated evaluation, and schema-based verification—readily supports dynamic strategy switching. A preliminary study is shown in Table 36 and Table 35. We mask structure names or add constraints to DSR-Bench's prompts, and test whether models can, (i) select BFS vs DFS for graph tasks, or (ii) choose between K-D tree and array for nearest-neighbor queries under varying complexities.

This extension underscores DSR-Bench's broader value: it serves as a fundamental, extensible framework for probing not only structural reasoning but also more advanced reasoning abilities.

Table 35: Dynamic strategy switching on graph algorithms: selecting the most appropriate traversal method. Results show number of correct selections out of 10.

| Task | GPT-4.1 | o4-mini |
|---|---|---|
| Shortest Path | 9.5/10 (BFS) | 10/10 (BFS) |
| Cycle Detection | 10/10 (DFS) | 10/10 (DFS) |
| Connectivity Detection | 10/10 (Both equal; 8 BFS, 2 DFS) | 10/10 (Both equal; 6 BFS, 4 DFS) |

Table 36: Dynamic strategy switching on nearest-neighbor queries: selecting the most appropriate data structure under different sizes and query numbers.

| Task | GPT-4.1 | o4-mini |
|---|---|---|
| 1 query, n=20–30 | 9/10 (array) | 10/10 (array) |
| 1 query, n=40–60 | 6/10 (array) | 10/10 (array) |
| n queries, n=5–10 | 10/10 (array) | 10/10 (array) |
| n queries, n=20–30 | 10/10 (kd-tree) | 10/10 (kd-tree) |

### A.14 EXAMPLE PROMPTS FOR EACH DATA STRUCTURE

> **Array.**
>
> An array is a list of elements, each indexed by a number starting from 0. The elements can be accessed using their index. Given an array [65, 65, 64, 8, 0]. Q: What is the value stored in index 2?

## Stack.

A stack is a data structure in which items are added and removed at the same end, maintaining a first-in, last-out (FILO) order. You should create a stack. There are two types of operations: 1. (push, k) appends an element k to the stack as the last element. 2. (pop) removes the last element in the stack. You are given an empty stack initially. Q: What is the state of the stack after the following operations: (push, 94) (push, 97) (pop) ...

## Queue.

A queue is a data structure in which items are added at one end and removed from the other, maintaining a first-in, first-out (FIFO) order. You should create a queue. There are two types of operations: 1. (enqueue, k) means an element k is appended to the queue as the last element. 2. (dequeue) means the first element of the queue is deleted. You are given an empty queue initially. Q: What is the final queue, when performing the following operations: (enqueue, 82) (enqueue, 90) (deuque) ...

## LRU Cache.

An LRU (Least Recently Used) cache is a fix-sized array-based data structure. It supports a single operation: (access, p) where 'p' is a page number. When a page is accessed, if it is already in the cache, it is moved to the most recently used position. If the page is not in the cache, and the cache is not full, the page is added. If the cache is full and the page is not present, the least recently used page is evicted to insert the new page. You should create a LRU cache with cache size (max number of different pages stored in cache) 2. Q: Initially, the cache is empty. What is the state of the LRU cache after the following sequence of operations: (access, 4) (access, 3) ...

## Priority Queue.

A max priority queue stores items each as a (value, priority) pair, where items are served in order of highest priority. It is implemented as a Fibonacci heap (a collection of heap-ordered trees linked in circular, doubly-linked lists, with lazy consolidation). It has four kinds of operations: 1. insert(value, priority): a. Create a new singleton node with the given value and priority and add it to the root list. b. Update the pointer to the maximum root if needed. 2. delete: remove and return the element with the highest priority (extract_max): a. Remove the max root from the root list. b. Add each of its children to the root list, clearing their parent pointers. c. Consolidate the root list by linking roots of equal degree until all roots have distinct degrees. d. Update the pointer to the new maximum root. 3. raise_key(value, new_priority): a. Locate the node and increase its priority to new_priority (>= current priority). b. If it now violates the heap property with its parent, cut it and add it to the root list, performing cascading cuts on marked parents. c. Update the pointer to the maximum root if needed. 4. decrease_key(value, new_priority): a. Locate the node and decrease its priority to new_priority (<= current priority). b. If any child's priority now exceeds the node's, cut those children and add them to the root list, clearing their parent pointers. After all operations, output the list of remaining (value, priority) pairs sorted by descending priority, breaking ties by insertion time (earlier inserts first). You are given an empty priority queue initially. The final state is the list of (value, priority) pairs as produced by a level-order traversal of the Fibonacci-heap forest, where you visit all roots first, then all their children, then all grandchildren, and so on; within each level, nodes are listed in descending priority order (breaking ties by larger value first). Q: What is the state of the priority queue after the following operations: (insert, 33, 72) (delete) ...

## Hashmap.

A hashmap is a dictionary that stores key-value pairs. It has two operations: (add, (key, value)) and (delete, key). 1. (add, (key, value)) adds a (key, value) pair to the hashmap by a) mapping key to a bucket using hash function, and b) put (key, value) pair in the bucket. If key exists in the bucket already, update its value. 2. (remove, key) removes the (key, value) pair by a) mapping key to the bucket using hash function, and b) find (key, value) pair in the bucket and remove it. Q: You are given an empty hashmap with 10 buckets initially. The hash function you will use is bucket = key (add, (31, 37)) (remove, 31) ...

## Trie.

A trie tree is a data structure that stores strings by sharing common prefixes with the following properties: Each node represents a character. The path from the root to a node spells out a prefix. It has two operations: 1. (insert, word) which inserts a word by a) Starting from the root node. b) For each character in the word, check if it exists in the current node's children. c) If it exists, move to that child node. d) If it doesn't exist, create a new child node for that character and move to it. e) Repeat until all characters in the word are processed. 2. (delete, word) which deletes a word by a) Traverse down the trie tree following the word's characters. b) Once the leaf node is reached, delete from bottom-up if a node has no children or is not part of another word. After all operations, the final state of the trie is represented as the pre-order traversal of the trie tree (i.e. a list of characters), where children are visited in sorted order of their characters. The root node is represented as an empty string. Q: You are given an empty trie tree, what is its final state after the following operations? insert hnftmtmisujb insert hnftmeutnb delete hnftmtmisujb ...

## Suffix Tree.

A suffix tree is a data structure that compactly represents all suffixes of a given string. It has the following properties: Given a string s of length n, there are n suffixes. Each path from the root to a leaf spells out a suffix of s. Edges are labeled with substrings. Internal nodes represent shared prefixes among suffixes. Leaves are labeled with the starting index of the suffix. The required output is the pre-order traversal of this suffix tree, collecting the edge labels encountered along the way. When performing the traversal, ensure that at each node the child edges are visited in lexicographical order, with the '$' edge prioritized to be visited before any other character. The final output should be a list that represents the flattened sequence of edge labels. Q: Given a word umcecatx$, what does its suffix tree look like?

## Skip List.

A skip list is a probabilistic data structure with multiple levels. The bottom layer is a standard sorted linked list. Each higher layer skips over more elements, allowing fast traversal. It has two operations: , and (delete, value) which deletes the value. 1. (insert, value) which inserts a value by a) Perform a search to find the position where the new value should go. b) Insert the new value to the bottom layer. c) Generate a random probability to decide whether to promote the node to the next level. d) Repeat step c until you stop or reach the maximum level. 2. (delete, value) which deletes the value by a) Search for the node at the top-most level. b) At each level where the node exists, remove the pointer to that node. c) Continue moving downward and remove the node at all lower levels. For each insert operation, use the level generation probabilities provided. If a probability is below 0.5 and the maximum level has not been reached, the node is promoted to the next level. The final state of the skip list should be represented as a list of lists, where each inner list corresponds to one level (from the highest level to level 0). Empty levels should not be included in the final output. Q: You are given an empty skip list with max level 3, what is the final state of the skip list after the following operations? insert 77 Level generation probabilities: 0.2391, 0.3462, 0.3111, 0.5204 insert 1 ...

## BST.

A Binary Search Tree is a hierarchical data structure in which each node holds a key (and optionally associated data) and has at most two children, conventionally called left and right. What makes it a "search" tree is its ordering rule: every key in a node's left subtree is strictly less than the node's key, and every key in its right subtree is strictly greater. This invariant recurses down the tree, so starting from the root you can locate, insert, or delete a key by repeatedly comparing and following the appropriate child link—just like playing a deterministic game of "higher or lower. The depth of a binary search tree is the number of nodes on the longest path from the root to a leaf node. You should create a binary search tree. Q: The root node is Node 13. Node 13's left child is Node 4, and its right child is Node 58. ...

## Heap.

A min-heap is a binary tree-based data structure that satisfies the heap property: every parent node is less than or equal to its children. It is implemented as an array-based binary heap. It has two operations: 1. (insert, k) appends an element k to the heap by: a. Adding the element to the end of the array. b. Swapping it with its parent while it is smaller, until it is in the correct position or becomes the root. 2. (delete) removes the root element: a. Replace the root with the last element in the array. b. Swap it with its smaller child while it is larger, preferring the left child in case of a tie, until it is in the correct position or becomes a leaf. You are given an empty heap initially. You should use a min-heap with array-based implementation. If a node has two children, the left child must be smaller than (or equal to) the right child. Q: What is the state of the heap after the following operations: (insert, 90) (delete) ...

## RB Tree.

A red-black tree is a self-balancing binary search tree. It has the following properties: 1. Each node is either red or black. 2. The root is always black. 3. All leaves (NIL nodes) are black. 4. Red nodes cannot have red children. 5. Every path from a node to its descendant NIL nodes has the same number of black nodes. Inserting a value into a red-black tree involves the following steps: a) Insert the value as you would in a regular binary search tree. b) Color the new node red. c) Fix any violations of the red-black tree properties. Suppose you have an empty red-black tree. Construct a red-black tree by inserting the following values in order: [21,66,36,1,85,42,77,54,46] The final state of the red-black tree should be a list of (value, color) pairs from pre-order traversal of the tree, where color is either 0 (if 'r'), 1 (if 'b'). Q: What is the final state of the red-black tree after construction?

## B+ Tree.

In a B+ tree, internal nodes store only keys for routing while all actual data is stored in the leaf nodes. There are two operations: (insert, key) to insert a key and (delete, key) to delete a key. When inserting a key into a leaf, add the key and sort the keys in ascending order. If the number of keys in a leaf reaches the specified order, split the leaf at its midpoint: the left leaf retains the lower half of keys, and a new right leaf is created with the upper half. The smallest key from the new right leaf is promoted to the parent node. For internal nodes, if an insertion causes the number of keys to reach the order, split the node at the midpoint, partition its keys and children into two nodes, and promote the median key to the parent. Deletion simply removes a key from the appropriate leaf without rebalancing. Q: Given an empty B+ tree with order 4 and the following sequence of operations, what is the final state of the tree as a pre-order traversal (a list of nodes' keys), ensuring that keys in each node are sorted in ascending order? insert 11 insert 12 ...

## K-D Tree.

A k-dimensional tree (KD-tree) is a binary space-partitioning data structure for organizing points in $R^k$. Starting with the entire point set, it recursively divides space by hyperplanes that are perpendicular to one of the coordinate axes: at each node a splitting dimension is chosen by cycling through the k axes, a split value is selected at the median along that dimension, and the node's two children hold the points on either side of that hyperplane. Suppose you have an empty KD tree. You should construct a KD tree with a set of given points by splitting the x-axis (1st axis) first, then the y_axis (2nd axis), then the 3rd, then 4th, ... And loop back to x-axis if you splitted with the last axis possible. For median of even numbers, always Use the latter one in the middle as the median (i.e. median of [1, 2] is 2, and median of [8, 6, 3, 4] is 6).Construct a KD-tree with the following points: [[47], [64], [67], [67], [9], [83], [21], [36], [87]]
If there's ever ties when sorting an axis, such as [56, 32] and [56, 12] when sortinhg by x-axis, please keep the original order in the given data.After that, please answer the following question:
Q: What is the pre-order traversal of the tree? Output in a nested list like the input.

## K-D Heap.

A kd-heap is a heap-based data structure used to prioritize exploration of regions or nodes in k-dimensional space. It is implemented as a binary min-heap, where each element is a tuple (priority, label), and the heap is ordered by priority in k dimension. and the heap is ordered by priority, where priority values are calculated using the euclidean distance from the k-dimensional priority to the origin. It has two operations: 1. (insert, k) appends an element k to the heap by: a. Adding the element to the end of the array. b. Swapping it with its parent while it is smaller, until it is in the correct position or becomes the root. 2. (delete) removes the root element: a. Replace the root with the last element in the array. b. Swap it with its smaller child while it is larger, preferring the left child in case of a tie, until it is in the correct position or becomes a leaf. You are given an empty heap initially. You should use a min-heap with array-based implementation. If a node has two children, the left child must be smaller than (or equal to) the right child. The answer should be a flat list of integers, which are the labels of the nodes in the heap. The answer should be in the same order as the nodes appear in the heap. Do not include the priorities in the label Q: What is the state of the heap after the following operations? (insert, ([48], 61)) (delete) ...

## Graph.

A graph consists of some nodes and edges. Each edge connects two nodes. Breadth-first search traverses the graph from a source node, and explores all neighbors of a node before moving to the level, visiting all nodes in increasing order of their distance from the source. You should perform breadth-first search on a graph. If there are multiple neighbors to explore for a given node, prioritize the neighbors with the smallest value. Q: The graph consists of nodes [80, 81, 64, 40, 56, 24, 36, 53], and edges [(80, 53), (81, 64), (81, 40), (81, 36), (81, 53), (64, 24), (64, 53), (40, 53), (56, 36), (24, 36), (36, 53)]. What is the breadth-first search path starting from node 64?

### DSU.

A Disjoint-Set Union (DSU) maintains a partition of elements into disjoint sets. Each set is a tree, leading to a forest of trees. It supports two operations: - find(x): Return the root of the set containing x by following parent pointers (with path compression). - union(x, y): Merge the set containing x and the set containg y. The union function is done by finding the roots of the set containing x and the set containing y. If the roots are different, we merge the two sets by attaching the root with higher rank as the parent of the root with lower rank. If the ranks are equal, we can pick the root of the set containing x as the parent and increment its rank by 1. Here, rank is a heuristic upper bound on the height of v's tree, which increases when two equal-rank trees merge. Initially, each element x in the input list is a set. It is its own parent, and its rank is 0. When asked for the final state of the DSU, return a list of find(x) for each x in the initial list, in their original order. Q: The initial list of elements is: [77, 91, 17, 83, 20, 88, 94, 99, 78, 43, 38, 73, 6]. What is the final state of the DSU after the following union operations? union(20, 38) union(38, 99)

### Geom Graph.

A random geometric graph consists of nodes and edges. Each edge connects two nodes. To create a random geometric graph from input points, we calculate the euclidean distance between each pair of points. If the distance is less than a given threshold, we add an edge between those two points, and assign edge weight of that edge equal to the euclidean distance between the its nodes. Breadth-first search traverses the graph from a source node, and explores all neighbors of a node before moving to the level. You should create a random geometric graph given the following data points: [[63.7], [26.98], [4.1], [1.65], [81.33], [91.28], [60.66], [72.95], [54.36]] The threshold for creating an edge is 30. After the graph is created, perform a breath-first-search starting from node [63.7]. Q: What is the final states of the graph? Output the breath-first-search of nodes represented by their original coordinate.

### Bloom Filter.

A Counting Bloom Filter is a probabilistic data structure used for set membership queries, with the added ability to delete elements. It maintains an array of counters of size m and uses k independent hash functions. It supports two operations: 1. (insert v): To insert an element v, each hash function determines a position in the count array and increments the counter at that position; 2. (delete v): To delete an element v, each corresponding counter is decremented (ensuring that counters never drop below zero). You should use a custom hash function described as: a) Convert the input item to a string. b) Initialize a hash accumulator to 0. c) For each character in the string, update the accumulator as: h = h * 131 + ord(character). d) Add a given salt value to the accumulator. e) Finally, compute the result as h modulo m (the size of the count array). For each item and a salt with value i (defined as the i-th hash function, where i=0, ..., (k-1)), this computation deterministically produces an index in the range [0, m-1]. Q: You are given an empty Counting Bloom Filter with m = 20 and k = 3. What is the final state of the Counting Bloom Filter, represented as its count array (a list of integer counts), after the following operations? insert 7 insert 87 ...

**DAWG.**

A Directed Acyclic Word Graph (DAWG) encodes a set of lowercase words (a-z) as a compressed trie in a directed acyclic graph. Each node has an is_end flag ('T' for true, and 'F' for false) indicating whether the path from the root to that node spells a complete word. Each edge carries a single-character label, extending prefixes by one letter. Starting from an empty DAWG, apply a sequence of operations of two types: 1. insert(word): a) Begin at the root node. b) For each character c in word: - If no c-labeled edge exists, create a new child node (is_end='F') and attach it. - Move along the c-edge to that child node. c) After the final character, set is_end='T' on the current node to mark a complete word. 2. delete(word): a) Begin at the root and follow each character's edge to the terminal node of word. b) Set is_end='F' on that terminal node so it's no longer recognized as a word. c) As you backtrack toward the root, at each node: - If the node has no children and is_end is 'F', remove it from its parent's children. After all operations, you should minimize the DAWG by merging identical suffix-subtrees: a) Recursively process every node from the leaves up. b) At each node, compute a signature: (node.is_end, sorted list of (char, child.signature) for each child) c) Use a registry mapping signatures to nodes: - If a node's signature exists, replace it with the registered node. - Otherwise, register this node under its signature. d) After this pass, all identical suffix-subtrees share a single node, yielding the minimal DAWG. To export the final DAWG, perform a breadth-first traversal from the root and record each node as (prefix, is_end), where prefix is the string formed by following edges from the root. The prefix for the root node is an empty string. Outgoing edges are visited in ascending lexicographical order. Q: You are given an empty Directed Acyclic Word Graph (DAWG), what is its final state after the following operations? insert ash insert asdtov insert isy ...

### A.15 THE USE OF LARGE LANGUAGE MODELS (LLMS)

We have used LLMs as a general-purpose assistive tool for grammar correction, writing polish, and minor code suggestions only.

