# OpenReview forum: "Can LLMs Reason Structurally? An Evaluation via the Lens of Data Structures"
_ICLR.cc/2026/Conference — Submitted to ICLR 2026_

### Official Review · Reviewer_aaAi · 2025-10-25

**Soundness:** 3
**Presentation:** 3
**Contribution:** 3
**Rating:** 4
**Confidence:** 4

**Summary:**

This paper, Can LLMs Reason Structurally? An Evaluation via the Lens of Data Structures, introduces DSR-Bench, the first systematic benchmark to evaluate large language models’ (LLMs) structural reasoning—their ability to understand and manipulate relationships like order, hierarchy, and connectivity through canonical data structures. Covering 20 data structures, 35 operations, and 4,140 problem instances, DSR-Bench provides interpretable, contamination-resistant, and fully automated evaluation. The benchmark spans five suites (main, challenge, spatial, natural, and code) to test generalization across abstract, multimodal, and language-grounded settings. Results on ten SOTA models, including GPT-5, Claude 3.7, Gemini 2.5 Pro, DeepSeek-R1, and Llama 3.3, reveal substantial weaknesses—top scores plateau around 0.498 on complex tasks. The findings expose persistent failures in multi-attribute, multi-hop, spatial, and language-based structural reasoning, even for frontier reasoning models

**Strengths:**

The paper offers a comprehensive, principled diagnostic framework bridging algorithmic and structural reasoning, which is both interpretable and reproducible. It evaluates across multiple reasoning modalities, from formal operations to natural language and code-based tasks, achieving impressive breadth and methodological rigor. The authors benchmark both closed- and open-source LLMs with rich ablations (prompting strategies, spatial dimensionality, data distributions), offering deep insights into model limitations and prompting effects. The hierarchical design allows fine-grained localization of reasoning failures, making DSR-Bench a valuable tool for community-wide evaluation and progress tracking.

**Weaknesses:**

Despite strong contributions, the benchmark’s model coverage remains limited—only ten models are tested, lacking more varied model sizes and major closed-source reasoning systems, which constrains external validity. The evaluation focuses heavily on synthetic, symbolic reasoning and may underrepresent noisy or multimodal real-world inputs. Moreover, while the work claims to reveal algorithmic limitations, the analysis is largely empirical, without deep mechanistic investigation into why reasoning breaks down. The code suite section exposes that models fail to reason over their own code but does not propose mitigation pathways. Finally, performance saturation at ~0.5 suggests the tasks might not scale linearly in difficulty, risking benchmark ceiling/floor effects.

**Questions:**

1. How would results change if more diverse model scales (e.g., 7B–400B) and closed-source reasoning models (e.g., Gemini Ultra, Claude 3 Opus) were included?

2. Can the authors analyze internal reasoning traces or attention mechanisms to identify why models fail specific structural operations (e.g., multi-hop tree balancing)?

3. Have the authors considered multimodal structural reasoning (e.g., visual graphs, tables) as a next step to bridge symbolic and embodied reasoning?

4. How sensitive is DSR-Bench performance to prompt phrasing and temperature—does structural reasoning remain consistent under naturalistic, noisy instructions?

5. Could future work explore training-time regularizers or inductive biases explicitly targeting structural reasoning, informed by DSR-Bench diagnostics?

---

> ### Author Response · Authors · 2025-11-21
>
> We thank the reviewer for their feedback. We are pleased that they view our framework as “principled” and “comprehensive”, with strong interpretability and reproducibility. We appreciate the recognition of the breadth and rigor of our evaluations, benchmarking of both closed- and open-source LLMs under rich ablations. The reviewer also highlights our hierarchical design for precisely diagnosing reasoning failures and considers DSR-Bench a valuable tool for the community.
>
> We clarify some misunderstandings and address the reviewer’s concern below.
>
> ### W1: Model coverage
> - Lacking `“varied model sizes”`: While models of different sizes are often not available for closed-source models, our benchmark still spans multiple scales: GPT-5, o4-mini, Llama-3.3-70b.
> - Lacking `“major closed-source reasoning models”`: Our evaluation already includes 7 frontier closed-source models (4 of which are reasoning models, including GPT-5-2025-08-07 and Gemini-2.5-Pro-2025-06-17) and 3 open-source models.
>
> ### W2: Noisy or multimodal real-world inputs
> - `“heavily on synthetic, symbolic reasoning”`: As we explain in lines 98-99 & lines 243-247, the synthetic nature of DSR-Bench is a strength: it minimizes the risk of contamination and provides a controlled, deterministic testbed to evaluate reasoning skills.
> - `“underrepresent noisy, multimodal real-world inputs”`: DSR-Bench already includes a spatial suite for high-dimensional structured data and a natural suite for natural-language, noisy descriptions. While the current scope focuses on text-based reasoning, we invite future work to extend the benchmark to multimodal settings (e.g., images).
>
> ### W3: Mechanistic investigation
> - `“without deep mechanistic investigation into why reasoning breaks down”`: We emphasize that this is a “datasets and benchmarks” paper. Our focus is to introduce the dataset, benchmark models on it, and highlight key observations to showcase its value. Therefore, mechanistic interpretability and related techniques, while useful, fall outside the scope.
> - Moreover, mechanistic interpretation is not feasible on the large, closed-source models we benchmark. We only have API access and cannot inspect activations or model weights. This constraint is precisely what underscores the importance of our benchmark: DSR-Bench provides a structured, controlled environment for probing black-box models, enabling systematic evaluation and concrete hypotheses about their structural reasoning abilities without needing internal access.
>
> ### W4: Mitigation pathways to improve reasoning
> - `“does not propose mitigation pathways”`: Again, because this paper is in the “datasets and benchmarks” track, methodological innovation falls outside our scope. However, as the reviewer notes, DSR-Bench reveals critical limitations of current models through diagnostic, fine-grained analysis. This is precisely what enables future work to develop targeted improvements.
>
> ### W5: Task difficulty
> - `“performance saturation at ~0.5 suggests the tasks might not scale linearly in difficulty”`: As shown in Table 1, model performance varies substantially across models and tasks. A detailed task-level breakdown for each model is provided in Appendices 4 and 5. The score of 0.5 is the average achieved by the best-performing model, indicating substantial room for improvement in future model development.
>
> > Q1. More diverse model scales and closed-source reasoning models
>
> This question is addressed in W1. We cover closed-source reasoning models that are more recent than Gemini Ultra and Claude 3 Opus.
>
> > Q2. Internal reasoning traces or attention mechanism
>
> This question is addressed in W3. Based on our benchmarking results, we provide extensive quantitative and qualitative analysis, including manual inspection of reasoning traces, and highlight key observations in Sections 4.1–4.4.
>
> > Q3. Multimodal structural reasoning
>
> This question is addressed in W2. We welcome future work extending DSR-Bench, which is currently text-based, to multimodal reasoning. Thank you for the suggestion.
>
> > Q4. Prompt phrasing and naturalistic, noisy instructions
>
> We already include an ablation on how prompt phrasing affects performance in Appendix A.12. The natural suite in DSR-Bench is specifically designed to evaluate naturalistic, noisy instructions.
>
> > Q5. training-time regularizers or inductive biases
>
> As this is a “datasets and benchmarks” paper, we invite future research to propose methodological improvements based on the limitations revealed by DSR-Bench. Thank you for the suggestion.
>
> We hope these clarifications help answer your questions and highlight the core aim and strengths of DSR-Bench. We would be happy to respond to any additional questions. Thank you!

---

### Official Review · Reviewer_UyrF · 2025-10-28

**Soundness:** 3
**Presentation:** 3
**Contribution:** 2
**Rating:** 4
**Confidence:** 4

**Summary:**

The authors propose an evaluation suite on reasoning of structural data, DSR-bench, to benchmark and understand whether LLMs can reason about and manipulate structural relations. Experiment results show that LLMs uniformly fail on the benchmark, especially on tasks which require spatial or natural language reasoning. This means the problem needs to be better addressed to further improve LLMs' capabilities.

**Strengths:**

1. The paper is well written and very easy to understand. The presentation is clear and systematic in supporting the authors' arguments and showing experiment results.

2. The experiment is carefully designed and executed. The design of the experiments are comprehensive to provide insights on understanding LLMs' capabilities in terms of structural data.

3. The investigated question is important. LLMs need to understand structural data to better serve as code generator/interpreters.

**Weaknesses:**

1. Novelty seems to be a bit limited. As the authors discuss in section 2, there are quite some existing works on testing LLMs on algorithmic execution capabilities. The authors argue that their work is distinct in that they use a holistic approach using data structures (and presumably more comprehensive in the sense of algorithm tasks). However, it feels to me that the current benchmark looks like an ensemble of previously investigated questions, which means that the novelty is a bit limited. Moreover, previous works have also showed that LLMs are not so good in code execution, so can the authors clarify their novelty here? Also see [1, 2, 3, inter alia] for more code execution and related natural language execution tasks.

2. In Figure 1, is the illustration of DAWG the same as the the logic of the question presented to the LLMs? To my understanding the prototypical DAWG scans and store characters from left to right. It seems that the DAWG illustrated stores from right to left. Is it a typo? If the questions presented are in the same order, is it possible that LLMs did not fully understand the question as it is not the most conventional one and instructions are not clear enough?

3. Why are LLMs particularly bad on spatial data? Do multi-modal models perform better compared to traditional LLMs on it?

[1] La Malfa, E., Weinhuber, C., Torre, O., Lin, F., Marro, S., Cohn, A., ... & Wooldridge, M. (2024). Code simulation challenges for large language models. arXiv preprint arXiv:2401.09074.

[2] La Malfa, E., Weinhuber, C., Torre, O., Lin, F., Huang, X. A., Marro, S., ... & Wooldridge, M. (2025). Code Simulation as a Proxy for High-order Tasks in Large Language Models. arXiv preprint arXiv:2502.03568.

[3] Liu, C., Dylan Zhang, S., Ibrahimzada, A. R., & Jabbarvand, R. (2024). Codemind: A framework to challenge large language models for code reasoning. arXiv e-prints, arXiv-2402.

**Questions:**

The authors are welcome to address the weaknesses above.

---

> ### Author Response · Authors · 2025-11-21
> **Response (Part 1)**
>
> We thank the reviewer for their feedback. We are pleased that the reviewer found our writing and presentation to be clear, our claims to be well-supported, our experiments to be comprehensive, and our motivation to be important.
>
> We address the reviewer’s concerns below.
>
> ## W1. Novelty & existing works
>
> We respectfully disagree with the reviewer’s point about limited novelty compared to related works. We would like to elaborate on our contributions to existing works.
>
> ### Contrast with existing algorithmic reasoning works:
> - **Our benchmark is not a simple collection of existing tasks**: in the main data-structure suite, only 2 of 35 tasks overlap with prior benchmarks. In addition, the specialized suites (spatial, natural, code) that we design introduce novel test scenarios for algorithmic reasoning that were previously underexplored.
>
> - **To elaborate on the related works in Section 2** (also added to Appendix A.1 in the updated manuscript):
>   - *[Zhou et al., 2022; 2024; Lee et al., 2024]* study arithmetic tasks on small transformers (addition, subtraction, sine, square root), which target a different aspect of algorithmic reasoning.
>   - *[Wang et al., 2023a; Fatemi et al., 2024]* are graph benchmarks for LLMs (e.g., connectivity, cycle detection); our graph tasks (BFS, DFS) do not appear in their task sets and are integral parts of our hierarchical design, serving as preliminaries for harder tasks (e.g., DAWG).
>   - *The CLRS-Text Benchmark [Markeeva et al., 2024]* covers 30 classical algorithms (e.g., sorting, greedy algorithms), with only BFS and DFS overlapping with ours.  In contrast, DSR-Bench focuses on data structures as fundamental building blocks for these algorithms, enabling finer-grained diagnosis and offering more interpretable and actionable insights into reasoning failures. Our prompt design, evaluation pipeline, and specialized suites are also new and different.
>
> ### Contrast with coding evaluations:
> - **DSR-Bench is intentionally designed to evaluate general reasoning without external tools such as code execution**, as we explained in our paper (Lines 36-40 in the Introduction, Lines 138-144 in the Related Works, and Lines 439-445 in Section 4.4).  This design choice with recent evaluation trends (e.g., Google & OpenAI’s IMO assessments without Lean use). Enabling code execution with an interpreter would primarily test syntax/API usage rather than structural reasoning, and is more susceptible to contamination. Our code suite serves as an auxiliary probe, complementing our main contribution of the investigation on structural reasoning by assessing the ability of LLMs to use code as an aid for reasoning tasks (see our response to Reviewer RkGR W3).
>
> - **Contributions to the three referenced papers:** We thank the reviewer for pointing us to these references and have added all three to our Related Work section. [3] primarily evaluates code-generation and code-understanding capabilities rather than algorithmic reasoning. While [1, 2] also investigate the algorithmic reasoning capacity of LLMs through the lens of code simulation, we clarify that our work is complementary in three aspects:
>   - The DSR-Bench code suite is explicitly designed to study LLM’s ability to understand and manipulate relationships through the lens of data structure tasks.
>   - We focus on understanding when and how code helps or harms structural reasoning, rather than using code simulation as the primary lens to probe algorithmic reasoning.
>   - Our code suite also allows for controlled analysis of the impact of different code-usage regimes through varying how LLMs are allowed to use code (e.g., with or without tool execution, optional or enforced).
>
> ## W2. Typo & unconventional representation
>
> ### Typo:
> - Thank you for pointing this out! **This is indeed a typo, but in the illustrative figure only.** All DAWG tasks in DSR-Bench use the conventional left-to-right definition (if by “left-to-right” the reviewer is referring to “from starting-letter to ending-letter”). We have updated Figure 1 in the updated revision.
>
> - **We would also like to emphasize that Figure 1 is purely illustrative**: it is intended to provide a high-level overview of the structures and relationship types in DSR-Bench. None of the toy examples shown in Figure 1 (nor the figures themselves) is part of the actual benchmark - the prompts in the figure are simplifications.
>
> ### Unconventional representation
> - **We already study unconventional or customized problem framing** in the “Implicit prior may hinder instruction following’’ section (starting at line 365). There, we show that models can cling to entrenched priors (e.g., default tie-breaking rules) even when prompts explicitly specify alternative rules, highlighting precisely the kind of robustness issue the reviewer raises, and it is a critical limitation for practical use with user-defined constraints.

---

> ### Author Response · Authors · 2025-11-21
> **Response (Part 2)**
>
> ## W3. Why LLMs are bad with spatial data
>
> ### Spatial data:
>
> - **Reason for performance degradation**: Our results suggest that LLMs struggle with spatial data partly because higher-dimensional tasks require more sequential operations and longer intermediate state tracking, which compounds errors over steps. Thank you for the question.
>
> - **To better understand why performance degrades with distribution shifts on spatial tasks**, we conducted a detailed error analysis on KD-Tree tasks under uniform vs. non-uniform (circle) data. For a human reasoner, the two settings are similarly difficult: across 30 questions per group, tie-triggering duplicated indices occur at comparable rates and the total number of median operations is similar (156 vs. 145), indicating well-matched difficulty. However, models are much more error-prone on non-uniform data: they correctly identify the root node in 13/30 uniform cases but only 5/30 circle cases, violate the “latter-median” rule more frequently, and show more completely incorrect median choices, along with new axis-confusion errors that do not appear under uniform data. These patterns suggest that LLMs are particularly brittle when reasoning over “close coordinates’’ (near-ties and clusters), even though such cases are trivial for humans.
>
> ### Multimodal LLMs:
>
> - **We clarify that DSR-Bench is a purely text-based dataset**, where multimodal LLMs do not directly apply. However, we agree that testing multi-modal models is an interesting future direction. For example, rendering 2D spatial data as images and evaluating vision-language models might allow us to better evaluate visual understanding on geometric structures. However, this approach limits evaluation to 2D, whereas DSR-Bench incorporates higher-dimensional KD data. We’ve updated Section 5 (“Discussion and Conclusion”)  to include this future direction.
>
> We hope these clarifications help address your questions and further highlight the novelty and strengths of DSR-Bench. Please let us know if you have any further concerns—we are committed to responding to them thoroughly. Thank you!

---

### Official Review · Reviewer_XAHe · 2025-10-30

**Soundness:** 3
**Presentation:** 3
**Contribution:** 3
**Rating:** 4
**Confidence:** 4

**Summary:**

The paper proposes **DSR-Bench**, a synthetic benchmark for “structural reasoning” framed via classic data structures (20 structures, 35 operations) with multiple suites (main, challenge, spatial, natural, code). The evaluation is automatic via schema-checked outputs; the authors report broad gaps across modern LLMs and several prompting/tool-use variants.

**Strengths:**

1. Clear writing; figures/tables are easy to read.
2. A coherent, unified design principle with code to synthesize more tasks—useful for the community.
3. Large experimental sweep with many variants; results are carefully organized.

**Weaknesses:**

1. **Positioning/novelty**. Many insights (multi-hop difficulty, distribution shifts, NL brittleness, limited benefit from self-written code) are already familiar. The paper’s claim about “LLMs reasoning structurally” is not convincingly isolated because representation and execution are entangled throughout. ( See Weakness#2 )

2. **Representation vs. execution not disentangled.** Each task has (i) a representation component—does the model correctly parse the structure as presented?—and (ii) an execution component—can it carry out the specified operations. The study heavily emphasizes execution and under-specifies representation:

    2.1. How exactly are high-dimensional objects (2D grids, graphs, spatial point sets) encoded? JSON? code-like syntax? natural language? chain of insertions?

    2.2. Without concrete prompt schemas/examples per family, it’s hard to tell if performance drops arise from poor parsing/grounding vs. failed reasoning on operations.

3. **Execution ablations are shallow.** Now fixing the representation to 2.1 ( and particularlly for tasks like BST/Graph ), if operations are given as pseudo-code-ish directives, do results change when you switch to (a) Python-style code, (b) pure NL imperatives ( in DSR-Bench-natural you already tested when both representation and operations are switched to NL ), or (c) step-checked traces? These swaps would reveal whether failures stem from programming formality vs. actual manipulation of state.

4. **Context length realism.** For harder instances, solutions often grow quickly (exponentially). It’s unclear whether the runs hit or neared context limits and how accuracy scales with available tokens. A context-budget ablation (and confirmation that max tokens were utilized on hard tasks) is missing.

**Questions:**

1. **Priority Queue (multi-attribute) prompt.** Please include a concrete example that shows how multiple attributes are encoded in the input and in the operation spec. Right now most examples shown are basic Queue; the multi-attribute claim (e.g., around L319) is important but not inspectable.


2. **Aggregation in Table 2.** Is this an average over all DSR-Bench-main tasks? If so, how does GPT-4.1+CoT reach ~94% while GPT-5 is ~79%? And how do these aggregates reconcile with hard-structure results (e.g., DAWG ~21% in Table 23)?


3. **Encoding of high-dimensional/spatial data.** Provide exact prompt formats (few full examples) for 2D, graphs, and spatial point sets.


4. **Distribution shift & CoT (Table 6).** Is GPT-4.1 reported without CoT there? Does adding CoT improve robustness under non-uniform spatial distributions or dimensionality shifts?

---

> ### Author Response · Authors · 2025-11-21
> **Response (Part 1)**
>
> We thank the reviewer for their feedback. We are delighted that they find our paper well-written and our contribution “useful for the community” (providing a “coherent [and] unified” design). We also appreciate that they found our experiments exhaustive, large-scale, and “carefully organized”.
>
> We address the reviewer’s concerns below.
>
> ## W1. Novelty
>
> We clarify that our paper focuses on **LLMs’ algorithmic reasoning abilities**, as motivated in the Introduction. In Related Work (lines 123–129), we explain our novelty relative to existing algorithmic reasoning benchmarks and evaluation works: DSR-Bench takes a holistic and fundamental approach with data structures, which are interpretable and fine-grained. We introduce new tasks and new spatial, natural, and code suites that probe previously underexplored scenarios. Our contributions and observations are therefore novel within the context of algorithmic reasoning. A detailed discussion of each prior work is in our response to Reviewer UyrF’s W1.
>
> ## W2. Entanglement of representation vs. execution
>
> We respectfully disagree with the reviewer’s point that DSR-Bench fails to disentangle the abilities to understand the representation of a data structure and to execute the required operations.
>
> - **We clarify that our hierarchical design already allows us to probe these two aspects separately**, especially for more complex structures. For instance, for BST, we include representation-focused tasks (pre-order, in-order, post-order traversals, depth) alongside execution-focused tasks (insertion, deletion, compound). These representation-focused tasks serve as preliminaries to execution-focused tasks, allowing us to quantify how much error comes from mis-encoding the structure versus failing to carry out operations over an already-understood structure. For simpler data structures in the linear family, we additionally inspect intermediate outputs to confirm that the model has correctly parsed the representation before failing on later steps.
>
> - **For the same data structure, we evaluated across settings that demand different types of representational understanding** — formal, spatial, natural language, and code-based. This allows us to (i) attribute some failures to mismatches in representation or grounding between, for example, formal vs natural language, and (ii) attribute others to execution challenges under harder regimes (e.g., non-uniform spatial distributions)
>
> Moreover, as explained in Section 3.2, each data structure task comes with carefully designed descriptions to avoid ambiguity in representation parsing. Per the reviewer’s request, we have added example prompts for all 20 data structures in Appendix A.14, in the updated manuscript.
>
> ### W2.1. How are high-dimensional objects represented?
>
> - We think there’s a **misunderstanding**; we do not have “2D grids” in the benchmark.
>
> - Spatial data points are encoded in a simple nested-array format over its coordinates. For example, a K-D tree instance (where K=3) is presented as:
> ``
>  Construct a KD-tree with the following points: [[47, 64, 56], [67, 67, 4], [9, 83, 90], [21, 36, 32], [87, 70, 54], [88, 88, 92], [12, 58, 45], [65, 39, 32], [87, 46, 78]]
> ``
> - Graphs are encoded by explicitly listing nodes and edges, e.g.:
> ``
> The graph consists of nodes [80, 81, 64, 40, 56, 24, 36, 53], and edges [(80, 53), (81, 64), (81, 40), (81, 36), (81, 53), (64, 24), (64, 53), (40, 53), (56, 36), (24, 36), (36, 53)].
> ``
>
> ### W 2.2. Adding concrete prompt examples
>
> In the revised version, we added example prompts for all 20 data structures in Appendix A.14. This makes the exact input/output schemas and instructions fully transparent.
> We have also prepared a public HuggingFace repository containing the full dataset and prompts, which will be released upon acceptance to comply with the ICLR’s anonymity policy.

---

> ### Author Response · Authors · 2025-11-21
> **Response (Part 2)**
>
> ## W3. Ablation on execution
>
> We are not certain if we fully understand the question and would greatly appreciate any clarification from the reviewer. We believe the reviewer may be asking whether failures come from programming formality vs. state manipulation. If so, DSR-Bench has already tested this along several axes.
> - **Python-style code.** The question of whether code representations help LLMs reason about structures is directly addressed by our code suite (Section 4.4, Appendix A.9), which requires models to represent operations in code before operation execution. Comparing CodeEnforce/CodeMaybe to the default prompts, we find that accuracy is essentially unchanged (Table 8).
> - **Pure NL.** Our natural suite corresponds directly to case (b). BST/graph/queue tasks are described using pure natural-language narratives, while preserving the same synthetic instance distributions as the formal setting. We observe systematic drops there, isolating difficulties in grounding and manipulation under NL. Mixing formal representations in a real-world setting would likely lead to confusing prompts.
> - **“Step-checked traces”**: We would appreciate clarification on what exactly is meant by “step-checked traces”. We currently interpret this as intermediate steps. Our Stepwise prompting variant (Section 4.1, Appendix A.5) encourages models to produce intermediate reasoning steps in a structured format. This is effective on simpler tasks, but on challenging tasks, this yields modest gains and does not close the gap.
>
> ## W4. Context length constraints
>
> We designed our experiments so that context length does not introduce unfair comparisons across models. Several instruction-tuned models we evaluate (DeepSeek-V3, Claude-3.5-Sonnet, Gemini-2.0-Flash) only allow around 8,000 output tokens, so we set a global maximum of 8,000 output tokens for instruction-tuned models, and 15,000 (short), 30,000 (medium), and 45,000 (long) for reasoning models. For the main results in Table 1, we found that all outputs did not exceed the token budgets, so context length is not a limiting factor for performance.
>
> > Q1. Priority queue encoding
>
> Priority queues are encoded in (value, priority) pairs. Below is an example prompt snippet:
> ``
> Q: What is the state of the priority queue after the following operations:
> insert (22, 77)
> insert (34, 21)
> raise_key (34, 83)
> insert (85, 95)
> insert (40, 68)...
> ``
> We have added prompt examples for all data structures (including multi-attribute ones) in Appendix A.14 for easy inspection.
>
>
> >  Q2. Aggregation in Table 2.
>
> Yes, the results in Table 2 are averaged over all DSR-Bench-main tasks. To answer your question directly:
> - GPT-5 (0.79 in Table 1) is evaluated with its default reasoning setup (default prompt, medium thinking effort) on DSR-Bench-main.
> - GPT-4.1 + CoT (0.94 in Table 2) is an instruction-tuned model with an explicit CoT example added. Under the same default prompt setting, GPT-4.1 averages only 0.59 in Table 2, well below GPT-5’s 0.79, so GPT-4.1 is not outperforming GPT-5; CoT simply boosts it on easier tasks.
> - Regarding DAWG: the ~21% DAWG score in Table 23 is for a single hard task. Most DSR-Bench-main tasks (arrays, queues, simple trees, etc.) are near 100% with CoT, so when we average over all tasks and data structures, these easy cases dominate, and the global mean can still be 0.94 despite low accuracy on hard structures like DAWG, Bloom Filter, and K-D Heap.
>
> > Q3. Encoding of high-dimensional data
>
> Please see our response to W2.1.
>
> > Q4. Distribution shift and CoT
>
> In Table 6, GPT-4.1 is reported with Stepwise (which explicitly elicits intermediate thinking steps in Structured Output, but no examples are provided). To address the reviewer’s question and investigate whether CoT helps mitigate this distribution shift issue, we conducted a small ablation study on K-D Tree accuracy on medium-length tasks using GPT-4.1-2025-04-14 with CoT prompts.
>
> | Distribution | GPT 4.1 CoT accuracy |
> |-------------|------------------------|
> | Uniform     | 0.48 (0.02)           |
> | Blob        | 0.29 (0.02)           |
> | Circle      | 0.37 (0.12)           |
> | Moon        | 0.29 (0.02)           |
>
> Even with CoT, the decrease in accuracy on the shifted distribution remains, suggesting that the model still relies on pattern memorization even when a step-by-step example is provided.
>
> We hope these clarifications address the reviewer’s questions and further highlight the core aim and strengths of DSR-Bench. We would be happy to respond to any additional concerns or questions. Thank you!

---

> > ### Comment · Reviewer_XAHe · 2025-11-26
> >
> > Thank you for the added prompt examples; they make several points clearer. While I do appreciate these clarifications, my concern about the need to better **disentangle** and **ablate** representation vs. execution remains.
> >
> > 1. **Construction of structured objects is not well ablated.**
> >    For BST, the structure is presented via a series of natural-language construction statements (e.g., *“The root node is Node 13. Node 13’s left child is Node 4, and its right child is Node 58. …”*). For Graph, there is no construction stage: the structure is given directly as a list of nodes and edges in pseudo code. For Priority Queues, you use pseudo-code style construction, e.g. `(insert, 33, 72)`, `(delete)`. These different choices across data structures make it hard to assess how much error comes from understanding the *representation* itself, as opposed to later reasoning over it.
> >
> > 2. **Execution has a similar design space that is under-explored.**
> >    Just as there are multiple ways to present the object, there are multiple ways to present the operations. Currently, most execution prompts are in natural language. For example, in the Graph tasks you give an NL description of BFS and then ask “What is the breadth-first search path…”. It would be natural to also provide BFS in pseudo code or actual Python and then ask the model to execute that as well. One might expect a code-style representation to match better with code-style execution, and it is not clear how performance would change when these are aligned or deliberately misaligned.
> >
> > 3. **BST is set up well; other structures could benefit from a similar treatment.**
> >    In BST, you already have a good separation between representation understanding subtasks (insert, delete, compound, depth) and execution subtasks (the different traversals). For other structures such as Graph, you go straight to traversal without any explicit representation understanding subtasks. In my view, there are potentially very interesting insights to be gained if you push this further, which would also speak more directly to your Section 4.1 title, *“CAN LLMS UNDERSTAND AND MANIPULATE DATA STRUCTURES?”*
> >
> >    I am asking these questions in part because heap-related tasks are a clear failure case: from your tree-related results (including BST), it seems strong LLMs often have a reasonable understanding of tree structure, but struggle with performing swaps and maintaining invariants. This would be a nice example of “understands, but cannot manipulate” if analyzed more deeply. On the other hand, although 2D grids or images are not included here, there is existing work showing that, because LLMs are trained on sequences, presenting a 2D grid as a simple nested list like `[[...], [...]]` can already cause issues with aligning the y-axis (GPT3 had it, not sure if GPT5 overcomes it somehow). That would be a case where the difficulty is primarily in *understanding* the representation itself.
> >
> > As I mentioned in my original review, I think this paper has real potential, but it needs more careful ablation and control along these lines. I therefore maintain my rating of 4.

---

### Official Review · Reviewer_TkGR · 2025-11-01

**Soundness:** 3
**Presentation:** 3
**Contribution:** 3
**Rating:** 4
**Confidence:** 4

**Summary:**

This paper introduces a novel benchmark to evaluate LLMs’ structural reasoning capabilities through tasks grounded in data structures. The benchmark spans a broad spectrum of structures, including sequences, temporal orderings, key–value mappings, tree-like hierarchies, connectivity and group membership, as well as composite structures. The reasoning operations cover construction, inspection, manipulation, and their compositions. The authors stratify task difficulty by input length. They curate five suites, including main, challenge, natural language, code, and spatial, to assess LLMs’ structural reasoning in real-world settings. The experimental results show that current LLMs still struggle in structural reasoning.

**Strengths:**

- The problem is well-defined.
- The experimental evaluation is thorough and solid.
- The findings are informative, highlighting critical limitations of current LLMs.

**Weaknesses:**

- Lack of statistical analysis of the dataset (e.g., how many instances are used in the evaluation?).
- It would be beneficial to include mechanistic interpretability analysis. See [1] for reference, which provides deeper insights even with simpler data structures.
- Missing SFT and RL (e.g., GRPO) experiments. If models are trained on these tasks, how much does performance improve?
- Even human reasoners may struggle with parts of this benchmark, since deterministic program-based tasks are not naturally handled by probabilistic models. A potentially helpful approach is tool use with coding (enabling models to generate code and invoke external interpreters), combined with sufficient in-context knowledge (e.g., definitions and demonstrations of relevant algorithms and data structures). It is recommended to adopt more comprehensive tool-use baselines across all suites, not only the code suite.

*[1] Grokked Transformers are Implicit Reasoners: A Mechanistic Journey to the Edge of Generalization*

**Questions:**

What is the difference between “Stepwise” and “Zero-Shot CoT,” given that both prompt the model to reason step by step without demonstrations? Are these merely different prompt wordings—e.g., “include a steps field in your output” versus “let’s think step by step”—or do they reflect distinct prompting strategies?

---

> ### Author Response · Authors · 2025-11-21
>
> We thank the reviewer for their feedback and are delighted that they find our “problem well-defined”, “experimental evaluation thorough and solid”, and “the findings informative, highlighting critical limitations of current LLMs”.
>
> We take this opportunity to clarify several points and address the concerns raised.
>
> ## W1. Statistics of the dataset
>
> - As highlighted in Line 17 in the Abstract and elaborated in Lines 90-92 in the Introduction, "DSR-Bench comprises 4,140 problem instances spanning 20 data structures (grouped into six relationship categories; see Figure 1) and 35 operations across three difficulty levels (short, medium, and long) and five evaluation suites (main, challenge, spatial, natural, and code).’’ And from Line 266, “each problem evaluated three times (166,230 total evaluations)”.
>
> ## W2+3. Mechanistic interpretability, SFT, and RL
>
> - **We emphasize that this is a “datasets and benchmarks’’ paper**: our focus is to introduce the dataset, benchmark models on it, and highlight key observations to showcase its value. Therefore, these techniques, while useful, fall outside the scope.
>
> - **Moreover, mechanistic interpretability and RL are not feasible on the large, closed-source models we benchmark.** We only have API access, so we cannot inspect the model weights and activations for mechanistic interpretability methods. Currently, RL fine-tuning is only available to o4-mini through the OpenAI API, and with limited flexibility. These constraints are precisely what underscore the importance of our benchmark: DSR-Bench provides a structured, controlled environment for probing these frontier models, enabling systematic evaluation and concrete hypotheses.
>
> For future work, we agree that MI and fine-tuning studies on smaller open-source models are promising. In the updated manuscript, we have added this to Section 5 (“Discussion and Conclusion”) and cited [1] to encourage such follow-up research using DSR-Bench as a controlled testbed, highlighted in red.
>
> ## W4. Coding and tool-use
>
> - **Disabling coding and tool-use was intentional as we focus on “general reasoning”**, as explained in Lines 36-40 in the Introduction, Lines 138-144 in the Related Works, and Lines 439-445 in Section 4.4. To further elaborate on the motivation:
>
>   - *Our benchmark is designed to evaluate general reasoning in LLMs*, independent of external tools such as code interpreters or formal solvers. This type of general/pure reasoning underpins almost all problem-solving, beyond coding or domains where coding is applicable.
>
>   - *Our motivation aligns with emerging evaluation trends*, such as recent Gemini-Deep-Think and OpenAI assessments on the IMO competition, which disallowed code or tool use (e.g., Lean) to test end-to-end general reasoning.
>
>   - *We target structural reasoning*, a core aspect of general reasoning. As real-world tasks often involve handling data and relationships, data structures provide a clear and controlled lens for assessing LLMs’ practical utility.
>
>   - *Structural reasoning is no longer directly evaluated when tool-use is enabled*. The ability being tested shifts toward writing correct syntax and using function APIs, and it carries a high risk of data contamination due to the widespread availability of coding tutorials and online platforms.
>
> - **“Tool use with coding and interpreter” is already included as a baseline**.
>
>   - *The code suite serves as an auxiliary probe*: it tests whether, in a compromised setting where coding is allowed but no interpreter is available, LLMs can reason over the code they themselves write. This still demands structural reasoning—just expressed in formal syntax—and we find no improvement over natural-language reasoning.
>
>   - *We include a “CodeOnly” baseline*, where an external interpreter is available, to illustrate an upper bound for pure reasoning without tool use. However, this baseline does not target the core ability DSR-Bench aims to evaluate; it serves primarily as *a reference point*.
>
>
> > Q1: Difference between “Stepwise” and “0-CoT”
>
> They are distinct. “Stepwise” explicitly adds a “steps: list[Step]” attribute in the JSON schema of Structured Output, while “0-CoT” appends the phrase “Let’s think step by step” to the prompt. The details of all prompting strategies (with examples) are in Appendix A.3.
>
> We hope these clarifications help answer your questions and highlight the core aim and strengths of DSR-Bench. Please let us know if you have any further concerns—we are committed to responding to them thoroughly. Thank you!

---

### Meta-Review · Area_Chair_cWfe · 2026-01-06

**Summary:**

This work received unanimously negative-learning scores initially. Post-rebuttal, one reviewer maintained their score, while I would hazard that the other reviewers may have increased their scores somewhat, but overall that would still put the quantitative regard as borderline, mixed, positive leaning.

I have read in detail the authors' (private) offical comment regarding the reviews. I partially agree with some of their views, while I disagree with others. To their credit, they are cautious in their statements and do not make sweeping claims. Nonetheless, I would encourage the authors in the future to really try to understand the reviewer feedback; none of us write reviews or papers that are perfectly worded or expressed.

More crucially beyond the scores, there are a number of niggling remaining concerns, not necessarily fatal, but together put the paper as not quite ready for acceptance to a highly-selective venue like ICLR. In particular, the novelty seems to be "only" moderate; I have read the reviewers' concerns and the authors' rebuttal, but am only partially convinced. The novelty can be much more clearly articulated in the future, so that it can be judged properly; it's very common for dataset/benchmark papers to have an explicit comparison table so that the differences from previous works are seen at a glance. The concern about disentangling representation vs. execution is valid, and only partially addressed in my view; the reviewer had properly considered the rebuttal and raised more detailed counter-arguments. The authors did not fully understand the concern about floor effects; yes it's a hard task, but part of the value of a benchmark is to differentiate competing models, measure progress, etc. and not only to be difficult. (Perhaps separating out easy/medium/hard subsets of the benchmark is an easy fix).

Overall, there is certainly some value in this work, but does not meet the high bar for acceptance in its current form, especially in comparison to other submissions to ICLR 2026.

**Reviewer Concerns:**

(Pls see above)

**Reviewer Scores:**

-- XAHe explicitly maintained their score of 4.
-- I would hazard that the other 3 reviewers have a 50/50 chance of raising their scores to 6

---

### Decision · Program_Chairs · 2026-01-26

Reject